

# B-type D-branes in hybrid models

**Johanna Knapp⋆ and Robert Pryor†**

School of Mathematics and Statistics, University of Melbourne
Parkville, VIC 3010, Australia

⋆ johanna.knapp@unimelb.edu.au , † robert.pryor@unimelb.edu.au

## Abstract

A hybrid model is a fibration of a Landau-Ginzburg orbifold over a geometric base. We study B-type D-branes in hybrid models. Imposing B-type supersymmetry on the boundary action, we show that D-branes are specified by matrix factorisations in the fibre direction, together with some geometric data associated to the base. We also deduce conditions for the compatibility of these branes with the bulk orbifold actions and R-symmetry. We construct examples of hybrid B-branes which are generalisations of well-studied branes in geometric and Landau-Ginzburg models. Hybrid models can arise at limiting points of the stringy Kähler moduli space of Calabi-Yaus, and can be realised as phases of the corresponding gauged linear sigma models (GLSMs). Using GLSM techniques, we establish connections between geometric branes and hybrid branes. As explicit examples, we consider one- and two-parameter Calabi-Yau hybrids with a $\mathbb{P}^1$-base.

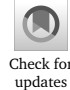

# 1 Introduction

D-branes on Calabi-Yau spaces are of central importance in many areas of string theory, ranging from string model building, to the mathematical structures behind homological mirror symmetry. How well D-branes are understood depends very much on their type, and on the region of the moduli space in which the theory is located. In geometric (large volume), regions of the moduli space, topological D-branes that preserve B-type supersymmetry are objects in the derived category of coherent sheaves associated to the Calabi-Yau. As one moves away from geometric regions in the moduli space, the corresponding branes are much less studied. A notable exception are Landau-Ginzburg models, where B-branes have been identified to be objects in the category of matrix factorisations of the Landau-Ginzburg superpotential [1, 2]. Landau-Ginzburg orbifold points can be found in "small volume", regions in the stringy Kähler moduli space of Calabi-Yau hypersurfaces. In such settings, there is a categorical equivalence between the category of matrix factorisations, and the derived category of coherent sheaves [3] which can be understood in a physics language in terms of D-brane transport in gauged linear sigma models (GLSMs) [4, 5].

The aim of this work is to study B-type D-branes in hybrid models. Hybrid models generalise non-linear sigma models and Landau-Ginzburg models in the sense that they are Landau-Ginzburg models fibred over some compact base manifold. Actions for hybrid models with $(2, 2)$ and $(0, 2)$ supersymmetry have been constructed in [6]. From a field theory perspective, a hybrid model can be understood as a special case of a non-linear sigma model with a potential. The target space of the sigma model is the total space $Y$ of a suitably chosen holomorphic vector bundle over a compact Kähler base $B$. Since $Y$ is non-compact, the model allows for a holomorphic superpotential. There are additional conditions to have a consistent CFT in the IR with well-defined R-charges. Models which satisfy these conditions are referred to as "good hybrids", or "true hybrids", as opposed to "bad", or "pseudo-hybrids" [6,7]. We consider good hybrids on a strip and impose B-type boundary conditions. Similar considerations regarding non-linear sigma models with potential which are not necessarily hybrids have been made in [8]. As in Landau-Ginzburg theories, B-type supersymmetry is not necessarily preserved on worldsheets with boundaries, leading to a hybrid version of the Warner problem [9]. This can be resolved by introducing additional degrees of freedom on the boundary, leading to matrix factorisations of the superpotential. In addition, the hybrid B-branes have a "geometric component", coming from holomorphic vector bundles and gauge connections on $B$ that can be lifted to $Y$. Concretely, hybrid B-branes can be described by a triple of data $(E, A, Q)$, where $E$ is a vector bundle on $Y$ endowed with a holomorphic structure induced by the connection $A$, which has a $(1, 1)$ curvature form, and where $Q$ is an odd holomorphic endomorphism of $E$ which squares to the hybrid superpotential. In practice, these bundles are obtained by pulling back holomorphic line bundles over $B$. In concrete constructions, a hybrid B-brane is initially described locally on a coordinate patch of $Y$. We refer to the corresponding structure as a "local matrix factorisation". We then show how to globalise this structure to form a hybrid B-brane which is defined on all of $Y$. The resulting object is called a "global matrix factorisation". The choice of vector bundle and connection is constrained by the condition that $Q$ is holomorphic, as well as by the structure of the local matrix factorisations. This is new compared to Landau-Ginzburg matrix factorisations. Furthermore, we deduce conditions under which hybrid B-branes are compatible with the R-symmetry and orbifolds.

Categories of hybrid B-branes, and in particular the term "global matrix factorisation", have been introduced in the mathematics literature in [10]. To our current level of understanding, our physics construction gives special cases of the categories described there. Even earlier accounts on D-brane categories associated to hybrid models can be found in [11,12].

Orbifolds of hybrid models naturally arise at certain loci in the stringy Kähler moduli space

of Calabi-Yaus. We use standard constructions of matrix factorisations to build specific examples of hybrid branes in two models which are linked to well-studied Calabi-Yau threefolds. This link can be established via the gauged linear sigma model (GLSM) [4], where the hybrid models and the Calabi-Yau non-linear sigma models arise in different limiting regions of the FI-theta parameter space, which can be identified with the stringy Kähler moduli space. The GLSM hemisphere partition function [13–15] computes the central charge of a GLSM B-brane. Lifting hybrid branes to the GLSM gives us access to this data. Moreover, we can take a basis of geometric branes and use the techniques of [5] to transport them to the hybrid phase. Comparing to the hybrid branes, we make a proposal for a basis of branes in these hybrid models and extract the analytic continuation matrices from the GLSM partition function.

Hybrid B-branes have also appeared in other recent works. In [16–18], B-branes in certain hybrid models have been studied in the context of homological projective duality and GLSMs. In [19], D-brane transport has been studied from a mathematical perspective for a one-parameter model that we will discuss in sections 4.2, 5.2, and appendix B. Our work gives a physics derivation of hybrid B-branes, and uses GLSMs to implement D-brane transport. To our understanding, our results agree with the results of these works.

The article is organised as follows. In section 2, we review good hybrid models without boundary, before focusing on B-type boundary conditions, deriving the Warner problem and solving it following [5,8]. This gives the physics derivation of the hybrid B-branes mentioned above. We also derive conditions for the compatibility of hybrid B-branes with the bulk R-symmetry and orbifold actions. In section 3, we show how to construct examples of hybrid B-branes in practice, and discuss how the structure of hybrid models is reflected in their properties. In particular, we show how a B-brane over a local patch on $Y$ can be extended to an object that is globally defined. Moreover, we discuss a specific class of hybrid branes described in terms of Clifford modules. We illustrate our findings in a simple toy example. In section 4, we discuss explicit constructions of B-branes in hybrids with $B = \mathbb{P}^1$ which are associated to one- and two-parameter Calabi-Yau threefolds: a one-parameter model associated to the complete intersection $\mathbb{P}^5[3,3]$, and a two-parameter model associated to the resolved hypersurface $\mathbb{P}_{11222}[8]$. We identify objects that are in some sense "canonical", and propose a set of branes that generates the RR-charge lattice. In section 5, we find GLSM lifts of our branes, and use the hemisphere partition function to transport a well-studied basis of geometric branes to the hybrid phase, thereby collecting evidence that we have indeed found a basis of hybrid branes that generates the lattice of RR-charges. Further technical details can be found in the appendix.

## 2 Action, B-type supersymmetry, and boundary conditions

In this section, we review $\mathcal{N} = (2,2)$ hybrid models following [6]. We then focus on B-type supersymmetry and worldsheets with boundary. We compute the B-type supersymmetry variation of the hybrid action on a worldsheet with boundary, which leads to the Warner problem for hybrids. By adding additional boundary degrees of freedom, we solve the Warner problem, following [5,8], and show that B-branes in hybrid models are a generalisation of the familiar matrix factorisations, referred to as global matrix factorisations.

### 2.1 Two-dimensional $\mathcal{N} = (2,2)$ hybrid models

To define a hybrid model, we begin with a target space $Y$ given by the total space of a rank-$r$ holomorphic vector bundle, $X$

$$Y = \text{tot}(X \xrightarrow{\pi} B),\tag{1}$$

where the base manifold $B$ is a $d$-dimensional compact Kähler manifold. Note that in this setup, $Y$ itself is a complex $(d+r)$-dimensional non-compact Kähler manifold. In addition, we assume that there is a holomorphic function

$$W : Y \to \mathbb{C}, \tag{2}$$

on $Y$, known as the superpotential. To define a hybrid model, we require that the critical locus of $W$ coincides with the base manifold

$$dW^{-1}(0) = B. \tag{3}$$

Following [6], we refer to a space $Y$ which can be endowed with such a superpotential as a hybrid geometry. Once a superpotential has been fixed, we then refer to the pair $(Y, W)$ as a hybrid model. The action of an $\mathcal{N} = (2,2)$ hybrid model is given by that of a supersymmetric non-linear sigma model of maps from a worldsheet $\Sigma$ into a non-compact target space given by a hybrid geometry $Y$. An action with a more general non-compact Kähler target space defines a model which we will refer to as a non-linear sigma model with potential. If the target space is $\mathbb{C}^N$ or an orbifold thereof, we refer to the resulting model as a Landau-Ginzburg model. This is equivalent to a hybrid model for which $B$ is a point. The field content of a hybrid model is as follows:

$$\begin{aligned}
\phi^\alpha &: \Sigma \to Y, \\
\psi_+^\alpha &\in \Gamma(K^{1/2} \otimes \phi^*(TY)), \\
\psi_-^\alpha &\in \Gamma(\overline{K}^{1/2} \otimes \phi^*(TY)),
\end{aligned} \tag{4}$$

where $\alpha = 1, 2, \ldots, d+r$, the worldsheet $\Sigma$ is a Riemann surface, $TY$ is the holomorphic tangent bundle of $Y$, $K = \Omega^{1,0}(\Sigma)$ is the canonical bundle of $\Sigma$, and $\overline{K} = \Omega^{0,1}(\Sigma)$ is the anti-canonical bundle of $\Sigma$. Given these fields, we consider the following action:

$$\begin{aligned}
S_{\text{hyb}} = \int_\Sigma d^2x \bigg( &- g_{\alpha\overline{\beta}} \partial^\mu \phi^\alpha \partial_\mu \overline{\phi}^{\overline{\beta}} + 2i g_{\alpha\overline{\beta}} \overline{\psi}_-^{\overline{\beta}} \overleftrightarrow{D_+} \psi_-^\alpha + 2i g_{\alpha\overline{\beta}} \overline{\psi}_+^{\overline{\beta}} \overleftrightarrow{D_-} \psi_+^\alpha + R_{\alpha\overline{\beta}\gamma\overline{\delta}} \psi_+^\alpha \psi_-^\gamma \overline{\psi}_-^{\overline{\beta}} \overline{\psi}_+^{\overline{\delta}} \\
&- \frac{1}{4} g^{\alpha\overline{\beta}} \partial_\alpha W \partial_{\overline{\beta}} \overline{W} - \frac{1}{2} D_\alpha \partial_\beta W \psi_+^\alpha \psi_-^\beta - \frac{1}{2} D_{\overline{\alpha}} \partial_{\overline{\beta}} \overline{W} \overline{\psi}_-^{\overline{\alpha}} \overline{\psi}_+^{\overline{\beta}} \bigg),
\end{aligned} \tag{5}$$

where we have introduced the light cone coordinates $x^\pm = x^0 \pm x^1$, and integrated out the auxiliary fields. Furthermore, we have defined $D_\alpha$ to be the covariant derivative associated with the Levi-Civita connection on $Y$, while $D_\pm$ is the covariant derivative realised by the pullback of this connection along $\phi$

$$\begin{aligned}
D_+ \psi_\pm^\alpha &= \partial_+ \psi_\pm^\alpha + \frac{\partial \phi^\beta}{\partial x^+} \Gamma_{\beta\gamma}^\alpha \psi_\pm^\gamma, \\
D_- \psi_\pm^\alpha &= \partial_- \psi_\pm^\alpha + \frac{\partial \phi^\beta}{\partial x^-} \Gamma_{\beta\gamma}^\alpha \psi_\pm^\gamma,
\end{aligned} \tag{6}$$

where $\Gamma_{\beta\gamma}^\alpha$ are the Christoffel symbols associated to the Levi-Civita connection on $Y$. It is sometimes convenient to distinguish base and fibre fields. For the scalars $\phi^\alpha$ we write

$$\phi^\alpha = \{y^I; \varphi^i\}, \text{ where } I = 1, \ldots, d, \text{ and } i = 1, \ldots, r. [1] \tag{7}$$

We can now explain the motivation for the potential condition (3). The action (5) contains a bosonic potential term

$$V_{bos} \propto \int_\Sigma d^2x\, g^{\alpha\overline{\beta}} \partial_\alpha W \partial_{\overline{\beta}} \overline{W}, \tag{8}$$

---

[1]For specific models, $B$ is often a complex projective space $\mathbb{P}^n$. In this situation, we are instead lead to introduce homogeneous coordinates $[z_0 : \ldots : z_n]$ on $B$, and denote the fibre coordinates by $x_i$.

which will lead to the suppression of fluctuations of fields away from the locus on which $dW = 0$. The potential condition then ensures that this locus is exactly $B$. One can show that the action (5) is invariant under the following supersymmetry transformations:

$$\delta\phi^\alpha = \epsilon_+\psi_-^\alpha - \epsilon_-\psi_+^\alpha,$$
$$\delta\psi_\pm^\alpha = \pm 2i\overline{\epsilon}_\mp\partial_\pm\phi^\alpha + \epsilon_\pm\left(\Gamma_{\beta\gamma}^\alpha\psi_+^\beta\psi_-^\gamma - \frac{1}{2}g^{\alpha\overline{\beta}}\partial_{\overline{\beta}}\overline{W}\right), \tag{9}$$

where the $\mathcal{N} = (2,2)$ supersymmetry variations are given by

$$\delta = \epsilon_+Q_- - \epsilon_-Q_+ - \overline{\epsilon}_+\overline{Q}_- + \overline{\epsilon}_-\overline{Q}_+, \tag{10}$$

for supercharges $Q_\pm$, and $\overline{Q}_\pm$, and complex Grassmann parameters $\epsilon_\pm$, and $\overline{\epsilon}_\pm$.

So far, we have not utilised the fibration structure of the hybrid models in question. This structure becomes relevant when we impose the condition that the CFT in the IR has well-defined left and right R-charges. Theories with this property are referred to as "good hybrids" [6,7]. In section 2.4 below, we will show that the conditions for good hybrids are compatible with the presence of B-type boundary conditions. To start off, we recall the definitions of the axial and vector R-symmetries, as well as the left and right R-symmetries, and the relationship between the two. The various R-symmetries are specified by representations of $U(1)$ with the following forms

$$\rho_{V,A} : U(1)_{V,A} \to \mathbb{C}^*, \quad e^{i\theta} \mapsto e^{iq_{V,A}\theta}, \quad q_{V,A} \in \mathbb{Z}, \tag{11}$$

$$\rho_{L,R} : U(1)_{L,R} \to \mathbb{C}^*, \quad e^{i\theta} \mapsto e^{iq_{L,R}\theta}, \quad q_{L,R} \in \mathbb{Z}. \tag{12}$$

The axial and vector R-symmetries are then determined in terms of the left and right R-symmetries by the following relationship between the weights (i.e. the R-charges), of the corresponding representations

$$q_V = q_L + q_R, \text{ and } q_A = -q_L + q_R. \tag{13}$$

In an ordinary compact non-linear sigma model, R-symmetry is realised by assigning left and right R-charges of zero to the chiral multiplets. This amounts to the following infinitesimal transformations of the constituent fields

$$\delta_L\phi^\alpha = 0, \quad \delta_L\psi_+^\alpha = 0, \quad \delta_L\psi_-^\alpha = -i\alpha\psi_-^\alpha,$$
$$\delta_R\phi^\alpha = 0, \quad \delta_R\psi_-^i = 0, \quad \delta_R\psi_+^\alpha = -i\alpha\psi_+^\alpha, \tag{14}$$

where $\alpha$ is a real parameter. However, due to the non-trivial scaling behaviour of the potential $W$, this R-symmetry does not hold for a hybrid model. To remedy this, we require the existence of a vertical holomorphic Killing vector field $V$ on $Y$, under which $W$ is homogeneous

$$\mathcal{L}_V W = W, \text{ with } \mathcal{L}_V \pi^*\omega = 0, \text{ for all } \omega \in \Omega^\bullet(B). \tag{15}$$

This is known as the good hybrid condition, and models satisfying it are known as good hybrids [6]. From now on, we will only consider good hybrids. For good hybrids, one can show that $V$ can be written as [6]

$$V = \sum_{i=1}^n q_i\varphi^i\partial_{\varphi^i}, \tag{16}$$

for some charges $q_i \in \mathbb{Q}_{\geq 0}$. We then define infinitesimal transformations $\delta_{KV}$ under the action of the Killing vector $V$

$$\delta_{KV}\phi^\alpha = i\alpha\mathcal{L}_V\phi^\alpha, \text{ and } \delta_{KV}\psi_\pm^\alpha = i\alpha V_{,\beta}^\alpha\psi_\pm^\beta,$$
$$\delta_{KV}\overline{\phi}^{\overline{\alpha}} = -i\alpha\mathcal{L}_{\overline{V}}\overline{\phi}^{\overline{\alpha}}, \text{ and } \delta_{KV}\overline{\psi}_\pm^{\overline{\alpha}} = -i\alpha\overline{V}_{,\overline{\beta}}^{\overline{\alpha}}\overline{\psi}_\pm^{\overline{\beta}}, \tag{17}$$

where $V^{\alpha}_{,\beta}$ denotes the partial derivative $\partial_{\phi^{\beta}} V^{\alpha}$. These transformations can then be combined with the original R-symmetry transformations (14) by defining the following augmented R-symmetry transformations

$$\delta'_L = \delta_L + \delta_{KV}, \text{ and } \delta'_R = \delta_R + \delta_{KV}. \tag{18}$$

One then finds that R-symmetry is restored at the classical level, and in particular

$$\delta'_L S_{\text{hyb}} = \delta'_R S_{\text{hyb}} = 0. \tag{19}$$

The vector R-symmetry is automatically preserved at the quantum level. However, to retain a non-anomalous axial R-symmetry, we have to further impose that $c_1(T_Y) = 0$, which holds when $Y$ is a non-compact Calabi-Yau. For all examples we will consider, the canonical bundle of $Y$ will be trivial, which is a stronger condition.

### 2.1.1 An example of a good hybrid

A simple example of a good hybrid can be modelled on the following geometry [6]:[2]

$$Y = \text{tot}(\mathcal{O}(-2) \to \mathbb{P}^1). \tag{20}$$

To define a hybrid geometry, we need to fix a superpotential. Since $Y$ is a non-trivial manifold, specifying $W$ amounts to specifying holomorphic functions in each local coordinate chart which are compatible with the transition functions of $Y$. To construct such a $W$, we utilise the local coordinates introduced in appendix A.1. In particular, we have coordinates $(u, \varphi_u)$ in the chart $\mathcal{U}_0$ and $(v, \varphi_v)$ in the chart $\mathcal{U}_1$. These coordinates are related as follows

$$u = v^{-1}, \text{ and } \varphi_u = v^2 \varphi_v. \tag{21}$$

We can then define the following local superpotentials

$$W_u = (1 + u^{2k+2})\varphi_u^{k+1}, \quad W_v = (v^{2k+2} + 1)\varphi_v^{k+1}, \quad k \in \mathbb{Z}_{\geq 1}, \tag{22}$$

which can be shown to define a global superpotential which obeys the potential condition (3). Motivation for considering this particular hybrid model comes from several directions. Firstly, we note that it can be interpreted as a hybrid generalisation of the Landau-Ginzburg description of an $A$-type minimal model [2, 20, 21]. Additionally, we note that for $k = 3$, it can also be seen as a simpler toy model for the octic hybrid which we will consider in section 4.1.

We claim that this model is a good hybrid. To see this, we first note that $Y$ possesses a vertical holomorphic Killing vector

$$V = \frac{1}{k+1} \varphi_u \partial_{\varphi_u}, \tag{23}$$

and thus the corresponding hybrid model has a non-anomalous vector R-symmetry. To see that we also have a non-anomalous axial R-symmetry, note that the volume elements in $\mathcal{U}_0$ and $\mathcal{U}_1$ are related as follows

$$du \wedge d\varphi_u = -dv \wedge d\varphi_v, \tag{24}$$

and so the canonical bundle of $Y$ is trivial. This then implies that $Y$ is Calabi-Yau.

The fact that the geometry $Y$ is Calabi-Yau does not necessarily mean that the model itself is Calabi-Yau. We say a hybrid model is Calabi-Yau if it lies in the stringy Kähler moduli space of a Calabi-Yau manifold. This is not the case for the model $(Y, W)$, even though the target

---

[2]Note that $Y$ can be realised as the total space of the cotangent bundle of $\mathbb{P}^1$.

space $Y$ is itself a (non-compact), Calabi-Yau. This is analogous to the fact that although $\mathbb{C}^n$ is trivially Calabi-Yau, not all Landau-Ginzburg models on $\mathbb{C}^n$ are Calabi-Yau.

We are also interested in orbifolded hybrid models, as these are often the models which are actually Calabi-Yau. From now on, we will consider the $\mathbb{Z}_{k+1}$-orbifold of this model which is generated by $e^{2\pi i J_0}$, where $J_0$ is the generator of the left-moving R-symmetry (18). Explicitly, this transformation acts on the local coordinates $(u, \varphi_u)$ as $\gamma \cdot (u, \varphi_u) = (u, \gamma \varphi_u)$ with $\gamma \in \mathbb{Z}_{k+1}$.

## 2.2 B-type supersymmetry and boundary terms

We consider properties of hybrid models with boundary in the presence of B-type supersymmetry. We show that in this setting, B-type boundary conditions correspond to a globally defined generalisation of the matrix factorisations familiar from the study of Landau-Ginzburg models. A similar setting for non-linear sigma models with potential has been considered in [8, 22]. Our results are consistent with those presented there. We will focus on the properties which are due to the fibration structure of hybrid models, and their implications on the form of the B-type boundary conditions.

As our worldsheet, we consider a flat strip

$$(x^0, x^1) \in \Sigma = \mathbb{R} \times [0, \pi].^3 \tag{25}$$

Generalisations to other worldsheet topologies can be achieved by passing to the B-twisted setting [22]. B-type supersymmetry is then defined by restricting to the $\mathcal{N} = 2_B$ subalgebra of the full $\mathcal{N} = (2, 2)$ supersymmetry algebra. This means setting $\epsilon_+ = -\epsilon_- := \epsilon$ in (9) so that

$$\delta = \epsilon \overline{Q} - \overline{\epsilon} Q, \quad \text{where} \quad Q = \overline{Q}_+ + \overline{Q}_-. \tag{26}$$

The supersymmetry transformations of the bulk fields then reduce to

$$\begin{aligned}
\delta \phi^\alpha &= \epsilon (\psi_+^\alpha + \psi_-^\alpha), \\
\delta \psi_\pm^\alpha &= -2i\overline{\epsilon} \partial_\pm \phi^\alpha \pm \epsilon \left( \Gamma_{\beta\gamma}^\alpha \psi_+^\beta \psi_-^\gamma - \frac{1}{2} g^{\alpha\overline{\beta}} \partial_{\overline{\beta}} \overline{W} \right).
\end{aligned} \tag{27}$$

It turns out to be convenient to make the following field redefinitions

$$\eta^\alpha = \psi_+^\alpha + \psi_-^\alpha, \quad \theta^\alpha = \psi_-^\alpha - \psi_+^\alpha. \tag{28}$$

Computing the variation of the action, the following boundary terms appear

$$\delta S_{hyb} = \epsilon \int_{\mathbb{R}} dx^0 \left( \frac{i}{2} \overline{\eta}^{\overline{\alpha}} \partial_{\overline{\alpha}} \overline{W} - g_{\alpha\overline{\beta}} (\partial_1 \overline{\phi}^{\overline{\beta}} \eta^\alpha + \partial_0 \overline{\phi}^{\overline{\beta}} \theta^\alpha) \right) \Big|_0^\pi + \text{c.c.} \tag{29}$$

The hybrid action is thus not supersymmetric on the boundary of $\Sigma$. This can be cured by adding boundary terms to the action. To cancel the terms from $\delta S_{hyb}$, we first add a standard boundary term that consists of bulk fields restricted to the boundary

$$S_{\partial\Sigma,\psi} = -\frac{i}{2} \int dx^0 g_{\alpha\overline{\beta}} \left( \psi_+^\alpha \overline{\psi}_-^{\overline{\beta}} + \overline{\psi}_+^{\overline{\beta}} \psi_-^\alpha \right) \Big|_0^\pi = \frac{i}{4} \int dx^0 \left( g_{\alpha\overline{\beta}} (\overline{\theta}^{\overline{\beta}} \eta^\alpha - \overline{\eta}^{\overline{\beta}} \theta^\alpha) \right) \Big|_0^\pi. \tag{30}$$

The combined variation of the bulk action and the standard boundary term is

$$\delta(S_{\text{hyb}} + S_{\partial\Sigma,\psi}) = \frac{i}{2} \int dx^0 \left( \epsilon \overline{\eta}^{\overline{\alpha}} \partial_{\overline{\alpha}} \overline{W} + \overline{\epsilon} \eta^\alpha \partial_\alpha W \right) \Big|_0^\pi = \frac{i\epsilon}{2} \int dx^0 \left( \overline{\eta}^{\overline{\alpha}} \partial_{\overline{\alpha}} \overline{W} \right) \Big|_0^\pi + \text{c.c.} \tag{31}$$

This is a hybrid generalisation of the well-known Warner term [9] familiar from the study of Landau-Ginzburg models [1, 2]. To cancel this term, and restore $\mathcal{N} = 2_B$ supersymmetry on $\partial\Sigma$, we must introduce additional degrees of freedom living on the boundary. Analysing the possible interactions involving such boundary degrees of freedom will eventually lead us to the notion of a hybrid B-brane.

---

[3] Depending on context, we will denote the time-like coordinate on $\Sigma$ either by $x^0$ or by $t$.

### 2.3 Solving the Warner problem for hybrids

The original approach to solve the Warner problem in Landau-Ginzburg models taken in [1, 2] was to place new degrees of freedom on the boundary which were components of a fermionic non-chiral boundary superfield. Imposing Dirichlet and/or Neumann boundary conditions then led to a solution of the Warner problem in terms of matrix factorisations of the superpotential involving boundary fermions. We will instead follow [5, 8, 22], and introduce a boundary interaction term which will lead to the matrix factorisation condition without the need to choose specific boundary conditions on the worldsheet, or to introduce explicit boundary fermions. This will lead us to a formulation of hybrid B-branes involving a generalisation of the familiar matrix factorisations. In particular, instead of a factorisation of a polynomial superpotential in terms of the endomorphisms of the Chan-Paton space, these branes will be specified by factorisations of a holomorphic function in terms of global sections of the endomorphism bundle of a certain vector bundle. We refer to this structure as a global matrix factorisation to emphasise this difference.

To cancel the hybrid Warner term, we first introduce a $\mathbb{Z}_2$-graded rank $k$ vector bundle $E = E^{ev} \oplus E^{odd}$ over $Y$, which we refer to as the Chan-Paton bundle. We equip $E$ with a connection $A$, and an odd endomorphism $Q \in \text{End}^{odd}(E)$. Building $\mathcal{N} = 2_B$-restoring boundary interactions will lead to further constraints on this data. Motivated by the boundary interaction utilised in [5] and [8], we try the following general boundary interaction

$$
\begin{aligned}
\mathcal{A}_t = &\dot{\phi}^\alpha A_\alpha + \dot{\overline{\phi}}^{\overline{\alpha}} A_{\overline{\alpha}} - \frac{i}{4}\left(F_{\alpha\beta}\eta^\alpha\eta^\beta + 2F_{\alpha\overline{\beta}}\eta^\alpha\overline{\eta}^{\overline{\beta}} + F_{\overline{\alpha}\overline{\beta}}\overline{\eta}^{\overline{\alpha}}\overline{\eta}^{\overline{\beta}}\right) \\
&+ \frac{1}{2}\left(\eta^\alpha D_\alpha(\overline{Q} - Q) + \overline{\eta}^{\overline{\alpha}}D_{\overline{\alpha}}(\overline{Q} - Q)\right) + \frac{1}{2}\{Q, \overline{Q}\},
\end{aligned}
\tag{32}
$$

where $F_{\alpha\beta} = \partial_\alpha A_\beta - \partial_\beta A_\alpha + i[A_\alpha, A_\beta]$ is the curvature of the connection $A$. One could also begin with a slightly more general boundary interaction in which the term $\{Q, \overline{Q}\}$ is swapped for $T^2$, with $T = i(Q + \overline{Q})$. The quantity $T$ can be interpreted physically as a tachyon profile [5]. This approach then leads to an extra term in $\mathcal{A}_t$ proportional to $\text{Re}(W)$. A similar term is encountered when studying Landau-Ginzburg models, and is usually dropped.

The interaction $\mathcal{A}_t$ should be considered as being inserted into the path integral of the theory analogously to a Wilson line

$$
Z = \int \mathcal{D}[\phi]\mathcal{D}[\psi]\mathcal{D}[A]Pe^{-i\int_{\partial\Sigma}dt\mathcal{A}_t}e^{-(S_{\text{hyb}}+S_{\partial\Sigma,\psi})}.
\tag{33}
$$

To solve the Warner problem, we aim to find conditions on the triple $(E, A, Q)$, such that the $\mathcal{N} = 2_B$ variation of our modified action vanishes, thus restoring boundary supersymmetry. This requires that the $\mathcal{N} = 2_B$ variation of $\mathcal{A}_t$ cancels that of $S_{hyb} + S_{\partial\Sigma,\psi}$ up to a total covariant derivative with respect to the boundary interaction $\mathcal{A}_t$ defined by

$$
\mathcal{D}_t X = \partial_t X + i[\mathcal{A}_t, X].
\tag{34}
$$

A lengthy but straightforward calculation shows that the boundary interaction has the following $\mathcal{N} = 2_B$ transformation

$$
\begin{aligned}
\delta\mathcal{A}_t = &i\mathcal{D}_t(\epsilon(\overline{Q} - i\eta^\alpha A_\alpha)) + \epsilon\Bigg(-i(\dot{\overline{\phi}}^{\bar{\alpha}}D_{\bar{\alpha}}Q + \dot{\phi}^\alpha D_\alpha\overline{Q}) + \frac{1}{2}\{D_\alpha\overline{Q}, \overline{Q}\}\eta^\alpha + \dot{\phi}^\beta F_{\alpha\beta}\eta^\alpha \\
&- \dot{\overline{\phi}}^{\overline{\alpha}}F_{\overline{\alpha}\overline{\beta}}\overline{\eta}^{\overline{\beta}} - \frac{1}{2}\{D_{\overline{\beta}}Q, \overline{Q}\}\overline{\eta}^{\overline{\beta}} + \frac{1}{2}D_{\overline{\beta}}\overline{Q}^2\overline{\eta}^{\overline{\beta}} + \frac{i}{4}[Q, F_{\alpha\beta}]\eta^\alpha\eta^\beta + \frac{i}{4}[Q, F_{\overline{\alpha}\overline{\beta}}]\overline{\eta}^{\overline{\alpha}}\overline{\eta}^{\overline{\beta}} \\
&+ \left(D_{\overline{\beta}}D_\alpha\overline{Q} - D_\alpha D_{\overline{\beta}}Q\right)\eta^\alpha\overline{\eta}^{\overline{\beta}}\Bigg) - i\dot{\epsilon}\overline{Q} + \text{c.c},
\end{aligned}
\tag{35}
$$

where we have considered a $t$-dependent parameter $\epsilon$ so as to be able to read off the contribution of $\mathcal{A}_t$ to the boundary Noether charges associated to the $\mathcal{N} = 2_B$ supersymmetry. The $\mathcal{N} = 2_B$ transformation of the total action is then

$$\delta\left(-i\int_{\partial\Sigma}\mathcal{A}_t + S_{hyb} + S_{\partial\Sigma,\psi}\right) = \int dx^0\left(\epsilon\left(\frac{i}{2}\overline{\eta}^{\overline{\alpha}}D_{\overline{\alpha}}\left(\overline{W} - \overline{Q}^2\right) - (\dot{\overline{\phi}}^{\dot{\overline{\alpha}}}D_{\dot{\alpha}}Q + \dot{\phi}^\alpha D_\alpha\overline{Q})\right.\right.$$
$$\left.\left. -\frac{i}{2}\{D_\alpha\overline{Q},\overline{Q}\}\eta^\alpha + i\dot{\phi}^\alpha F_{\alpha\beta}\eta^\beta + i\dot{\overline{\phi}}^{\overline{\alpha}}F_{\overline{\alpha}\overline{\beta}}\overline{\eta}^{\overline{\beta}} + \frac{i}{2}\{D_{\overline{\beta}}Q,\overline{Q}\}\overline{\eta}^{\overline{\beta}} + \frac{1}{4}[Q,F_{\alpha\beta}]\eta^\alpha\eta^\beta\right.\right. \quad (36)$$
$$\left.\left. + \frac{1}{4}[Q,F_{\overline{\alpha}\overline{\beta}}]\eta^{\overline{\alpha}}\eta^{\overline{\beta}} - i\left(D_{\overline{\beta}}D_\alpha\overline{Q} - D_\alpha D_{\overline{\beta}}Q\right)\eta^\alpha\overline{\eta}^{\overline{\beta}}\right)\right)\Bigg|_0^\pi + \text{c.c.},$$

where we have taken $\epsilon$ to be $t$-independent, and implicitly discarded the total covariant derivative terms. We thus see that boundary supersymmetry is restored if the following three conditions, along with their complex conjugates, are met:

$$F_{\alpha\beta} = 0, \quad (37)$$

$$D_\alpha(W - Q^2) = 0, \quad (38)$$

$$D_{\overline{\alpha}}Q = 0. \quad (39)$$

The first of these conditions is satisfied if the connection $A$ furnishes a curvature tensor of type $(1,1)$. This is the same condition obtained in the non-linear sigma model case [5, 23]. The Dolbeault operator corresponding to a connection on a vector bundle with a $(1,1)$ curvature form defines a holomorphic structure on said bundle, since

$$(\overline{\partial}_A)^2 = -i[\overline{D},\overline{D}] = F^{(0,2)} = 0. \quad (40)$$

We can thus interpret the pair $(E,A)$ as a holomorphic vector bundle $\mathcal{E}$ on $Y$. The remaining conditions are satisfied if $Q$ is holomorphic with respect to this holomorphic structure on $\mathcal{E}$, as well as satisfying the following condition

$$Q^2 = W\mathbb{1}_E.^4 \quad (41)$$

In the setting of Landau-Ginzburg models, such an endomorphism is called a matrix factorisation of $W$. The essential difference in the current setting, however, is that our target space is a hybrid geometry, rather than $\mathbb{C}^n$ or some orbifold thereof. This means that the condition (41) must hold over the entire fibration $Y$. We refer to such an object as a global matrix factorisation, while an endomorphism satisfying (41) in a local chart of $Y$ is called a local matrix factorisation. We remark that an ordinary matrix factorisation in a Landau-Ginzburg model is also global in a rather trivial way. A Landau-Ginzburg model can be considered as a hybrid model with $B = \{pt\}$, and so in this case being globally defined is equivalent to being defined at a single point.

As a sanity check, suppose that $B = \{pt\}$, so that $Y = \mathbb{C}^n$. In this case, we hope to recover the usual notion of a Landau-Ginzburg B-brane realised as a matrix factorisation. Since $Y$ is contractible in this case, $E$ must be a trivial bundle. We can thus specify the connection $A$ globally using only a single local trivialisation. Now since

$$\overline{\partial}_A(A_{\overline{\alpha}}d\overline{x}^{\overline{\alpha}}) = \frac{1}{2}F_{\overline{\alpha}\overline{\beta}}d\overline{x}^{\overline{\alpha}} \wedge d\overline{x}^{\overline{\beta}} = 0, \quad (42)$$

we have by the $\overline{\partial}_A$-Poincaré lemma that $A_{\overline{\alpha}}d\overline{x}^{\overline{\alpha}} = \overline{\partial}\Lambda$ for some function $\Lambda$ valued in the Lie algebra of the boundary gauge group. We can then make a gauge transformation with parameter $-i\Lambda$ to set $A_{\overline{\alpha}} = 0$, and thus $A = 0$. We can perform this procedure locally for an arbitrary

---

[4]We could also have $Q^2 = W + c$ for some constant $c$, but we will see in section 2.4 that we necessarily have $c = 0$ for R-graded D-branes.

hybrid B-brane, but for the case at hand, the triviality of $E$ implies that it holds globally. We thus see that the gauge part of a hybrid $B$-brane on $Y = \mathbb{C}^n$ is trivial, and so we are left with precisely the definition of a Landau-Ginzburg B-brane, as expected.

We conclude that B-branes in hybrid models are represented by triples $(E, A, Q)$, where $E$ and $A$ define a holomorphic $\mathbb{Z}_2$-graded vector bundle over the target space, $Y$, and where $Q$ is a global matrix factorisation of the hybrid superpotential $W$.

## 2.4 Boundary R-symmetry

Having introduced new degrees of freedom on the boundary, we have to show that they can be assigned R-charges which are consistent with the bulk R-charge structure which defines good hybrid models. We do so by demanding consistency with the known R-charges of the bulk fields, and the superpotential. We denote the action of the total vector R-symmetry on the bosonic bulk fields by

$$R_\lambda \cdot \phi = e^{\delta_V + 2\delta_{KV}} \phi = (y^I, \lambda^{2q_i} \varphi^i), \tag{43}$$

where $\lambda = e^{i\alpha} \in U(1)$. Then recalling that the superpotential has weight 2 under the bulk vector R-symmetry, we have

$$W(R_\lambda \cdot \phi) = \lambda^2 W(\phi). \tag{44}$$

Now, since $W$ has vector R-charge $+2$, the matrix factorisation condition $Q^2 = W \mathbb{1}_E$ implies that $Q$ must have vector R-charge $+1$. Noting that $Q$ is an endomorphism of the Chan-Paton bundle, we see that the boundary R-charge acts on it in the adjoint representation, and so

$$R(\lambda) Q(R_\lambda \cdot \phi) R(\lambda)^{-1} = \lambda Q(\phi). \tag{45}$$

Combining the transformations for $Q$ and $W$, we see that for all $\lambda \in U(1)$

$$\begin{aligned}
&R(\lambda) Q(R_\lambda \cdot \phi)^2 R(\lambda)^{-1} = \lambda^2 Q(\phi)^2 \\
\Rightarrow &R(\lambda)(W(R_\lambda \cdot \phi) + c) R(\lambda)^{-1} = \lambda^2 (W(\phi) + c) \mathbb{1}_E \\
\Rightarrow &\lambda^2 W(\phi) + c = \lambda^2 W(\phi) + \lambda^2 c \\
\Rightarrow &c = \lambda^2 c,
\end{aligned} \tag{46}$$

and thus conclude that $c = 0$ is a necessary condition for an R-symmetric hybrid brane, as claimed previously. We now note that (32) includes the term $\{Q, Q^\dagger\}$. From (45) we know that under an R-symmetry transformation with parameter $\lambda \in U(1)$, this term this transforms as

$$\{Q, Q^\dagger\} \mapsto |\lambda|^2 \{Q, Q^\dagger\} = \{Q, Q^\dagger\}. \tag{47}$$

This implies that $\mathcal{A}_t$ is invariant under boundary R-symmetry transformations. We now use this fact to determine the R-transformation of the gauge field $A$. To do so, we compute the R-transformation of the term $\dot{\phi}^\alpha A_\alpha$ in (32)

$$R_\lambda \cdot \left( \dot{\phi}^\alpha A_\alpha \right) = (\lambda^{2q_\alpha} \dot{\phi}^\alpha) R(\lambda) A_\alpha (R_\lambda \cdot \phi) R(\lambda)^{-1}, \tag{48}$$

which implies that we must have the transformation property

$$R(\lambda) A_\alpha (R_\lambda \cdot \phi) R(\lambda)^{-1} = \lambda^{-2q_\alpha} A_\alpha(\phi), \tag{49}$$

where it is to be understood that $q_I = 0$ for the base coordinates. It is straightforward to check that the remaining terms of $\mathcal{A}_t$ are compatible with, and fully determined by, these

transformation properties. In summary, the boundary interaction is R-symmetric if $E$ admits a $U(1)$ action satisfying the following properties

$$R(\lambda)Q(R_\lambda \cdot \phi)R(\lambda)^{-1} = \lambda Q(\phi),$$
$$R(\lambda)A_\alpha(R_\lambda \cdot \phi)R(\lambda)^{-1} = \lambda^{-2q_\alpha}A_\alpha(\phi). \tag{50}$$

We call a hybrid B-brane satisfying the conditions (50) R-graded, and represent it by a quadruple $(E, A, Q, R)$.

## 2.5 Orbifolds

We now consider the case of an orbifolded theory. The canonical orbifold of a hybrid model is that generated by $U(1)_L$ acting on the bulk fields as $e^{2\pi i J_0}$, where $J_0$ is the conserved charge associated to $U(1)_L$ [6]. In general, there can also be different orbifold actions. Here we will deduce the necessary conditions for well-defined branes in orbifolded hybrid models without focusing on a specific choice of orbifold.

Suppose a hybrid model is orbifolded by a finite cyclic group $\Gamma \cong \mathbb{Z}_d$ for some $d \in \mathbb{Z}_{\geq 2}$. We write the action of $\Gamma$ on the bulk coordinates as

$$\gamma \cdot \phi = (y^I, \gamma^{\omega_i}\varphi^i), \text{ where } \gamma \in \Gamma, \text{ and } \omega_i = 0, 1, \dots, d-1. \tag{51}$$

Now, consider an R-graded hybrid B-brane $(E, A, Q, R)$. We know how $\Gamma$ acts on the bulk degrees of freedom, but for the brane to inherit the orbifold invariance of the bulk theory, we must pick some representation of $\Gamma$ on the Chan-Paton bundle under which $Q$ is invariant. Calling this representation $\rho$, and recalling that $W$ is an endomorphism of $E$, we see that the required condition on $Q$ is

$$\rho(\gamma)^{-1}Q(\gamma \cdot \phi)\rho(\gamma) = Q(\phi), \quad \gamma \in \Gamma. \tag{52}$$

This implies that $\mathcal{A}_t$ should be invariant under orbifold transformations. We now use this fact to determine the transformation properties of the gauge field $A$ under orbifold transformations. To do so, we compute the orbifold transformation of the term $\dot{\phi}^\alpha A_\alpha$ in (32)

$$\gamma \cdot (\dot{\phi}^\alpha A_\alpha) = (\gamma^{\omega_\alpha}\dot{\phi}^\alpha)\rho(\gamma)^{-1}A_\alpha(\gamma \cdot \phi)\rho(\gamma), \tag{53}$$

which implies that $A$ must have the transformation property

$$\rho(\gamma)^{-1}A_\alpha(\gamma \cdot \phi)\rho(\gamma) = \gamma^{d-\omega_\alpha}A_\alpha(\phi), \text{ for all } \gamma \in \Gamma. \tag{54}$$

An argument analogous to that in [24] then shows that in addition, $R$ and $\rho$ should commute

$$R(\lambda)\rho(\gamma) = \rho(\gamma)R(\lambda), \text{ for all } \gamma \in \Gamma, \text{ and } \lambda \in U(1). \tag{55}$$

Furthermore, charge integrality is guaranteed in the open string sector of an orbifolded hybrid B-model if the orbifold and R-symmetry actions are compatible with the underlying $\mathbb{Z}_2$ grading of $E$, namely

$$R(e^{i\pi})\rho(e^{\frac{2\pi i}{d}}) = \sigma_E, \tag{56}$$

where $\sigma_E$ is the $\mathbb{Z}_2$-grading of $E$. We can now represent an R-graded B-brane in an orbifolded hybrid model by a tuple $\mathfrak{B} = (E, A, Q, \rho, R)$.

In summary, we have found that a hybrid B-brane is a global matrix factorisation $Q$ over a holomorphic vector bundle $(E, A)$. Such a brane is R-graded if this bundle admits a $U(1)$ action under which $Q$ and $A$ transform as per (50). It is compatible with an orbifold of the model by a cyclic group $\Gamma$ if this bundle admits a representation of $\Gamma$ under which $Q$ and $A$ transform according to (52) and (54). The orbifold and R transformations are themselves compatible if they commute, (55), and ensure charge integrality if they are compatible with the $\mathbb{Z}_2$-grading of $E$ (56).

# 3 Construction of hybrid B-branes

In this section, we will discuss the construction of B-branes in general hybrid models defined on a hybrid geometry $Y = \text{tot}(X \to \mathbb{P}^n)$. To do so, we will first focus on how to explicitly describe holomorphic line bundles on $Y$, as well as connections with a $(1,1)$ curvature form. We then describe a method for constructing global matrix factorisations by gluing together local matrix factorisations with Clifford algebra realisations. Finally, we will obtain conditions on our global matrix factorisations which ensure R-gradability and orbifold invariance. Throughout this section, we work mostly with hybrid B-branes constructed from holomorphic line bundles, $\mathcal{O}(n)$ with $n \in \mathbb{Z}$. Our constructions do however generalise to the case of orbibundles of the form $\mathcal{O}\left(\frac{m}{n}\right)$ with $n, m \in \mathbb{Z}$ coprime. This generalisation will be important for the example considered in section 4.2, and is discussed in some more detail in section 3.6.3.

## 3.1 Holomorphic line bundles and connections over hybrid geometries

To describe the holomorphic line bundles over $Y$, we recall that any holomorphic line bundle on $B = \mathbb{P}^n$ is isomorphic to $\mathcal{O}(k)$ for some $k \in \mathbb{Z}$ (see e.g [25]). To construct line bundles over the total space $Y$, we pull these line bundles back by the projection map $\pi : Y \to \mathbb{P}^n$, as depicted in the following diagram:

$$\begin{array}{ccc} \pi^*\mathcal{O}(k) & \longrightarrow & \mathcal{O}(k) \\ \downarrow & & \downarrow \pi_E \\ Y & \xrightarrow{\ \pi\ } & \mathbb{P}^n \end{array} \tag{57}$$

All holomorphic line bundles on $Y$ arise in this way. To see this, recall that the Picard group of the total space of a vector bundle is isomorphic to that of its base (see e.g [26]), and so $\text{Pic}(Y) \cong \text{Pic}(\mathbb{P}^n) \cong \mathbb{Z}$. We thus consider Chan-Paton bundles of the form

$$E \cong \pi^*\mathcal{O}(k_1) \oplus \pi^*\mathcal{O}(k_2) \oplus \ldots \oplus \pi^*\mathcal{O}(k_{2\ell}), \quad k_i \in \mathbb{Z}. \tag{58}$$

To obtain connections on $E$ which have $(1,1)$ curvature forms, we will pull back the Chern connection on $B$ along the projection map $\pi$. We will explicitly describe this procedure for a single line bundle $E = \mathcal{O}(k)$, since the generalisation to a sum of line bundles is straightforward. To begin, we recall that a Hermitian metric on a vector bundle $E$ is a section of $(E \otimes \overline{E})^*$ which restricts to a Hermitian inner product on the fibres of $E$.[5]

We first review the construction of the Chern connection on $E$. Recall that the bundle $E$ is specified by the local trivialisations $\mathcal{U}_i$, and the transition functions $\tau_{i,i+1}$ with $i = 0, 1, \ldots, n$ (see appendix A.2). We then introduce local coordinates $(u_1, \ldots, u_n)$, and $(v_1, \ldots, v_n)$ for the charts $U_i$ and $U_{i+1}$ of $\mathbb{P}^n$.[6] The coordinate change between these charts is (as per section A.1)

$$u_j = v_j v_i^{-1}, \text{ for } i \neq j, \text{ and } u_i = v_i^{-1}. \tag{59}$$

Since $(E \otimes \overline{E})^*$ is a holomorphic line bundle, its fibres are $\mathbb{C}$. This means that a section of $(E \otimes \overline{E})^*$ is completely specified by a set of functions $s_i : \mathbb{C}^n \to \mathbb{C}$ which are compatible with its transition functions

$$s_{i+1} = \tau_{i,i+1} \overline{\tau}_{i,i+1} s_i, \quad i = 0, 1, \ldots, n. \tag{60}$$

---

[5]We denote the complex conjugate of the bundle $E$ by $\overline{E}$, and the dual bundle of $E$ by $E^*$.

[6]Note that we use a subscript $i$ for both the chart $U_i$, and for the coordinates $u_i$ and $v_i$ which live on $U_i$ and $U_{i+1}$ respectively.



For a line bundle, the requirement that a section of $(E \otimes \overline{E})^*$ restricts to a Hermitian inner product is equivalent to the restriction that these local sections should be positive definite. We thus see that a Hermitian metric on $E$ is fully specified by a set of positive functions $h_i : \mathbb{C}^n \to \mathbb{R}_{\geq 0}$ such that

$$h_{i+1} = \tau_{i,i+1} \overline{\tau}_{i,i+1} h_i = |u_i|^{2k} h_i, \quad i = 0, 1, \dots, n. \tag{61}$$

An obvious solution to this equation in the patches $\mathcal{U}_0$ and $\mathcal{U}_1$ is given by

$$h_i = \frac{1}{(1 + |u_1|^2 + \dots + |u_n|^2)^k}, \text{ and } h_{i+1} = \frac{1}{(1 + |v_1|^2 + \dots + |v_n|^2)^k}, \tag{62}$$

and similarly in the remaining patches $\mathcal{U}_i$. Note that in the chart $U_i$ the metric acts on sections $s, t \in \Gamma(E)$ as

$$h(s, t) = s \overline{t} h_i. \tag{63}$$

We have thus obtained a hermitian metric on $E$. Since the structure group of a line bundle is one dimensional, the metric connection corresponding to $h$ is simply given by

$$A = \partial \log(h). \tag{64}$$

We can then work explicitly in the patches $U_i$ and $U_{i+1}$ of $\mathbb{P}^n$ to find the following local expressions for $A$

$$A_i = \frac{-k(\overline{u}_1 du_1 + \dots + \overline{u}_n du_n)}{1 + |u_1|^2 + \dots + |u_n|^2}, \text{ and } A_{i+1} = \frac{-k(\overline{v}_1 dv_1 + \dots + \overline{v}_n dv_n)}{1 + |v_1|^2 + \dots + |v_n|^2}. \tag{65}$$

We can then verify that this expression is a local connection form by checking that it satisfies the following transformation property

$$A_{i+1} = \tau_{i,i+1}^{-1} A_i \tau_{i,i+1} + \tau_{i,i+1}^{-1} d\tau_{i,i+1} = A_i + k u_i^{-1} du_i, \text{ for } i = 0, 1, \dots, n. \tag{66}$$

We can also verify that this connection is compatible with the Hermitian bundle metric $h$. To do so, we work locally in the patch $U_i$ and find

$$\begin{aligned}
h(\nabla s, t) + h(s, \nabla t) &= (ds_i \overline{t}_i + s_i d\overline{t}_i) h_i + (A_i + \overline{A}_i) s_i \overline{t}_i h_i \\
&= (ds_i + A_i s_i) \overline{t}_i h_i + s_i (d\overline{t}_i + \overline{A}_i \overline{t}_i) h_i \\
&= d(s_i \overline{t}_i) h_i + s_i \overline{t}_i \left( (A_i + \overline{A}_i) h_i \right) \\
&= dh(s, t),
\end{aligned} \tag{67}$$

where $s, t \in \Gamma(E)$, $\nabla = d + A$ denotes the covariant derivative associated with the connection $A$, and where the last equality is via an explicit computation. We have thus found a Hermitian connection on $E$. Finally, we note that the $(0, 1)$-part of this connection coincides with the obvious Dolbeault operator on $E$

$$\begin{aligned}
\nabla^{(0,1)} s &:= \pi^{(0,1)} \nabla s \\
&= \pi^{(0,1)} (ds_i + A_i s_i) \\
&= \pi^{(0,1)} \left( \left( \partial s_i - \frac{k(\overline{u}_1 du_1 + \dots + \overline{u}_n du_n)}{1 + |u_1|^2 + \dots + |u_n|^2} \right) + \overline{\partial} s_i \right) \\
&= \overline{\partial} s_i \\
&= \overline{\partial}_E s_i,
\end{aligned} \tag{68}$$

where $\pi^{(0,1)}$ projects complex one-forms onto their anti-holomorphic parts. The hermitian connection we have constructed is known as the Chern connection on $E$. It is the unique Hermitian connection on $E$ for which $\nabla^{(0,1)} = \overline{\partial}_E$. We have chosen to utilise this connection for the construction of hybrid B-branes since it is always guaranteed to have a $(1,1)$ curvature form.

Now that we have constructed the Chern connection on $E \to B$, we must pull it back to a connection on the bundle $\pi^*E \to Y$. The standard pullback connection construction gives a covariant derivative on $\pi^*E$ defined by

$$(\pi^*\nabla)_X(\pi^*s) = \pi^*(\nabla_{d\pi[X]}s), \tag{69}$$

where $s : \mathbb{P}^n \to \mathcal{O}(k)$ is a section of $E$, and $X$ is a vector field on $Y$. This expression is given locally in the chart $\mathcal{U}_i$ by

$$(\pi^*\nabla)_X(\pi^*s) = \sigma_u^1\left( \partial s_i - \frac{k(\overline{u}_1 du_1 + \ldots + \overline{u}_n du_n)}{1 + |u_1|^2 + \ldots + |u_n|^2} \right) + \sigma_{\overline{u}}^1 \overline{\partial} s_i \,, \tag{70}$$

where $X = \sigma_u^1 \partial_u + \sigma_{\overline{u}}^1 \partial_{\overline{u}} + \sigma_u^2 \partial_{\varphi_u} + \sigma_{\overline{u}}^2 \partial_{\overline{\varphi}_u} \in TY$. It is then straightforward to verify that the corresponding connection $\pi^*A$ is the Chern connection on $\pi^*E$ by verifying the same properties as for the Chern connection on $E$. We can thus conclude that $\pi^*A$ is guaranteed to have a $(1,1)$ curvature form. We note that in this gauge, $(\pi^*A)_{\phi_u} = (\pi^*A)_{\overline{\phi}_u} = 0$, that is to say that the fibre components of the connection specifying a hybrid B-brane vanish locally up to a gauge transformation. This observation fits with the intuition of a hybrid model as a local product of a geometric base with a flat Landau-Ginzburg model.

Before moving on, we comment briefly on the case where $E$ is a sum of line bundles. Denoting the Chern connection for the line bundle $\pi^*\mathcal{O}(k_i)$ by $\pi^*A^{(k_i)}$, one finds that the Chern connection on $E = \pi^*\mathcal{O}(k_1) \oplus \pi^*\mathcal{O}(k_2) \oplus \ldots \oplus \pi^*\mathcal{O}(k_{2\ell})$ is simply given by

$$\pi^*A^{(E)} = \begin{pmatrix} \pi^*A^{(k_1)} & 0 & \ldots & 0 \\ 0 & \pi^*A^{(k_2)} & \ldots & 0 \\ \vdots & \vdots & \ddots & 0 \\ 0 & 0 & 0 & \pi^*A^{(k_{2\ell})} \end{pmatrix}. \tag{71}$$

Throughout the remainder of this paper it should be assumed that, unless otherwise specified, all connections used to construct hybrid B-branes are Chern connections.

## 3.2 Global matrix factorisations over hybrid geometries

So far, we have constructed holomorphic line bundles and connections with $(1,1)$ curvature forms over hybrid geometries. To complete the construction of hybrid B-branes, we also need to construct matrix factorisations that are globally defined over $B$. As previously discussed, a global matrix factorisation is an odd and holomorphic endomorphism of the Chan-Paton bundle which squares to the hybrid superpotential, $W$. For the case at hand, this means that a global matrix factorisation is a global section

$$Q \in \Gamma\left( \mathrm{End}^{odd}(\pi^*E) \right), \tag{72}$$

such that $Q^2 = W$. We seek to represent such a global section as a collection of local sections which are compatible with the transition functions of this bundle. We think of these local sections as matrix factorisations over the patches of $Y$. Concretely, we take a collection of odd endomorphisms $Q_i\left(u_j; \varphi_u^j\right) \in \mathrm{End}^{odd}(\mathcal{U}_i) \cong \mathrm{End}^{odd}(\mathbb{C}^{n+r})$, which satisfy the gluing conditions

defined by the transition functions of the bundle $\text{End}(\pi^*E)$, and square to $W_i$, the hybrid superpotential on the chart $\mathcal{U}_i$. Concretely, this means that the $Q_i$ must satisfy

$$Q_i = (\pi^*\tau_{ij})Q_j(\pi^*\tau_{ij})^{-1}\,, \tag{73}$$

where $\tau_{ij}$ are the transition functions of the base bundle $E$, as well as the local matrix factorisation condition

$$Q_i^2 = W_i\,. \tag{74}$$

We refer to the $Q_i$ as local matrix factorisations over the chart $\mathcal{U}_i$, and view (73) as a condition for gluing them together into a global matrix factorisation $Q$ over the hybrid geometry $Y$.

We have now reduced the problem of giving a global matrix factorisation of a hybrid superpotential as an endomorphism of some holomorphic vector bundle, to the problem of constructing a number of ordinary matrix factorisations subject to a compatibility condition. This reduction allows us to treat the hybrid model on $Y$ very explicitly as a Landau-Ginzburg model fibred over a geometric base. We have shown that to understand B-branes in this model, we can take local patches of $Y$ where the hybrid model reduces to a Landau-Ginzburg model, construct matrix factorisations, and then glue these factorisations together into a global matrix factorisation defined on the entire fibration. We emphasise that in contrast to the study of Landau-Ginzburg models, the construction of a global matrix factorisation furnishes us with the additional data of a Chan-Paton bundle which is necessary to fully specify a hybrid B-brane.

In sections 3.5, and 3.6, we will treat the Chan-Paton bundle of a hybrid B-brane as an unknown, subject only to the condition that it should be of the form (58), and show that once we fix a system of local matrix factorisations $Q_i$ on a cover of $Y$, compatibility with the gluing condition (73) along with the requirement of holomorphicity, are powerful constraints which fix this bundle uniquely up to an overall normalisation. In this way, we can view the process of globalising a matrix factorisation as determining a set of gauge charges compatible with both the structure of the matrix factorisation, and the geometry itself.

Before moving on, we note that since the Chern connection on $\pi^*E$, as well as the R-symmetry and orbifold transformations of the boundary interaction (32) can be given by commuting diagonal matrices, these actions are invariant under the gluing condition (73). This means that R-symmetry and orbifolding are automatically globally defined operations. This fact will be useful later, when we construct explicit examples of global matrix factorisations.

## 3.3 R-symmetry for local matrix factorisations

In section 2.4 we obtained conditions (50) for the R-gradability of a global matrix factorisation $Q$. We now briefly explain how these conditions descend to the level of local matrix factorisations. To do so, we once again consider the case of a hybrid model defined on $Y = \text{tot}(X \to \mathbb{P}^n)$ with a hybrid superpotential $W$, along with a hybrid B-brane $(E, A, Q)$. Utilising the invariance of $R(\lambda)$ under the gluing condition (73), it is sufficient to work locally in the chart $\mathcal{U}_i$. We consider the case of a local matrix factorisation $Q_u$ such that

$$W_u = Q_u^2 = \sum_{i=1}^m f_{iu}(u_j; x_{j,u})g_{iu}(u_j; x_{j,u})\,, \tag{75}$$

for some functions $f_{iu}$ and $g_{iu}$ on $\mathcal{U}_i$, where $u_j$ are the usual inhomogeneous coordinates on $B = \mathbb{P}^n$, and the $x_{j,u}$ are the fibre coordinates on $Y$. Note that in the case that all $f_{iu}$ and $g_{iu}$ are $1 \times 1$, such a local matrix factorisation has a canonical Clifford realisation. This situation will be discussed in detail in section 3.6. To begin, we note that the good hybrid condition (15)

implies that

$$
\begin{aligned}
0 &= \mathcal{L}_V(W_u) - W_u \\
&= \mathcal{L}_V(Q_u^2) - Q_u^2 \\
&= \sum_{i=1}^m \left( (\mathcal{L}_V f_{iu}) g_{iu} + f_{iu} \mathcal{L}_V g_{iu} - f_{iu} g_{iu} \right) \\
&= \sum_{i=1}^m \left( (\mathcal{L}_V f_{iu} - y_{f_i} f_{iu}) g_{iu} + f_{iu}(\mathcal{L}_V g_{iu} - (1 - y_{f_i}) g_{iu}) \right), \text{ for some } y_{f_i} \in \mathbb{C}.
\end{aligned}
\tag{76}
$$

This equality will hold if our local matrix factorisation is quasi-homogeneous in the sense that

$$
\mathcal{L}_V f_{iu} = y_{f_i} f_{iu}, \text{ and } \mathcal{L}_V g_{iu} = (1 - y_{f_i}) g_{iu}.
\tag{77}
$$

The known form of the vertical Killing vector $V$ will then fix the charges $y_{f_i} \in \mathbb{Q}$. From now on, we will assume that this condition holds unless otherwise stated. Under this assumption, we have the following equivalent condition for the R-gradability of a local matrix factorisation

$$
f_{iu}(R_\lambda \cdot \phi) = \lambda^{2 y_{f_i}} f_{iu}(\phi), \text{ and } g_{iu}(R_\lambda \cdot \phi) = \lambda^{2(1 - y_{f_i})} g_{iu}(\phi).
\tag{78}
$$

### 3.4  Orbifolding for local matrix factorisations

Similarly to the previously discussed case of R-symmetry, we can reduce the condition (52) for the orbifold compatibility of a global matrix factorisation to the level of local matrix factorisations. As in the previous section, we take a brane $(E, A, Q)$ over a hybrid geometry $Y = \text{tot}(X \to \mathbb{P}^1)$ with a hybrid superpotential $W$. Utilising the invariance of $\rho$ under the gluing condition (73), it is again sufficient to work locally on the patch $\mathcal{U}_i$. For convenience, we restrict to the case of an abelian orbifold group $\Gamma \cong \mathbb{Z}_d$. We take a representation, $\rho$ of the orbifold group $\Gamma$ on the local Chan-Paton space $\mathcal{U}_i$. For a local matrix factorisation of the form (75), the orbifold invariance of $W_u$ implies that

$$
\sum_{i=1}^m f_{iu}(\gamma \cdot \phi) g_{iu}(\gamma \cdot \phi) = \sum_{i=1}^m f_{iu} g_{iu}.
\tag{79}
$$

The natural way to ensure that this equality holds is to demand that for some $p_{f_i} \in \mathbb{Z}$

$$
f_{iu}(\gamma \cdot \phi) = \gamma^{p_{f_i}} f_{iu}(\phi), \text{ and } g_{iu}(\gamma \cdot \phi) = \gamma^{d - p_{f_i}} g_{iu}(\phi) = \gamma^{-p_{f_i}} g_{iu}(\phi), \text{ where } \gamma^d = 1.
\tag{80}
$$

This condition is guaranteed to hold for a quasi-homogeneous global matrix factorisation in a model for which the orbifold group is generated by $e^{2\pi i J_0}$, which is relevant for Calabi-Yau hybrids.

### 3.5  Hybrid B-branes on $Y = \text{tot}(\mathcal{O}(-2) \to \mathbb{P}^1)$

We briefly discuss the application of the ideas of section 3.2 to the toy model (20) with superpotential (22). In particular, we show how to build hybrid B-branes which generalise the Landau-Ginzburg matrix factorisations of the form $\varphi^{k+1} = \varphi^n \varphi^{k+1-n}$, where $n = 0, 1, \ldots, k+1$. To do so, we first pull back the holomorphic line bundles $\mathcal{O}(q)$ with $q \in \mathbb{Z}$ on $\mathbb{P}^1$ to holomorphic line bundles, $\pi^* \mathcal{O}(q)$ over $Y$. We then make an ansatz that the Chan-Paton bundle is of rank 2

$$
E = \pi^* \mathcal{O}(q_1) \oplus \pi^* \mathcal{O}(q_2), \quad q_i \in \mathbb{Z}.
\tag{81}
$$

We will see that the process of constructing local matrix factorisations will force relations between the charges $q_1$ and $q_2$, thus constraining the form of $E$. The next ingredient we need is a connection on $E$ with a $(1,1)$ curvature form. As usual, we choose to equip $E$ with the holomorphic structure corresponding to the pullback of the Chern connection on the base bundles

$$A^{(E)} = \pi^* A^{(q_1)} \oplus \pi^* A^{(q_2)}. \tag{82}$$

We now seek to construct global matrix factorisations of the hybrid superpotential $W$. This will then fix the Chan-Paton bundle (81), and complete our construction of hybrid B-branes on $Y$. To construct a family of concrete examples of such factorisations, we begin with local matrix factorisations $Q_u^{(n,I)} \in \text{End}(\mathcal{U}_0) \cong \text{End}(\mathbb{C}^2)$ over the patch $\mathcal{U}_0$ (see appendix A.2) of the form

$$Q_u^{(n,I)} = \begin{pmatrix} 0 & E_u \\ J_u & 0 \end{pmatrix}, \text{ with } E_u = \prod_{i \in I}(u - \omega_i)\varphi_u^n, \text{ and } J_u = \prod_{i \in I^c}(u - \omega_i)\varphi_u^{k+1-n}, \tag{83}$$

where $I$ are the labels of some subset of the set of $2(k+1)$'st roots of $-1$, and $I^c$ is its complement. Recalling that global matrix factorisations are odd, global, holomorphic sections of $\text{End}(E)$, we can apply the transition functions of $E$ to obtain an expression for the hybrid matrix factorisation in $\mathcal{U}_1$. Using the form of the transition functions for a line bundle given in appendix A.2, we thus have

$$
\begin{aligned}
Q_v^{(n,I)}(v; \varphi_v) &= \tau_{vu}^{(E)}(u; \varphi_u) Q_u^{(n,I)}(u; \varphi_u) \tau_{vu}^{(E)}(u; \varphi_u)^{-1} \\
&= \begin{pmatrix} u^{-q_1} & 0 \\ 0 & u^{-q_2} \end{pmatrix} \begin{pmatrix} 0 & \prod_{i \in I}(u - \omega_i)\varphi_u^n \\ \prod_{i \in I^c}(u - \omega_i)\varphi_u^{k+1-n} & 0 \end{pmatrix} \begin{pmatrix} u^{q_1} & 0 \\ 0 & u^{q_2} \end{pmatrix} \\
&= \begin{pmatrix} 0 & u^{q_2-q_1} \prod_{i \in I}(u - \omega_i)\varphi_u^n \\ u^{q_1-q_2} \prod_{i \in I^c}(u - \omega_i)\varphi_u^{k+1-n} & 0 \end{pmatrix} \\
&= \begin{pmatrix} 0 & v^{2n+q_1-q_2-|I|} \prod_{i \in I}(1 - v\omega_i)\varphi_v^n \\ v^{-(2n+q_1-q_2-|I|)} \prod_{i \in I^c}(1 - v\omega_i)\varphi_v^{k+1-n} & 0 \end{pmatrix}.
\end{aligned}
\tag{84}
$$

In the first equality we have applied the transition functions of the bundle $\text{End}(E)$ to transform to the local trivialisation $\mathcal{U}_1$, while in the fourth equality we have applied the coordinate change of the manifold $Y = \text{tot}\left(\mathcal{O}(-2) \to \mathbb{P}^1\right)$ to express the result purely in the coordinates of the chart $\mathcal{U}_1$ of $Y$. We thus see that for $Q_v$ to be holomorphic, the charges $q_1$ and $q_2$ must satisfy the following relation

$$q_2 = q_1 + 2n - |I|, \tag{85}$$

which yields the following form for $Q_v$

$$Q_v^{(n,I)} = \begin{pmatrix} 0 & \prod_{i \in I}(1 - v\omega_i)\varphi_v^n \\ \prod_{i \in I^c}(1 - v\omega_i)\varphi_v^{k+1-n} & 0 \end{pmatrix} =: \begin{pmatrix} 0 & E_v \\ J_v & 0 \end{pmatrix}. \tag{86}$$

We have now fully specified a family of global matrix factorisations $Q^{(n,I)}$, in terms of a family of pairs of local matrix factorisations $(Q_u^{(n,I)}, Q_v^{(n,I)})$. Furthermore, we have found that the Chan-Paton bundle of the corresponding B-branes is constrained to be of the form

$$E = \pi^* \mathcal{O}(q_2 + |I| - 2n) \oplus \pi^* \mathcal{O}(q_2). \tag{87}$$

Next, we analyse when this class of branes is compatible with the R-grading and orbifolding of our hybrid model. To do so, we first invoke the R-symmetry transformation property for hybrid B-branes deduced earlier in (45). Writing $R(\lambda) = \mathrm{diag}(r_1, r_2)$ with $r_1, r_2 \in \mathbb{Q}$, and recalling that $u$ and $\varphi_u$ have (vector), R-charges of $0$ and $\frac{2}{k+1}$ respectively, we find that

$$
\begin{aligned}
R(\lambda)Q_u^{(n,I)}(R_\lambda \cdot \phi)R(\lambda)^{-1} &= R(\lambda)Q_u^{(n,I)}(u, \lambda^{\frac{2}{k+1}}\varphi_u)R(\lambda)^{-1} \\
&= \begin{pmatrix} \lambda^{r_1} & 0 \\ 0 & \lambda^{r_2} \end{pmatrix} \begin{pmatrix} 0 & \lambda^{\frac{2n}{k+1}}E_u \\ \lambda^{\frac{2(k+1-n)}{k+1}}J_u & 0 \end{pmatrix} \begin{pmatrix} \lambda^{-r_1} & 0 \\ 0 & \lambda^{-r_2} \end{pmatrix} \\
&= \lambda \begin{pmatrix} 0 & \lambda^{r_1 - r_2 + \frac{2n}{k+1} - 1}E_u \\ \lambda^{-(r_1 - r_2 + \frac{2n}{k+1} - 1)}J_u & 0 \end{pmatrix} \\
&= Q_u^{(n,I)}(u, \varphi_u) \iff r_2 = r_1 + \frac{2n}{k+1} - 1.
\end{aligned} \tag{88}
$$

We can thus conclude that the $U(1)_V$-action on $E$ is given by

$$
R(\lambda) = \lambda^{r_1} \begin{pmatrix} 1 & 0 \\ 0 & \lambda^{\frac{2n}{k+1} - 1} \end{pmatrix}. \tag{89}
$$

We proceed similarly to obtain the necessary conditions for orbifold invariance. Denoting the generator of our orbifold group by $\gamma = e^{\frac{2\pi i}{k+1}} \in \mathbb{Z}_{k+1}$, we have that $\gamma \cdot (u, \varphi_u) = (u, \gamma\varphi_u)$. Then invoking the transformation property (52) and writing $\rho(\gamma) = \mathrm{diag}(\gamma^{\alpha_1}, \gamma^{\alpha_2})$ for some $\alpha_1, \alpha_2 \in \{0, 1, \ldots, k\}$, we find that

$$
\begin{aligned}
\rho(\gamma)^{-1}Q_u^{(n,I)}(\gamma \cdot \phi)\rho(\gamma) &= \rho(\gamma)^{-1}Q_u^{(n,I)}(u, \gamma\varphi_u)\rho(\gamma) \\
&= \begin{pmatrix} \gamma^{-\alpha_1} & 0 \\ 0 & \gamma^{-\alpha_2} \end{pmatrix} \begin{pmatrix} 0 & \gamma^n E_u \\ \gamma^{k+1-n}J_u & 0 \end{pmatrix} \begin{pmatrix} \gamma^{\alpha_1} & 0 \\ 0 & \gamma^{\alpha_2} \end{pmatrix} \\
&= \begin{pmatrix} 0 & \gamma^{n + \alpha_2 - \alpha_1}E_u \\ \gamma^{-(n + \alpha_2 - \alpha_1)}J_u & 0 \end{pmatrix} \\
&= Q_u^{(n,I)}(u, \varphi_u) \iff \gamma^{\alpha_1} = \gamma^{n + \alpha_2}.
\end{aligned} \tag{90}
$$

Our orbifold representation is thus given by

$$
\rho(\gamma) = \gamma^{\alpha_2} \begin{pmatrix} \gamma^n & 0 \\ 0 & 1 \end{pmatrix}, \tag{91}
$$

and so we get $k+1$ possible representations of the orbifold group on the Chan-Paton bundle specified by our choice of $\alpha_2$. We denote the representations corresponding to these branes by $\rho^{(\alpha_2)}$, where $\alpha_2 = 0, 1, \ldots, k$. The condition for charge integrality in the orbifolded theory is

$$
R(e^{i\pi})\rho^{(\alpha_2)}(\gamma) = e^{2\pi i \left(\frac{n + \alpha_2}{k+1} + \frac{r_1}{2}\right)}\sigma_E, \tag{92}
$$

and so we end up with the following relation between the orbifold and R-charges

$$
\frac{r_1}{2} + \frac{n + \alpha_2}{k+1} \in \mathbb{Z}. \tag{93}
$$

This shows that when specifying a hybrid B-brane in the orbifolded model, the overall normalisation of the R-charge is fixed only up to the freedom to redefine $R(\lambda) \to \lambda^2 R(\lambda)$. This is a feature familiar from the Landau-Ginzburg case [24, 27].

Finally, we verify that the Chern connection on $E$ satisfies (49). To do so, recall that in the gauge used in section 3, we have the following expressions for the Chern connection in $\mathcal{U}_0$

$$
A_u^{(E)} = -\frac{\bar{u}}{1 + |u|^2} \begin{pmatrix} q_1 & 0 \\ 0 & q_2 \end{pmatrix}, \text{ and } A_{\varphi_u}^{(E)} = \begin{pmatrix} 0 & 0 \\ 0 & 0 \end{pmatrix}. \tag{94}
$$

We then see that under an R-symmetry transformation

$$R(\lambda)A_u^{(E)}(R_\lambda \cdot \phi)R(\lambda)^{-1} = -\frac{\lambda^{r_1-r_1}\overline{u}}{1+|u|^2}\begin{pmatrix} 1 & 0 \\ 0 & \lambda^{\frac{2n}{k+1}-1} \end{pmatrix}\begin{pmatrix} q_1 & 0 \\ 0 & q_2 \end{pmatrix}\begin{pmatrix} 1 & 0 \\ 0 & \lambda^{-\frac{2n}{k+1}+1} \end{pmatrix} \tag{95}$$
$$= \lambda^{-2(0)}A_u^{(E)},$$

and somewhat trivially

$$R(\lambda)A_{\varphi_u}^{(E)}(R_\lambda \cdot \phi)R(\lambda)^{-1} = \begin{pmatrix} 1 & 0 \\ 0 & \lambda^{\frac{2n}{k+1}-1} \end{pmatrix}\begin{pmatrix} 0 & 0 \\ 0 & 0 \end{pmatrix}\begin{pmatrix} 1 & 0 \\ 0 & \lambda^{-\frac{2n}{k+1}+1} \end{pmatrix} \tag{96}$$
$$= \begin{pmatrix} 0 & 0 \\ 0 & 0 \end{pmatrix} = \lambda^{-2\frac{1}{4}}A_{\varphi_u}^{(E)}.$$

We thus see that indeed, the Chern connection on $E$ transforms as expected under an R-symmetry transformation.

In summary, we have found a family of R-graded hybrid B-branes in the model (20) given by the data

$$\mathfrak{B}^{(n,I,q_2,\alpha_2)} = \left(\pi^*\mathcal{O}(q_2+|I|-2n) \oplus \pi^*\mathcal{O}(q_2), A^{(E)}, Q^{(n,I)}, \rho^{(\alpha_2)}, R(\lambda)\right). \tag{97}$$

The gluing condition and the condition that the global matrix factorisation $Q^{(n,I)}$ should be holomorphic have completely determined the form of the Chan-Paton bundle up to a choice of overall gauge charge normalisation $q_2$. This data can also be rephrased as a family of twisted complexes

$$\mathfrak{B}^{(n,I,q_2,\alpha_2)}: \quad \pi^*\mathcal{O}(q_2+2n-|I|)_{r_1+\frac{2n}{k+1}-1,\alpha_2} \overset{\longrightarrow}{\underset{\longleftarrow}{}} \pi^*\mathcal{O}(q_2)_{r_1,n+\alpha_2}, ^7 \tag{98}$$

where the arrows in the above diagram represent the global matrix factorisation $Q$. This presentation is the one which we will predominantly use in later sections.

The method by which we have constructed this class of branes is rather ad hoc. In the next section, we will discuss a more systematic method which applies to hybrid B-branes which can be realised locally as matrix factorisations of Clifford type. We will also see that the hybrid B-branes $\mathfrak{B}^{(n,I,q_2,\alpha_2)}$ belong to this class.

## 3.6 Hybrid B-branes of Clifford type

So far, we have obtained a formulation for the data of a hybrid B-brane as a tuple $(E, A, Q, \rho, R)$. We have also discussed a useful class of holomorphic vector bundles $(E, A)$, where $E$ is a sum of pulled back holomorphic line bundles over a hybrid geometry, and $A$ is the corresponding Chern connection. In this section, we will present a systematic way to construct a large class of global matrix factorisations $Q$ for Chan-Paton bundles of this form. To do so, we will take local matrix factorisations of Clifford type defined over the charts of our hybrid geometry, and study how to glue them together to form a global matrix factorisation. We will see that as in the example of section 3.5, this globalisation process determines the form of the Chan-Paton bundle $(E, A)$ uniquely up to an overall normalisation.

Matrix factorisations realised in terms of Clifford algebras are familiar from the study of Landau-Ginzburg models, and form a much studied class of examples in this context. We will see that in the context of hybrid models, globalising local matrix factorisations of this type will lead us to endow the fibres of $E$ with the structure of a Clifford algebra module. Furthermore, we will see that the transition functions of $E$ then become isomorphisms between these Clifford algebra modules. This defines a structure known as a Clifford bundle, which has been studied in the setting of spin geometry, see e.g [28].

---

[7]From now on, unless otherwise specified, all complexes are to be regarded as twisted by the corresponding superpotential.

### 3.6.1 Globalisation of local matrix factorisations of Clifford type

Let $Y = \text{tot}(X \xrightarrow{\pi} B)$ be a hybrid geometry over a base $B = \mathbb{P}^n$, with $X = \bigoplus_{i=1}^{r} \mathcal{O}(n_i)$ a holomorphic vector bundle of rank $r \in \mathbb{Z}_{\geq 0}$, given by a sum of holomorphic line bundles on $B$. We fix a superpotential $W$, and consider a hybrid B-brane $(E, A, Q)$ on $Y$, with $E$ given by a holomorphic line bundle on $Y$ of rank $2^m$ for some $m \in \mathbb{Z}_{\geq 0}$. We assume that $E$ can be realised as a sum of pulled back holomorphic line bundles over $B$

$$E \cong \bigoplus_{k=1}^{2^m} \pi^* \mathcal{O}(k_i), \quad k_i \in \mathbb{Z}. \tag{99}$$

Recall that such a vector bundle can be specified by the local trivialisations $\pi^* \mathcal{U}_i$, and the transition functions $\tau_{i,i+1}$, with $i = 0, 1, \ldots, n$ (see appendix A.2). We introduce local coordinates $(u_1, \ldots, u_n; x_{1u}, \ldots, x_{ru})$, and $(v_1, \ldots, v_n; x_{1v}, \ldots, x_{rv})$ on $Y$ over the charts $\mathcal{U}_i$ and $\mathcal{U}_{i+1}$. The coordinate change between these charts is given in appendix A.2:

$$u_j = v_i^{-1} v_j, \text{ for } j \neq i, \quad u_i = v_i^{-1}, \text{ and } \quad x_{ju} = v_i^{-k_j} x_{jv}. \tag{100}$$

We suppose further that over the chart $\mathcal{U}_i$ of $Y$ we have a local matrix factorisation of the local hybrid superpotential $W_u$ of Clifford type:

$$Q_u = \sum_{i=1}^{m} \left( f_{i,u}(u_j, x_{k,u}) \eta_i + g_{i,u}(u_j, x_{k,u}) \overline{\eta}_i \right), \tag{101}$$

where $f_{i,u}$ and $g_{i,u}$ are holomorphic functions on $\mathcal{U}_i$ which are not identically zero,[8] and where $\eta_1, \ldots, \eta_m, \overline{\eta}_1, \ldots, \overline{\eta}_m$ are the generators of a $2^m$-dimensional complex Clifford algebra specified by the anti-commutation relations

$$\{\eta_i, \overline{\eta}_j\} = \delta_{ij}, \quad \{\eta_i, \eta_j\} = \{\overline{\eta}_i, \overline{\eta}_j\} = 0. \tag{102}$$

We now take the local Chan-Paton space upon which $Q_u$ acts to be a highest weight module for our Clifford algebra. Explicitly, we introduce the following basis

$$M|_{\mathcal{U}_i} = \{|0\rangle, \overline{\eta}_i |0\rangle, \ldots, \overline{\eta}_{i_1} \overline{\eta}_{i_2} \ldots \overline{\eta}_{i_k} |0\rangle, \ldots, \overline{\eta}_1 \overline{\eta}_2 \ldots \overline{\eta}_m |0\rangle\}, \quad i_1 < i_2 < \ldots < i_m, \tag{103}$$

where the vacuum vector $|0\rangle$ is annihilated by all $\eta_i$'s. We now seek to globalise the local matrix factorisation (101). This process will be constrained by both the local Clifford module structure and the required holomorphicity of $Q$, and will uniquely determine the charges $k_i$ of $E$ up to an overall normalisation.

Using the fact that $E$ decomposes into a sum of pulled back line bundles, we write the transition function $\tau_{i+1,i}$ in the basis $M$ as

$$\overline{\eta}_{i_1} \overline{\eta}_{i_2} \ldots \overline{\eta}_{i_k} |0\rangle \mapsto v_i^{I_k + a_0} \overline{\eta}_{i_1} \overline{\eta}_{i_2} \ldots \overline{\eta}_{i_k} |0\rangle, \text{ where } a_0, I_k(\overline{\eta}_{i_1}, \ldots, \overline{\eta}_{i_k}) \in \mathbb{Z}. \tag{104}$$

We will show that the $a_i := I_1(\overline{\eta}_i)$ are constrained (and in many cases, fixed uniquely), by the transformation properties of the coefficient functions appearing in (101) under a change of coordinates, while the remaining $I_k(\overline{\eta}_{i_1}, \ldots, \overline{\eta}_{i_k})$ are then fixed by the Clifford algebra structure. The only freedom left in the transition functions of $E$ will then be a choice of the normalisation factor $a_0$, which, as we will show, can be interpreted as a vacuum gauge charge.

Noting that a change of coordinates (A.12) on $Y$ from $\mathcal{U}_i$ to $\mathcal{U}_{i+1}$ can only introduce singularities in $v_i$, we see that the coefficients of (101) transform as

$$f_{iu} = v_i^{n_{f_i}} f_{iv}, \text{ and } g_{iu} = v_i^{-m_{g_i}} g_{iv}, \text{ for some } n_{f_i}, m_{g_i} \in \mathbb{Z}, \tag{105}$$

---

[8]The case where one or more coefficients are zero will be considered separately in section 3.6.2.

where we choose $f_{iv}$ and $g_{iv}$ uniquely by factoring off all poles and the lowest order zero in $v_i$.[9] Note that this ensures that $f_{iv}$ and $g_{iv}$ are holomorphic and non-zero at $v_i = 0$. Recalling that, with respect to the basis (103), $Q_u$ is simply a matrix, we now examine the transformation properties of certain elements under the transition functions of $E$ and a change of coordinates on $Y$

$$
\begin{aligned}
\langle 0 | \, \eta_i(Q_u) \, | 0 \rangle = g_{iu} &\mapsto v_i^{(a_i+a_0)-(a_0)} v^{-m_{g_i}} g_{iv} = v_i^{a_i - m_{g_i}} g_{iv} \,, \\
\langle 0 | \, (Q_u) \overline{\eta}_i \, | 0 \rangle = f_{iu} &\mapsto v_i^{(a_0)-(a_i+a_0)} v^{n_{f_i}} f_{iv} = v_i^{-a_i + n_{f_i}} f_{iv} \,.
\end{aligned}
\tag{106}
$$

Holomorphicity in the chart $\mathcal{U}_{i+1}$ then implies the inequality

$$
m_{g_i} \le a_i \le n_{f_i} \,.
\tag{107}
$$

For all non-trivial cases, we find that this inequality degenerates to an equality

$$
n_{f_i} = m_{g_i} = a_i \,.
\tag{108}
$$

We will assume that this condition holds for the rest of this section. We have thus determined the charges $a_1, \dots, a_n$. We claim that the Clifford algebra structure ensures that this is enough to determine the remaining parameters $I_k(\overline{\eta}_{i_1}, \dots, \overline{\eta}_{i_k})$ of the the transition functions of E. In particular, we claim that

$$
I_k(\overline{\eta}_{i_1}, \dots, \overline{\eta}_{i_k}) = a_{i_1} + a_{i_2} + \dots + a_{i_k} \,, \text{ for } k = 1, 2, \dots, n \,.
\tag{109}
$$

This equation states that the transition functions of $E$ realise isomorphisms of Clifford algebra modules. We show that this result holds via induction on $k$. The $k = 0$ case is trivial, so we now assume that (109) holds for some $k$. Under the transition functions of $E$, and a change of coordinates on $Y$ we have

$$
\begin{aligned}
\left( \langle 0 | \, \eta_{i_k} \dots \eta_{i_1} \right) Q_u \left( \overline{\eta}_j \overline{\eta}_{i_1} \dots \overline{\eta}_{i_k} \, | 0 \rangle \right) = f_{ju} &\mapsto v^{n_{f_j}} v^{I_k - I_{k+1}} f_{jv} \,, \\
\left( \langle 0 | \, \eta_{i_k} \dots \eta_{i_1} \eta_{i_j} \right) Q_u \left( \overline{\eta}_{i_1} \dots \overline{\eta}_{i_k} \, | 0 \rangle \right) = g_{ju} &\mapsto v^{-m_{g_j}} v^{I_{k+1} - I_k} g_{jv} \,.
\end{aligned}
\tag{110}
$$

The holomorphicity of $Q$ and the fact that $n_{f_i} = m_{g_i} = a_i$ then imply that

$$
I_{k+1}(\overline{\eta}_j, \overline{\eta}_{i_1}, \dots, \overline{\eta}_{i_k}) = I_k(\overline{\eta}_{i_1}, \dots, \overline{\eta}_{i_k}) + a_j \,.
\tag{111}
$$

And so, by our inductive assumption,

$$
I_{k+1}(\overline{\eta}_j, \overline{\eta}_{i_1}, \dots, \overline{\eta}_{i_k}) = a_j + a_{i_1} + \dots + a_{i_k} \,,
\tag{112}
$$

as required. Hence, the Chan-Paton bundle $E$ is completely determined by the structure of the local matrix factorisation $Q_u$, and a choice of vacuum gauge charge $a_0$.

We have now provided a method for globalising a local matrix factorisation with a Clifford algebra realisation over $\mathcal{U}_i$ into a matrix factorisation over $\mathcal{U}_i \cup \mathcal{U}_{i+1}$. Repeating this process, we obtain a family of global matrix factorisations over all of $Y$, parameterised by a choice of vacuum gauge charge $a_0$. Additionally, we have seen that the fibres of the resulting Chan-Paton bundle are Clifford modules and that the transition functions of this bundle are isomorphisms between them. This is exactly the definition of a Clifford bundle.

---

[9]This definition ensures that in all non-trivial cases, $f_{iu}$ and $g_{iu}$ have the same form as $f_{iv}$ and $g_{iv}$.

### 3.6.2 The case of zero-coefficients

In the previous section, we assumed that the coefficient functions appearing in the local matrix factorisation (101) were not identically zero. We briefly discuss what happens when this assumption is dropped. This discussion will be used in sections 4.1 and 4.2 to construct hybrid B-branes which are D0-branes on the base $B$.

Suppose that in the same setup as the previous section, we have a local matrix factorisation of Clifford type with $1 \leq p \leq m$ zero coefficients

$$Q_u = \sum_{i=1}^{m-p} \left( f_{i,u}(u_j, x_{k,u}) \eta_i + g_{i,u}(u_j, x_{k,u}) \overline{\eta}_i \right) + \sum_{i=m-p+1}^{m} f_{m,u}(u_j, x_{k,u}) \eta_i \,. \tag{113}$$

Calculations which closely parallel those of the previous section then show that the gauge charges of the resulting hybrid B-brane are still highly constrained, but no longer entirely fixed. In particular, we have

$$m_{g_i} \leq a_i \leq n_{f_i}, \quad 1 \leq i \leq m-p \,, \tag{114}$$

$$a_i \leq n_{f_i}, \qquad m-p < i \leq m \,. \tag{115}$$

As in the previous section, we find that in all non-trivial cases, the first inequality degenerates to an equality. However, we see that the gauge charges of the Clifford generators corresponding to the zero-coefficients remain as free parameters. In the case of a single zero-coefficient, there is a nice physical interpretation of this extra parameter. This will be discussed in sections 4.1 and 4.2.

An inductive argument mirroring that of the previous section shows that the remaining components of the transition functions of $E$ satisfy a condition somewhat weaker than (109)

$$I_{k+1}(\overline{\eta}_j, \overline{\eta}_{i_1}, \dots, \overline{\eta}_{i_k}) = I_k(\overline{\eta}_{i_1}, \dots, \overline{\eta}_{i_k}) + a_j, \quad 1 \leq j \leq m-p \,, \tag{116}$$

$$I_{k+1}(\overline{\eta}_j, \overline{\eta}_{i_1}, \dots, \overline{\eta}_{i_k}) \leq I_k(\overline{\eta}_{i_1}, \dots, \overline{\eta}_{i_k}) + n_{f_i}, \quad m-p < j \leq m \,. \tag{117}$$

This is not always sufficient to guarantee that the Clifford module structure (109) holds. If there is only a single zero-coefficient (i.e $p = 1$), then after making a choice of $a_m \geq n_{f_m}$ we can always apply (116) to determine the remaining elements of the transition functions of $E$ and recover a Clifford algebra structure. However, if there are multiple zero-coefficients, higher order ambiguities will remain. For example, if $p = 2$, we will be left with undetermined elements $I_2$ which are subject only to the constraints

$$I_2(\overline{\eta}_{m-1}, \overline{\eta}_m) \leq a_{m-1} + n_{f_m}, \text{ and } I_2(\overline{\eta}_{m-1}, \overline{\eta}_m) \leq a_m + n_{f_{m-1}}. \tag{118}$$

In such cases, we can still make specific choices for these undetermined quantities which will force the Clifford algebra structure to hold, although this is no longer enforced by holomorphicity and the gluing condition

Before moving on, we note that the cases for which a single local matrix factorisation $Q_u$ and the requirement of holomorphicity fail to uniquely fix a global matrix factorisation arise when $Q_u$ represents an object which can be localised completely on the corresponding patch of $Y$. This can occur for D0-branes as well as for empty branes, as we will see later. We emphasise that even in these cases, a choice of a system $\{Q_i\}_{i \in I}$ of local matrix factorisations on a cover $\{\mathcal{U}_i\}_{i \in I}$ of $Y$ which are compatible with the gluing condition will always unambiguously specify a global matrix factorisation.

### 3.6.3 Orbi-bundles

The above discussion has focused on hybrid geometries and Chan-Paton bundles realised as direct sums of holomorphic line bundles. However, some of the known hybrid geometries are defined in terms of orbi-bundles, a generalisation of line bundles which allows for a certain class of singularities in the transition functions [6]. Physically, these orbi-bundles arise in connection with hybrid models on gerbes [29], and correspond to a certain type of orbifold structure in the fibre coordinates. The transition functions of the orbi-bundle $\pi : \mathcal{O}\left(\frac{m}{n}\right) \to \mathbb{P}^n$ for coprime integers $m$ and $n$ are simply

$$\tau_{ij} : U_i \cap U_j \to \mathbb{C}^*, \quad [z_0 : z_1 : \ldots : z_n] \mapsto \left(\frac{z_i}{z_j}\right)^{-\frac{m}{n}}. \tag{119}$$

All the considerations and results of the previous sections then go through unchanged, with the exception of considering holomorphicity only up to powers of $\frac{m}{n}$.

### 3.6.4 R-symmetry for hybrid B-branes of Clifford type

The results of section 3.3 readily apply to hybrid B-branes of Clifford type. It is straightforward to show that the R-symmetry transformations are compatible with the Clifford algebra structure. In particular, we find that the R-symmetry action, $R(\lambda)$ is fully specified by a choice of R-charge $r_0$ for the Clifford vacuum $|0\rangle$, along with assigning the Clifford generators $\overline{\eta}_i$ R-charges of $r_i := 2y_{f_i} - 1$. More explicitly, we deduce the following transformation properties for the Clifford generators

$$
\begin{aligned}
R(\lambda)\eta_i R(\lambda)^{-1} &= \lambda^{1-2y_{f_i}}\eta_i := \lambda^{-r_i}\eta_i, \\
R(\lambda)\overline{\eta}_i R(\lambda)^{-1} &= \lambda^{2y_{f_i}-1}\overline{\eta}_i := \lambda^{r_i}\overline{\eta}_i.
\end{aligned}
\tag{120}
$$

These properties ensure that given a local matrix factorisation of Clifford type, it is straightforward to determine its R-grading.

### 3.6.5 Orbifolding hybrid B-branes of Clifford type

Likewise, we can specialise the results of section 3.4 to hybrid B-branes of Clifford type. A simple calculation shows that the orbifold transformations are compatible with the Clifford module structure. We find that $\rho(\gamma)$ can be completely specified by a choice of vacuum orbifold charge $\alpha_0$ to the Clifford vacuum $|0\rangle$, along with assigning the Clifford generators $\overline{\eta}_i$ orbifold charges of $\alpha_i := -p_{f_i} (\text{mod } d)$. This leads to the following transformation properties for the Clifford generators

$$
\begin{aligned}
\rho(\gamma)\eta_i \rho(\gamma)^{-1} &= \gamma^{p_{f_i}}\eta_i := \gamma^{-\alpha_i}\eta_i, \\
\rho(\gamma)\overline{\eta}_i \rho(\gamma)^{-1} &= \gamma^{-p_{f_i}}\overline{\eta}_i := \gamma^{\alpha_i}\overline{\eta}_i,
\end{aligned}
\tag{121}
$$

where the opposite sign when compared to the R-symmetry transformations originates from the opposite transformation properties (45) and (52).

### 3.6.6 Hybrid B-branes of Clifford type on $Y = \text{tot}(\mathcal{O}(-2) \to \mathbb{P}^1)$

We now return to our running example of a hybrid model on $Y = \text{tot}(\mathcal{O}(-2) \to \mathbb{P}^1)$ with superpotential (22). In particular, we will give Clifford realisations of the hybrid B-branes $\mathfrak{B}^{(n,I,\alpha_2)}$ discussed in section (3.5). To do so, we begin with a local matrix factorisation on the patch $\mathcal{U}_0$:

$$Q_u^{(n,I)} = \prod_{i \in I}(u - \omega_i)\varphi_u^n \eta + \prod_{i \in I^c}(u - \omega_i)\varphi_u^{k+1-n}\overline{\eta}. \tag{122}$$

We see that the conditions (108), (78), and (80) are satisfied, so this local matrix factorisation will have a unique globalisation up to choices for the vacuum charges. In particular, we find that the corresponding hybrid B-branes can be given by the following family of (twisted), complexes

$$\pi^*\mathcal{O}(a_0 + 2n - |I|)_{r_0 + \frac{2n}{k+1} - 1, \alpha_0 - n} \overset{\longrightarrow}{\longleftarrow} \pi^*\mathcal{O}(a_0)_{r_0, \alpha_0}\,, \tag{123}$$

where $a_0, r_0, \alpha_0 \in \mathbb{Z}$ are choices of vacuum gauge, R- and orbifold charges respectively, and where the arrows in the above diagram represent the global matrix factorisation $Q$. The first and second terms in the above complex represent the globalisations of the Clifford module states $\overline{\eta}|0\rangle$ and $|0\rangle$, respectively. The vacuum charges $r_0$ and $\alpha_0$ are related by the charge integrality condition

$$\frac{r_0}{2} + \frac{\alpha_0}{k+1} \in \mathbb{Z}\,. \tag{124}$$

We thus see that under the identifications $r_1 = r_0$, $\alpha_0 = n + \alpha_2$, $q_2 = a_0$, these Clifford-type hybrid B-branes correspond to the hybrid B-branes $\mathfrak{B}^{(n,I,q_2,\alpha_2)}$ constructed in section 3.5.

## 3.7 A global description of hybrid B-branes with $B = \mathbb{P}^n$

The description of global matrix factorisations in terms of a collection of local matrix factorisations $Q_i$, glued together by the transition functions of the Chan-Paton bundle is applicable to hybrid models with an arbitrary base. However, for the case of $B$ being projective space, we can unify this description into a single factorisation $Q$ written in terms of projective coordinates which reduces locally to $Q_i$ in the patch $\mathcal{U}_i$. This construction will be used in sections 5.2.3 and 5.3.3 to assist with lifting hybrid B-branes to B-branes in the corresponding GLSM.

Consider a hybrid B-brane $\mathfrak{B} = (E, A, Q, \rho, R)$ on a hybrid geometry $Y$ over a base $\mathbb{P}^n$ with Chan-Paton bundle of the form (99). Assume additionally that this brane has a local Clifford realisation

$$Q_i = \sum_{a=1}^m \left( f_{ai} \eta_a + g_{ai} \overline{\eta}_a \right)\,, \tag{125}$$

over the patch $\mathcal{U}_i$. We assume that this brane satisfies the conditions (108), (50), and (80). We aim to lift the coefficient functions $f_{ai}$ and $g_{ai}$, on the patch $\mathcal{U}_i \cong U_i \times \mathbb{C}^r \cong \mathbb{C}^{n+r}$ to functions $\tilde{f}_a$ and $\tilde{g}_a$ on $Y$, which descend to the $f_{ai}$ and $g_{ai}$ in every coordinate patch.

We wish to introduce homogeneous coordinates $[z_0 : \ldots : z_n]$ on the base $\mathbb{P}^n$, as well as globally defined fibre coordinates $x_k$ with the appropriate transformation properties. To do so, we make use of the realisation of the total space of a sum of line bundles on $\mathbb{P}^n$ as the complement of a certain deleted set $\Delta$ in a weighted projective space (see e.g. [30, section 7.2])

$$Y = \text{tot}\left( \bigoplus_{i=1}^r \mathcal{O}(n_i) \to \mathbb{P}^n \right) \cong \mathbb{P}^{n+r}_{(1,\ldots,1,n_1,\ldots,n_r)} - \Delta\,. \tag{126}$$

We can then utilise the weighted projective coordinates for the base and fibre fields

$$(z_0, \ldots, z_n; x_1, \ldots, x_r) \sim (\lambda z_0, \ldots, \lambda z_n; \lambda^{n_1} x_1, \ldots, \lambda^{n_r} x_r)\,, \text{ with } \lambda \in \mathbb{C}^*\,. \tag{127}$$

We then homogenise the coefficient function $f_{ai}$

$$\tilde{f}_{a,i}(z_0, \ldots, z_n; x_k) := z_i^{k_i} f_{ai}\left( \frac{z_0}{z_i}, \ldots, \frac{z_{i-1}}{z_i}, \frac{z_{i+1}}{z_i}, \ldots, \frac{z_n}{z_i}; x_k \right)\,, \tag{128}$$

for some choice of $k_i \in \mathbb{Z}_{\geq 0}$ such that $\tilde{f}_{ai}$ is non-singular at $z_i = 0$, and where the $x_k$ are fibre coordinates defined on all of $Y$. Since $f_{ai}$ was defined for $z_i \neq 0$, the domain of $\tilde{f}_{ai}$ is now all of $Y$. We can do the same thing in the patch $\mathcal{U}_{i+1}$, and find

$$\tilde{f}_{a,i+1}(z_0, \ldots, z_n; x_k) := z_{i+1}^{k_{i+1}} f_{a,i+1}\left( \frac{z_0}{z_{i+1}}, \ldots, \frac{z_i}{z_{i+1}}, \frac{z_{i+2}}{z_{i+1}}, \ldots, \frac{z_n}{z_{i+1}}; x_k \right)\,, \tag{129}$$

for some $k_{i+1} \in \mathbb{Z}_{\geq 0}$ such that $\tilde{f}_{a,i+1}$ is non-singular at $z_{i+1} = 0$. We now have two a priori different lifts to $Y$. We thus need to show that there is a unique choice of $k_i$ and $k_{i+1}$ such that $\tilde{f}_{ai} = \tilde{f}_{a,i+1}$. Recall from (105) that $f_{a,i+1}(v_1, \ldots, v_n) = v_i^{-n_{f_a}} f_{a,i}(v_1 v_i^{-1}, \ldots, v_n v_i^{-1})$, and so

$$\tilde{f}_{a,i+1} = z_{i+1}^{k_{i+1}+n_{f_a}} z_i^{-n_{f_a}} f_{ai} . \tag{130}$$

Upon matching coefficients, we then see that $k_i = k_{i+1} = -n_{f_a}$. Continuing in this way for the remaining patches yields a unique and well defined lift of $f_{a,i}$ from $\mathcal{U}_i$ to $Y$.

$$\tilde{f}_a = z_i^{-n_{f_a}} f_{ai}\left(\frac{z_0}{z_i}, \ldots, \frac{z_{i-1}}{z_i}, \frac{z_{i+1}}{z_i}, \ldots, \frac{z_n}{z_i}; x_k\right) . \tag{131}$$

Proceeding similarly for the $g_{a,i}$, we thus obtain a global matrix factorisation written in terms of the globally defined weighted projective coordinates

$$Q = \sum_{a=1}^{m} \left(\tilde{f}_a(z_0, \ldots, z_n; x_1, \ldots, x_r)\eta_a + \tilde{g}_a(z_0, \ldots, z_n; x_1, \ldots, x_r)\overline{\eta}_a\right) . \tag{132}$$

It is now clear that $Q$ reduces to $Q_i$ in the local coordinates of any patch $\mathcal{U}_i$. Several examples of this construction will be given in the next section.

### 3.7.1   A global description of hybrid B-branes on $Y = \text{tot}(\mathcal{O}(-2) \to \mathbb{P}^1)$

To illustrate the global description of hybrid B-branes discussed in the previous section, we will once again return to the example of $\text{tot}(\mathcal{O}(-2) \to \mathbb{P}^1)$. Our goal is to give a presentation of the global matrix factorisations $\left(Q_u^{(n,I)}, Q_v^{(n,I)}\right)$ in terms of homogeneous coordinates. We introduce homogeneous coordinates $[z_0 : z_1]$ on the base $\mathbb{P}^1$ such that $u = \frac{z_1}{z_0}$ and $v = \frac{z_0}{z_1}$, as well as a weighted projective coordinate $\varphi$ of weight $-2$. We can then rewrite $Q_u^{(n,I)}$ and $Q_v^{(n,I)}$ in these coordinates as

$$\begin{aligned}
Q_u^{(n,I)} &= \prod_{i\in I}\left(\frac{z_1}{z_0} - \omega_i\right)\varphi^n \eta + \prod_{i\in I^c}\left(\frac{z_1}{z_0} - \omega_i\right)\varphi^{k+1-n}\overline{\eta}, \\
Q_v^{(n,I)} &= \prod_{i\in I}\left(1 - \frac{z_0}{z_1}\omega_i\right)\varphi^n \eta + \prod_{i\in I^c}\left(1 - \frac{z_0}{z_1}\omega_i\right)\varphi^{k+1-n}\overline{\eta}.
\end{aligned} \tag{133}$$

In the notation of the previous section, we thus find that $k_0 = k_1 = |I|$. This results in the following global realisation of $\left(Q_u^{(n,I)}, Q_v^{(n,I)}\right)$ in weighted projective coordinates:

$$Q^{(n,I)} = \prod_{i\in I}(z_1 - z_0\omega_i)\varphi^n \eta + \prod_{i\in I^c}(z_1 - z_0\omega_i)\varphi^{k+1-n}\overline{\eta}. \tag{134}$$

This expression is homogeneous in the base coordinates, and clearly reduces to the above local matrix factorisations in each patch.

## 4   Examples

In this section, we construct examples of explicit hybrid B-branes in two models. The main motivation for our choice of models comes from the fact that they sit in the stringy Kähler moduli space of some well-studied Calabi-Yau manifolds. The first of these is the degree 8 hypersurface in the toric resolution of the weighted projective space $\mathbb{P}^4_{11122}$ [31, 32]. This is a two-parameter Calabi-Yau manifold with Hodge numbers $(h^{11}, h^{21}) = (2, 86)$. We call the

associated hybrid model the octic hybrid. The second is the complete intersection of two cubics in $\mathbb{P}^5$ [33, 34]. This is a one-parameter Calabi-Yau with Hodge numbers $(1, 73)$. We refer to the hybrid model as the bicubic hybrid.

From now on, we will omit the pullback map $\pi^*$ from our notation when discussing line bundles, connections and transition functions over hybrid geometries. Furthermore, when we do not explicitly state the connection or holomorphic structure on the Chan-Paton bundle of a hybrid B-brane, the implication is that we are taking the pullback of the Chern connection on the corresponding base bundle.

## 4.1 The octic hybrid

The octic hybrid is a hybrid model with target space $Y = \text{tot}(\mathcal{O}^{\oplus 3} \oplus \mathcal{O}(-2) \to \mathbb{P}^1)$. This hybrid has been studied extensively in the closed string setting [6, 35–37].

Since $Y$ has a $\mathbb{P}^1$ base, we once again introduce the two coordinate patches $\mathcal{U}_0$ and $\mathcal{U}_1$ as per appendix A.2. We introduce a local base coordinate $u$ in the patch $\mathcal{U}_0$, along with local fibre coordinates $x_{i,u}$, with $i = 3, 4, 5, 6$. Note that $B = \mathbb{P}^1$ could alternatively be equipped with homogeneous coordinates $[z_0 : z_1]$ related to $u$ via $u = \frac{z_1}{z_0}$. This will be necessary in section 5.3.3, but for consistency with the existing literature we will call these homogeneous coordinates $[x_1 : x_2]$.[10] Note also that $x_{6,u}$ is the coordinate of the $\mathcal{O}(-2)$ fibre, while $x_{a,u}$ with $a = 3, 4, 5$ are the coordinates of the trivial $\mathcal{O}(0)$ fibres. The local coordinates on $\mathcal{U}_0$ and $\mathcal{U}_1$ are related as follows

$$u = v^{-1}, \quad x_{6,u} = v^2 x_{6,v}, \quad x_{a,u} = x_{a,u}, \quad a = 3, 4, 5. \tag{135}$$

We choose the hybrid superpotential given locally by

$$\begin{aligned}
W_u &= x_{3,u}^4 + x_{4,u}^4 + x_{5,u}^4 + (1 + u^8) x_{6,u}^4, \\
W_v &= x_{3,v}^4 + x_{4,v}^4 + x_{5,v}^4 + (v^8 + 1) x_{6,v}^4,
\end{aligned} \tag{136}$$

which is clearly holomorphic and obeys the potential condition. This choice of superpotential corresponds to a specific point in the complex structure moduli space which is sufficiently generic and suitable for studying D-branes, see section 5.3 for further comments in the context of GLSMs. The volume elements of $Y$ in $\mathcal{U}_0$ and $\mathcal{U}_1$ are related as follows:

$$du \wedge dx_{3,u} \wedge dx_{4,u} \wedge dx_{5,u} \wedge dx_{6,u} = -dv \wedge dx_{3,v} \wedge dx_{4,v} \wedge dx_{5,v} \wedge dx_{6,v}. \tag{137}$$

This shows that the transition functions of the canonical bundle $K$ of $Y$ are trivial, and so $K$ itself is necessarily trivial. This implies that $c_1(TY) = 0$, and so the axial R-symmetry of the octic hybrid is non-anomalous. Furthermore, the total space $Y$ supports a vertical holomorphic Killing vector

$$V = \sum_{a=3}^{6} q^a x_{a,u} \partial_{x_{a,u}}, \tag{138}$$

where $q^a = \frac{1}{4}$. It is then easy to check that $\mathcal{L}_V W = W$. Therefore, the octic hybrid also has a non-anomalous vector R-symmetry, and is thus a good hybrid. Additionally, we elect to take the standard orbifold generated by $e^{2\pi i J_0}$. This is specified by the following action on the local coordinates

$$\gamma \cdot u = u, \quad \gamma \cdot x_{i,u} = \gamma x_{i,u}, \quad \gamma \in \Gamma \cong \mathbb{Z}_4. \tag{139}$$

---

[10]This is why our labelling of the fibre coordinates starts at $x_{3u}$, rather than $x_{1u}$.

### 4.1.1 Some explicit octic hybrid B-branes

We now construct some explicit B-branes on the octic hybrid model. The examples we consider are motivated by constructions which are well known from the study of B-branes in Landau-Ginzburg models, as well as by geometric intuition where available.

The first class of hybrid B-branes consists of globalisations of the following local hybrid matrix factorisation

$$Q_{1,u} = W_u \eta + \overline{\eta} \,. \tag{140}$$

We note that $Q_{1,u}$ represents a trivial brane over $\mathcal{U}_0$, since its boundary potential $\{Q_{1,u}, \overline{Q}_{1,u}\} = (1 + |W_u|^2)\mathbb{1}$ is positive everywhere [5]. From the form of this local matrix factorisation and the discussion of sections 3.6.4 and 3.6.5, we can immediately read off the R-, and orbifold charges of the $\overline{\eta}_i$s. Furthermore, since $Q_{1,u}$ satisfies (108), we can also read off their gauge charges. In total, we find that $\overline{\eta}$ has gauge, R- and orbifold charges of

$$a = 0\,, \quad y = 1\,, \text{ and } \alpha = 0 \pmod 4\,, \tag{141}$$

respectively. Applying the corresponding transition functions then shows that over $\mathcal{U}_1$, we have that $Q_{1,v} = W_v \eta + \overline{\eta}$. We thus see that the corresponding global matrix factorisation $Q_1$ represents a trivial brane globally. The only remaining freedom when constructing these trivial branes, is then a choice of vacuum gauge R- and orbifold charges $(a_0, r_0, \alpha_0)$. The vacuum R- and orbifold charges are further linked by the requirement of charge integrality (56). The fact that $\rho$ is a representation of $\mathbb{Z}_4$ then limits us to choices of $\alpha_0$ such that $\gamma^{\alpha_0} = 1$, $\gamma \in \mathbb{Z}_4$. This means that without loss of generality, we have four choices of representation labelled by $\alpha_0 \in \{0, 1, 2, 3\}$. The condition of charge integrality then reduces to

$$r_0 \in \frac{\alpha_0}{2} + 2\mathbb{Z}\,. \tag{142}$$

This reflects the usual ambiguity under a shift of the R-symmetry by $R(\lambda) \mapsto \lambda^2 R(\lambda)$ which is observed in both NLSMs and orbifolded Landau-Ginzburg models [5]. We can now write the resulting family of hybrid B-branes as a family of complexes

$$\mathfrak{B}_1^{(a_0, r_0, \alpha_0)} : \qquad \mathcal{O}(a_0)_{r_0+1, \alpha_0} \; \overrightarrow{\longleftarrow} \; \mathcal{O}(a_0)_{r_0, \alpha_0}\,, \tag{143}$$

where the arrows represent the global matrix factorisation $Q_1$, the holomorphic line bundles denote the Chan-Paton bundle equipped with the Chern connection, and the first and second subscripts encode the R- and orbifold charges respectively. Furthermore, the first and second terms in the above complex represent the globalisations of the Clifford module states $\overline{\eta}|0\rangle$ and $|0\rangle$, respectively. This family is parameterised by a choice of vacuum gauge charge $a_0 \in \mathbb{Z}$, along with choice of vacuum R- and orbifold charges which satisfy (142).

The next class of hybrid B-branes we consider are localised on $u = v = 0$, which is not a point in $\mathbb{P}^1$. A brane localised on a deleted set is well-understood in the context of GLSMs where such branes are called empty branes [5], and so we will call such a brane a hybrid empty brane. Such B-branes, which are simultaneously empty and trivial, are a novel feature of hybrid models. They will be discussed in more detail later in sections 5.2.3, and 5.3.3. Any local matrix factorisation imposing the condition $u = v = 0$ should correspond to such a hybrid B-brane, but for concreteness we consider an explicit factorisation of the form

$$Q_{2,u} = \eta_1 + x_{6,u}^4 \overline{\eta}_1 + u\eta_2 + u^7 x_{6,u}^4 \overline{\eta}_2 + \sum_{a=3}^{5} \left( x_{a,u} \eta_a + \frac{1}{4} \frac{\partial W_u}{\partial x_{a,u}} \overline{\eta}_a \right)\,. \tag{144}$$

One can verify that the above local hybrid matrix factorisation satisfies the conditions (78), and (80), and thus immediately read off the following R- and orbifold charges for the Clifford generators

$$r_1 = r_2 = -1, \quad r_a = -\frac{1}{2}, \quad \text{and} \quad \alpha_1 = \alpha_2 = 0 \ (\text{mod } 4), \quad \alpha_a = -1 \ (\text{mod } 4). \tag{145}$$

The condition (108) holds for $\overline{\eta}_2$ and $\overline{\eta}_a$, allowing us to conclude that they have gauge charges $a_2 = -1$ and $a_a = 0$, respectively. However, it does not hold for $\overline{\eta}_1$. Instead, we have only the weaker condition that

$$m_{g_1} = -8 \le a_1 \le 0 = n_{f_1}, \tag{146}$$

leading to eight choices of globalisation. We emphasise that this apparent ambiguity remains because we have only specified a local matrix factorisation on $\mathcal{U}_0$, rather than choosing a system of compatible local matrix factorisations on a covering of $Y$. Once such a system is fixed, the patching procedure is again unique. These eight choices correspond to the eight ways of completing such a global matrix factorisation. Physically, this ambiguity arises because there is nothing preventing us from placing empty branes on the complement of the patch $\mathcal{U}_0$. A similar interpretation holds in all examples for which such an ambiguity arises. Despite this complication, one can still show that $I_2(\overline{\eta}_1, \overline{\eta}_2) = a_1 + a_2$, so for any choice of $a_1$ we will still obtain a Clifford algebra structure. We thus find eight global matrix factorisations which reduce locally to $Q_{2,u}$ over the patch $\mathcal{U}_0$. Over $\mathcal{U}_1$ they are given by

$$Q_{2,v}^{(n)} = v^n \eta_1 + v^{8-n} x_{6,v}^4 \overline{\eta}_1 + \eta_2 + x_{6,v}^4 \overline{\eta}_2 + \sum_{a=3}^{5} (x_{a,v} \eta_a + x_{a,v}^3 \overline{\eta}_a), \tag{147}$$

where $n := -a_1 \in \{0, 1, \dots, 8\}$ corresponds to the choice of Clifford structure. We thus obtain a family of eight global matrix factorisations $Q_2^{(n)} = (Q_{2,u}, Q_{2,v}^{(n)})$.[11] We now have a family $\mathfrak{B}_2^{(a_0, r_0, \alpha_0, n)}$ of hybrid B-branes corresponding to the data $(E, A, Q_2^{(n)}, \rho, R)$

$$
\begin{array}{cc}
\mathcal{O}(a_0 - n - 1)^{\oplus 3}_{r_0 - 3, \alpha_0 - 2} & \mathcal{O}(a_0 - n)_{r_0 - 1, \alpha_0} \\
\oplus & \oplus
\end{array}
$$

$$\mathcal{O}(a_0 - n - 1)_{r_0 - \frac{7}{2}, \alpha_0 - 3} \underset{\longleftarrow}{\overset{\longrightarrow}{}} \mathcal{O}(a_0 - n)_{r_0 - \frac{5}{2}, \alpha_0 - 3} \underset{\longleftarrow}{\overset{\longrightarrow}{}} \cdots \underset{\longleftarrow}{\overset{\longrightarrow}{}} \mathcal{O}(a_0 - 1)_{r_0 - 1, \alpha_0} \underset{\longleftarrow}{\overset{\longrightarrow}{}} \mathcal{O}(a_0)_{r_0, \alpha_0}. \tag{148}$$

$$
\begin{array}{cc}
\oplus & \oplus \\
\mathcal{O}(a_0 - 1)_{r_0 - \frac{5}{2}, \alpha_0 - 3} & \mathcal{O}(a_0)^{\oplus 3}_{r_0 - \frac{1}{2}, \alpha_0 - 1}
\end{array}
$$

This family is parameterised by a choice of vacuum gauge charge $a_0 \in \mathbb{Z}$ along with a choice of vacuum R- and orbifold charges $r_0 \in \mathbb{Q}$ and $\alpha_0 \in \mathbb{Z}_4$, as usual.

Our next class of hybrid B-branes is constructed by globalising a matrix factorisation which, in the Landau-Ginzburg context, is known as the canonical matrix factorisation. Over the chart $\mathcal{U}_i$, the canonical matrix factorisation is formed by splitting off a linear factor of $x_{a,u}$ in each of the fibre coordinates:

$$Q_{3,u} = \sum_{a=3}^{6} \left( x_{a,u} \eta_a + \frac{1}{4} \frac{\partial W_u}{\partial x_{a,u}} \overline{\eta}_a \right). \tag{149}$$

We can then immediately read off the R- and orbifold charges of the Clifford generators:

$$r_6 = r_a = -\frac{1}{2}, \quad \text{and} \quad \alpha_6 = \alpha_a = -1 \ (\text{mod } 4). \tag{150}$$

---

[11]We note that for $n = 1$, the two local matrix factorisations will have the same form over each patch. It is predominantly the hybrid B-branes corresponding to this global matrix factorisation which we will be interested in.

The $\eta$-coefficients of $Q_{3,u}$ are independent of the base variables. In a geometric setting, these coefficients can often be identified with the algebraic equations defining the brane, see also the GLSM constructions of geometric branes in appendix C. This leads us to expect that this matrix factorisation should represent a D2 brane on the base $\mathbb{P}^1$. Moreover, since the fibre components of this factorisation have the structure of a canonical Landau-Ginzburg matrix factorisation, we expect that this brane should correspond to the analytic continuation of the structure sheaf of the Calabi-Yau in the large volume region of the moduli space to the hybrid phase [5]. We will confirm this in section 5.3.

To obtain a complete hybrid B-brane, we must globalise $Q_{3,u}$ by deducing a set of transition functions which are compatible with the required holomorphicity of $Q_3$ and the gluing condition (73). Making the usual assumption that the holomorphic vector bundles appearing in the globalisation of this brane are pullbacks of line bundles on the base, we see that in the notation of section 3.6,

$$
\begin{aligned}
f_{6v} &= x_{6v}, & g_{6v} &= (1+v^8)x_{6v}^3, & (n_{f_6}, m_{f_6}) &= (2,2), \\
f_{av} &= x_{av}, & g_{av} &= x_{av}^3, & (n_{f_a}, m_{f_a}) &= (0,0),
\end{aligned}
\tag{151}
$$

so indeed this matrix factorisation satisfies the condition (108). We can thus conclude immediately that that the gauge charges of the Clifford generators are

$$
a_6 = 2, \quad a_3 = a_4 = a_5 = 0,
\tag{152}
$$

with all other components of the transition functions being fixed by the Clifford algebra structure (109). In particular, we see that the Clifford generator $\overline{\eta}_1$ which corresponds to the $\mathcal{O}(-2)$ fibre of $Y$ is the only generator with a non-trivial transformation under the transition functions of $E$. The corresponding family of global matrix factorisations $\mathfrak{B}_3^{(a_0, r_0, \alpha_0)}$ can then be written as

$$
\mathcal{O}(a_0+2)_{r_0-2, \alpha_0-4}
\underset{\mathcal{O}(a_0)_{r_0-\frac{3}{2}, \alpha_0-3}}{\overset{\mathcal{O}(a_0+2)^{\oplus 3}_{r_0-\frac{3}{2}, \alpha_0-3}}{\underset{\longleftarrow}{\overset{\longrightarrow}{\oplus}}}}
\cdots
\underset{\mathcal{O}(a_0)^{\oplus 3}_{r_0-\frac{1}{2}, \alpha_0-1}}{\overset{\mathcal{O}(a_0+2)_{r_0-\frac{1}{2}, \alpha_0-1}}{\underset{\longleftarrow}{\overset{\longrightarrow}{\oplus}}}}
\mathcal{O}(a_0)_{r_0, \alpha_0}.
\tag{153}
$$

We have thus constructed a family of R-graded hybrid B-branes in the octic hybrid model which globalise the local matrix factorisation (149).

Let us construct yet another family of hybrid B-branes. Once again, we start by globalising a local matrix factorisation inspired by a construction known from the Landau-Ginzburg case. These are a certain type of linear matrix factorisations associated to permutation branes in the CFT [38, 39]. Such matrix factorisations use the fact that one can write

$$
x^d + y^d = \prod_{k=0}^{d-1} (x - e^{-i\pi\frac{2k+1}{d}} y).
\tag{154}
$$

Evidently, this only works for Fermat-type polynomials, so when considering such matrix factorisations we have to consider the superpotential at special but suitably generic points in the complex structure moduli space as in (136). In particular, we take the following permutation-type local matrix factorisation over $\mathcal{U}_0$

$$
Q_{4,u} = (x_{3,u} - e^{-\frac{i\pi}{4}} x_{4,u})\eta_1 + h_u(x_{3,u}, x_{4,u})\overline{\eta}_1 + x_{5,u}\eta_2 + x_{5,u}^3\overline{\eta}_2 + x_{6,u}\eta_3 + (u^8+1)x_{6,u}^3\overline{\eta}_3,
\tag{155}
$$

where $h_u(x_{3,u}, x_{4,u}) = \prod_{k=1}^{3}(x_{3,u} - e^{-i\pi\frac{2k+1}{4}} x_{4,u})$. The structure of the matrix factorisation leads us to interpret this construction as a D2-brane wrapping the base $\mathbb{P}^1$, along with a permutation-type brane in the fibre. This brane satisfies the three conditions (108), (78), and (80). Proceeding similarly to the previous example, we thus find the following family of global matrix factorisations, $\mathfrak{B}_4^{(a_0, r_0, \alpha_0)}$:

$$
\mathcal{O}(a_0+2)_{r_0-\frac{3}{2},\alpha_0-3}
\begin{array}{c}
\mathcal{O}(a_0+2)^{\oplus 2}_{r_0-1,\alpha_0-2} \quad \mathcal{O}(a_0+2)_{r_0-\frac{1}{2},\alpha_0-1} \\
\underset{\longleftarrow}{\longrightarrow} \oplus \underset{\longleftarrow}{\longrightarrow} \oplus \underset{\longleftarrow}{\longrightarrow} \mathcal{O}(a_0)_{r_0,\alpha_0} \\
\mathcal{O}(a_0)_{r_0-1,\alpha_0-2} \quad\quad \mathcal{O}(a_0)^{\oplus 2}_{r_0-\frac{1}{2},\alpha_0-1}
\end{array}
\tag{156}
$$

We move on to discuss three different classes of hybrid B-branes which represent D0-branes in the base. Each of these will turn out to have a different interesting interpretation in the geometric phase of the associated GLSM, see section 5.3.3. The first of these classes of hybrid B-branes is constructed by factoring off a linear term depending on the base coordinates, as well as a linear factor of each of the $\mathcal{O}$-fibre coordinates, $x_{3,u}$, $x_{4,u}$, and $x_{5,u}$:

$$
Q_{5,u} = (u - e^{-\frac{i\pi}{8}})\eta_1 + g_u(u)x_{6,u}^4 \overline{\eta}_1 + \sum_{a=3}^{5}\left(x_{a,u}\eta_a + x_{a,u}^3 \overline{\eta}_a\right),
\tag{157}
$$

where $g_u(u) = \prod_{k=1}^{7}(u - e^{-i\pi\frac{2k+1}{8}})$. The $\eta_1$-coefficient of $Q_{5,u}$ indicates that the matrix factorisation describes a D0-brane in the base, located at $u = e^{-\frac{i\pi}{8}}$. Noting that $Q_{5,u}$ satisfies the conditions (108), (78), and (80), the now familiar procedure then leads us to the following family of hybrid B-branes $\mathfrak{B}_5^{(a_0, r_0, \alpha_0)}$:

$$
\mathcal{O}(a_0-1)_{r_0-\frac{5}{2},\alpha_0-3}
\begin{array}{c}
\mathcal{O}(a_0-1)^{\oplus 3}_{r_0-2,\alpha_0-2} \quad\quad \mathcal{O}(a_0-1)_{r_0-1,\alpha_0} \\
\underset{\longleftarrow}{\longrightarrow} \oplus \underset{\longleftarrow}{\longrightarrow} \cdots \underset{\longleftarrow}{\longrightarrow} \oplus \underset{\longleftarrow}{\longrightarrow} \mathcal{O}(a_0)_{r_0,\alpha_0} \\
\mathcal{O}(a_0)_{r_0-\frac{3}{2},\alpha_0-3} \quad\quad \mathcal{O}(a_0)^{\oplus 3}_{r_0-\frac{1}{2},\alpha_0-1}
\end{array}
\tag{158}
$$

The second class of hybrid B-branes which represent a D0-brane in the base are built upon a non-trivial factorisation of the base coordinates, along with a permutation type factorisation in the fibre

$$
Q_{6,u} = (u - e^{-\frac{i\pi}{8}})\eta + g_u(u)x_{6,u}^4 \overline{\eta} + (x_{3,u} - e^{-\frac{i\pi}{4}}x_{4,u})\eta_3 + h_u(x_{3,u}, x_{4,u})\overline{\eta}_3 + x_{5,u}\eta_4 + x_{5,u}^3 \overline{\eta}_4.
\tag{159}
$$

Noting that this matrix factorisation satisfies the conditions (108), (78), and (80), we obtain the following family of hybrid B-branes, $\mathfrak{B}_6^{(a_0, r_0, \alpha_0)}$:

$$
\mathcal{O}(a_0-1)_{r_0-2,\alpha_0-2}
\begin{array}{c}
\mathcal{O}(a_0-1)^{\oplus 2}_{r_0-\frac{3}{2},\alpha_0-1} \quad \mathcal{O}(a_0-1)_{r_0-1,\alpha_0} \\
\underset{\longleftarrow}{\longrightarrow} \oplus \underset{\longleftarrow}{\longrightarrow} \oplus \underset{\longleftarrow}{\longrightarrow} \mathcal{O}(a_0)_{r_0,\alpha_0} \\
\mathcal{O}(a_0)_{r_0-1,\alpha_0-2} \quad\quad \mathcal{O}(a_0)^{\oplus 2}_{r_0-\frac{1}{2},\alpha_0-1}
\end{array}
\tag{160}
$$

For a third and final construction of a hybrid B-brane which represents a D0-brane on the base, we start with the canonical local matrix factorisation (149) and add a linear term which intersects the base $\mathbb{P}^1$ with a divisor given in projective coordinates by the linear equation

$$
f(x_1, x_2) = \alpha x_1 + \beta x_2, \quad \alpha, \beta \in \mathbb{C}.
\tag{161}
$$

The condition $f = 0$ leads to $u = -\frac{\alpha}{\beta}$, and so we expect a D0-brane at this point in the base. A corresponding local matrix factorisation in $\mathcal{U}_0$ is

$$Q_{7,u} = x_{6,u}\eta_1 + (u^8 + 1)x_{6,u}^3\overline{\eta}_1 + \sum_{a=3}^{5}\left(x_{a,u}\eta_a + x_{a,u}^3\overline{\eta}_a\right) + (\alpha + u\beta)\eta_6. \tag{162}$$

This local matrix factorisation is of the type considered in section 3.6.2 with a single zero-coefficient. The globalisation of this local matrix factorisation will still possess a Clifford structure, but our gauge charges will be somewhat less constrained. In particular, we find the following restrictions on the allowable gauge charges

$$a_1 = 2, \quad a_3 = a_4 = a_5 = 0, \quad a_6 \leq -1. \tag{163}$$

To globalise this local matrix factorisation, we must thus choose both a vacuum gauge charge $a_0$, and the charge $a_6$ of $\eta_6$, resulting in a whole family of possible hybrid B-branes which globalise $Q_{7,u}$. It turns out that there is a nice physical interpretation for this family of branes. To see this, we apply the transition functions for fixed $a_6$ to determine the form of $Q_7$ in the patch $\mathcal{U}_1$

$$Q_{7,v}^{(a_6)} = x_{6,v}\eta_1 + (v^8 + 1)x_{6,v}^3\overline{\eta}_1 + \sum_{a=3}^{5}\left(x_{a,v}\eta_a + x_{a,v}^3\overline{\eta}_a\right) + v^{-a_6-1}(\alpha v + \beta)\eta_6. \tag{164}$$

In this patch we have the canonical matrix factorisation intersected with the term

$$v^n f(v, 1), \text{ where } n = -a_6 - 1 \in \mathbb{Z}_{\geq 0}. \tag{165}$$

This expression has a first order zero at $v = -\frac{\beta}{\alpha}$, which is the same zero observed in the patch $\mathcal{U}_0$. However, it also has an additional $n$'th order zero at $v = 0$. We interpret this configuration as representing a single D0-brane in the base at the point $v = -\frac{\beta}{\alpha}$, as well as a configuration of $n \in \mathbb{Z}_{\geq 0}$ additional D0-branes at the point $v = 0$ in the base. This is actually quite natural, since our local matrix factorisations in $\mathcal{U}_0$ describes all points of $\mathbb{P}^1$ other than $v = 0$. In particular, if we have a D0-brane on $B$ at some point in $\mathcal{U}_0$, we are free to place any configuration of D0 branes at $\{v = 0\} = \mathbb{P}^1 \setminus \mathcal{U}_0$. Somewhat curiously, we observe this phenomenon only for D0 base branes which are constructed by taking a hybrid matrix factorisation of a D2-brane in $B$ and intersecting this with a divisor in the base. For example, the D0 base branes discussed earlier in this section do not share the same ambiguity. For later reference, the resulting family of hybrid B-branes will be referred to as $\mathfrak{B}_7^{(a_0, r_0, \alpha_0, n)}$.

## 4.2 The bicubic hybrid

The bicubic hybrid geometry [19, 29, 36, 37, 40] can then be interpreted as either

$$Y = \text{tot}\left(\mathcal{O}(-1)^{\oplus 6} \to \mathbb{P}_{33}^1\right), \tag{166}$$

where $\mathbb{P}_{33}^1 := (\mathbb{C}^*)^2/(z_0, z_1) \sim (\lambda^3 z_0, \lambda^3 z_1)$ is a $\mathbb{P}^1$ with non-minimal weights, or as

$$Y = \text{tot}\left(\mathcal{O}\left(-\frac{1}{3}\right)^{\oplus 6} \to G_3\mathbb{P}^1\right), \tag{167}$$

where $G_3\mathbb{P}^1$ denotes a $\mathbb{Z}_3$-gerbe on $\mathbb{P}^1$, and $\mathcal{O}\left(-\frac{1}{3}\right)$ is an orbi-bundle as discussed in section 3.6.3 [29]. The hybrid superpotential of this model is given in the patches $\mathcal{U}_0$ and $\mathcal{U}_1$ respectively by

$$\begin{aligned} W_u &= G_{3,u}^1(x_{1,u}\ldots,x_{6,u}) + uG_{3,u}^2(x_{1,u}\ldots,x_{6,u}), \\ W_v &= vG_{3,v}^1(x_{1,v}\ldots,x_{6,v}) + G_{3,v}^2(x_{1,v}\ldots,x_{6,v}), \end{aligned} \tag{168}$$

where $G_3^1$ and $G_3^2$ are cubic polynomials in the fibre coordinates. The second interpretation of the hybrid geometry, (167), will be utilised for now, since working with orbi-bundles over an ordinary $\mathbb{P}^1$ is more natural from the hybrid perspective. We will switch to the other interpretation when we consider branes in the associated GLSM in section 5.2. The corresponding coordinate transformations are

$$u = v^{-1} \text{ , and } x_{i,u} = v^{\frac{1}{3}} x_{i,v} \,. \tag{169}$$

Utilising the above coordinate transformations, one can show that $d\Omega_u = -d\Omega_v$, where $\Omega_u$ is the holomorphic volume form on $\mathcal{U}_0$. Therefore, the canonical bundle of $Y$ is trivial, and so $c_1(T_Y) = 0$. This implies that the axial R-symmetry of the bicubic hybrid is non-anomalous. Next, we note that $Y$ possesses the following vertical holomorphic Killing vector

$$V = \sum_{i=1}^{6} q^i x_{i,u} \partial_{x_{i,u}} \,, \tag{170}$$

with $q^i = \frac{1}{3}$. It is then easy to check that indeed $\mathcal{L}_V W = W$, and so $Y$ has a non-anomalous vector R-symmetry. We conclude that the good hybrid condition is satisfied. Additionally, we take the standard orbifold generated by $e^{2\pi i J_0}$. The orbifold action on the bulk coordinates is

$$\gamma \cdot u = u \,, \quad \gamma \cdot x_{i,u} = \gamma x_{i,u} \,, \quad \gamma \in \Gamma \cong \mathbb{Z}_3 \,. \tag{171}$$

### 4.2.1 Some explicit bicubic hybrid B-branes

Let us construct some explicit B-branes on the bicubic hybrid model. Before we begin, note that since our hybrid geometry is now the total space of an orbi-bundle, we must generalise our notion of holomorphicity. In particular, transition functions and coordinate changes including integer powers of $\frac{1}{3}$ are to be permitted.

Our first class of bicubic hybrid B-branes is obtained by globalising the following local matrix factorisation over the patch $\mathcal{U}_0$

$$Q_{1,u} = G_{3,u}^1 \eta_1 + \overline{\eta}_1 + G_{3,u}^2 \eta_2 + u \overline{\eta}_2 \,. \tag{172}$$

This local matrix factorisation should correspond to a hybrid B-brane localised on $u = v = 0$, and thus should be a hybrid empty brane similar to the type considered previously for the octic (144). One can verify that this local hybrid matrix factorisation satisfies the conditions (78), and (80), and thus immediately read off the following R- and orbifold charges for the Clifford generators

$$r_1 = r_2 = 1 \,, \text{ and } \alpha_1 = \alpha_2 = 0 \,(\mathrm{mod}\,3) \,. \tag{173}$$

The condition (108) holds for $\overline{\eta}_2$, allowing us to conclude that it has gauge charge $a_2 = 1$. However, it does not hold for $\overline{\eta}_1$. Instead, we have only the weaker condition that

$$m_{g_1} = 0 \le a_1 \le 1 = n_{f_1} \,, \text{ with } a_1 \in \frac{1}{3}\mathbb{Z} \,, \tag{174}$$

leading to four choices of globalisation. For convenience, we define $n = 3a_1$ so that these branes correspond to $n = 0, 1, 2, 3$. Despite this complication, one can still show that $I_2(\overline{\eta}_1, \overline{\eta}_2) = a_1 + a_2$, and so for any choice of $a_1$ we will still obtain a Clifford algebra structure. We then obtain the following family of hybrid B-branes

$$\mathfrak{B}_1^{(a_0, r_0, \alpha_0, n)}: \quad \mathcal{O}\left(a_0 + \frac{n+3}{3}\right)_{r_0+2,\alpha_0} \overset{\longrightarrow}{\underset{\longleftarrow}{}} \begin{array}{c} \mathcal{O}\left(a_0 + \frac{n}{3}\right)_{r_0+1,\alpha_0} \\ \oplus \\ \mathcal{O}\left(a_0 + 1\right)_{r_0+1,\alpha_0} \end{array} \overset{\longrightarrow}{\underset{\longleftarrow}{}} \mathcal{O}(a_0)_{r_0,\alpha_0} \,, \tag{175}$$

where $n$ parameterises the four possible globalisations of $Q_{1,u}$. For a given choice of $n$, this family is parameterised by a choice of vacuum gauge charge $a_0 \in \frac{1}{3}\mathbb{Z}$ along with a choice of vacuum R- and orbifold charges $r_0 \in \mathbb{Q}$ and $\alpha_0 \in \mathbb{Z}_3$, as we saw in the case of the octic. The choice of $r_0$ and $\alpha_0$ is further constrained by the charge integrality condition (56), which for the bicubic hybrid reduces to

$$r_0 \in \frac{2\alpha_0}{3} + 2\mathbb{Z} \,. \tag{176}$$

We can also construct hybrid B-branes that correspond to the more familiar notion of trivial branes. To do so, we start with a local matrix factorisation

$$Q_{2,u} = W_u \eta + \overline{\eta} \,. \tag{177}$$

which clearly corresponds to the standard notion of a trivial Landau-Ginzburg B-brane as a trivial matrix factorisation. Globalising this factorisation yields a family of hybrid B-branes of the form

$$\mathfrak{B}_2^{(a_0,r_0,\alpha_0)} : \qquad \mathcal{O}(a_0)_{r_0-1,\alpha_0} \; \underset{\longleftarrow}{\overset{\longrightarrow}{\phantom{xx}}} \; \mathcal{O}(a_0)_{r_0,\alpha_0} \,. \tag{178}$$

As an example of a family of non-trivial hybrid B-branes we globalise the canonical local matrix factorisation over the patch $\mathcal{U}_0$

$$Q_{3,u} = \sum_{i=1}^{6} \left( x_{i,u} \eta_i + \frac{1}{3}\frac{\partial W_u}{\partial x_{i,u}}\overline{\eta}_i \right) . \tag{179}$$

Noting that this local matrix factorisation imposes no restrictions on the base coordinates, we expect it to correspond to a D2 brane on the base $\mathbb{P}^1$. Additionally, since this factorisation is of canonical Landau-Ginzburg type, we expect it to represent the analytic continuation of the structure sheaf on the corresponding Calabi-Yau [5]. One can verify that $Q_{3,u}$ satisfies the properties (108), (78), and (80), resulting in the following gauge, R- and orbifold charges for the Clifford generators

$$a_i = \frac{1}{3}, \quad r_i = -\frac{1}{3}, \text{ and } \alpha_i = -1 \,. \tag{180}$$

The corresponding brane is thus guaranteed to be fully compatible with the Clifford algebra structure, and our only remaining freedom is to pick vacuum charges $a_0$, $r_0$, and $\alpha_0$ such that $R(\lambda)$ and the representation $\rho(\gamma)$ satisfy the charge integrality condition. These considerations leave us with the following family of global matrix factorisations

$$\mathfrak{B}_3^{(a_0,r_0,\alpha_0)} : \quad \mathcal{O}(a_0+2)_{r_0-2,\alpha_0} \; \underset{\longleftarrow}{\overset{\longrightarrow}{\phantom{x}}} \; \mathcal{O}\left(a_0+\tfrac{5}{3}\right)^{\oplus 6}_{r_0-\frac{5}{3},\alpha_0+1} \; \underset{\longleftarrow}{\overset{\longrightarrow}{\phantom{x}}} \; \cdots \; \underset{\longleftarrow}{\overset{\longrightarrow}{\phantom{x}}} \; \mathcal{O}(a_0)_{r_0,\alpha_0} \,. \tag{181}$$

As a final class of hybrid B-branes on the bicubic, we obtain D0 branes in the base via the same construction used to obtain the octic hybrid branes $\mathfrak{B}_3^{(a_0,r_0,\alpha_0)}$. We start with the canonical local matrix factorisation (179), and intersect the brane with a linear divisor in the base given in projective coordinates by the following equation

$$f(p_1,p_2) := \alpha p_1 + \beta p_2, \quad \alpha,\beta \in \mathbb{C} \,. \tag{182}$$

This results in the following local matrix factorisation

$$Q_{4,u} = \sum_{i=1}^{6} \left( x_{i,u} \eta_i + \frac{1}{3}\frac{\partial W_u}{\partial x_{i,u}}\overline{\eta}_i \right) + f(1,u)\overline{\eta}_7 \,. \tag{183}$$

This local matrix factorisation satisfies the conditions (78), and (80), however due to the zero-coefficient of $\eta_7$, the condition (108) only holds for $i = 1, \ldots, 6$. As discussed in section 3.6.2,

this means that in addition to choosing vacuum gauge charges, we must select a charge $a_7$. This leads to the following hybrid gauge charge assignments

$$a_1 = \ldots = a_6 = \frac{1}{3}, \text{ and } a_7 \in \frac{1}{3}\mathbb{Z}_{\geq 3}. \tag{184}$$

The resulting branes share the interpretation of the octic hybrid B-branes $\mathfrak{B}_7^{(a_0, r_0, \alpha_0)}$ in section 4.1.1. In particular, they represent a single D0 brane in the base at $u = -\frac{\alpha}{\beta}$, along with a configuration of $n := 3(a_7 - 1)$ D0 branes in the base stacked at the point $v = 0$. The resulting family of hybrid B-branes $\mathfrak{B}_4^{(a_0, r_0, \alpha_0, n)}$, can be written as

$$
\mathcal{O}\left(a_0 + \frac{n+9}{3}\right)_{r_0-1, \alpha_0-1} \underset{\longleftarrow}{\overset{\mathcal{O}\left(a_0 + \frac{n+8}{3}\right)^{\oplus 6}_{r_0-\frac{2}{3}, \alpha_0}}{\longrightarrow}} \oplus \underset{\mathcal{O}(a_0 + 2)_{r_0-2, \alpha_0-1}}{\overset{\longrightarrow}{\longleftarrow}} \ldots \overset{\longrightarrow}{\underset{\longleftarrow}{\rightleftarrows}} \mathcal{O}(a_0)_{r_0, \alpha_0}. \tag{185}
$$

One can also construct permutation-type matrix factorisations if one restricts the cubic polynomials $G_3^i$ to a specific form. They are interesting in the context of D-brane transport, and will be considered in section 5.2 and appendix C on the respective GLSM.

# 5 A GLSM perspective on hybrid branes

Just like two-dimensional supersymmetric Landau-Ginzburg models or non-linear sigma models with B-branes, hybrid models can be studied on their own and without reference to the fact that they may arise at special loci of some moduli space. If however they do arise at such loci, linking the hybrids with models at other points in the same moduli space can give further information. In the case of Calabi-Yau hybrids, the relevant moduli space is the stringy Kähler moduli space, and hybrid points share the moduli space with more familiar models such non-linear sigma models with Calabi-Yau target spaces. The global picture is established by the GLSM, the FI-theta parameter space of which can be identified with the stringy Kähler moduli space. Different models arise as low-energy configurations, or phases, in different limiting regions of this parameter space. B-brane transport from one phase of a GLSM to another is understood in abelian GLSMs [5]. Hence, one can ask what happens to well-studied geometric B-branes when transported to a hybrid phase. This is facilitated by the hemisphere partition function [13–15] which computes the charges of such a brane, and can thus give support for some claims regarding the properties of certain hybrid branes which we have made in the previous sections.

This is not the first instance in which hybrid branes have been studied from the UV perspective of the GLSM. Hybrid branes arising as low-energy limits of GLSM branes have been discussed in [16–18]. Our main goal here is to make contact between the branes as they arise intrinsically in hybrid models, and their UV description in terms of the GLSM. With the help of the GLSMs of the bicubic and octic hybrids, we furthermore transport a basis of geometric branes to the corresponding hybrid phases and propose a basis of hybrid branes and mirror periods. We use the hemisphere partition function to support our proposal.

## 5.1 GLSM branes

We consider a Calabi-Yau GLSM specified by $(G, V, W, \rho_V, R)$. We will assume that the gauge group $G$ is abelian, but most statements hold more generally. The complex vector space $V$ is the space of the scalar components $\phi_i \in V$ ($i = 1, \ldots, \dim V$) of the chiral superfields. The

fields transform in the representation $\rho_V : G \to SL(V)$. The gauge charges $Q_i^a$ ($a = 1, \ldots, \mathrm{rk}G$) are the weights of $\rho_V$. There is a vector $U(1)$ R-symmetry $R : U(1) \to GL(V)$ with weights $R_i$. In the GLSM, there is an ambiguity in the choice of R-charge assignment. When restricting to a phase, this ambiguity has to be chosen such that it matches with the R-charges of the IR CFT. Unless otherwise specified, we will fix the ambiguity such that the GLSM R-charges align with the phase we refer to.[12] We consider compact models, and so there is a non-zero gauge invariant superpotential $W \in \mathrm{Sym}V^*$ with R-charge 2. The scalar components of the vector multiplet are denoted by $\sigma_a$, and take values in the Lie algebra $\mathfrak{g}_{\mathbb{C}}$ of $G$. The model has FI-theta parameters $t^a = \zeta^a - i\theta^a$, where $\zeta^a$ are the (real) FI-parameters and the theta angles $\theta^a$ are $2\pi$-periodic.

For $\sigma_a = 0$, the classical vacua are determined by the solutions of the D-term and F-term equations

$$\mu(\phi) = \zeta, \qquad dW(\phi) = 0, \tag{186}$$

where $\mu : V \to \mathfrak{g}^*$ is the moment map. The solutions to these equations determine the Higgs branch of the theory. Depending on the value of $\zeta$, the different solutions are the phases of the GLSM. When some or all $\sigma_a \neq 0$, which, upon taking into account quantum effects, is only allowed at codimension one-loci in the FI-theta parameter space, the theory can have Coulomb- and mixed branches. Phases of non-abelian GLSMs can be more complicated.

A GLSM B-brane is defined by $\mathcal{B} = (M, Q, \rho_M, r_*)$, where $M$ is a $\mathrm{Sym}V^*$-module, and $\rho_M$ and $r_*$ are the representations of the gauge group and the R-symmetry group on $M$, respectively. Finally, $Q$ is a matrix factorisation of $W$, i.e. an odd endomorphism on $M$ which is $G$-invariant and has R-charge 1, i.e. $Q$ satisfies

$$Q^2 = W \cdot \mathrm{id}_M, \quad \rho_M(g)^{-1}Q(g\phi)\rho_M(g) = Q(\phi), \quad \lambda^{r_*}Q(\lambda^R\phi)\lambda^{-r_*} = Q(\phi), \tag{187}$$

with $g \in G$ and $\lambda \in \mathbb{C}^*$.[13] As explained in [5], GLSM branes can be written in terms of complexes of Wilson line branes $\mathcal{W}(q_a)_r$, which are the irreducible components of $M$ with gauge charges $q_a$ R-charges and $r$.

### 5.1.1 Hemisphere partition function

Our main tool for analysing the low-energy properties of branes is the hemisphere partition function [13–15]. Its definition is

$$Z_{D^2}(\mathcal{B}) = C \int d^{\mathrm{rk}}\sigma \prod_{\alpha > 0} \langle \alpha, \sigma \rangle \sinh\langle \alpha, \sigma \rangle \prod_{i=1}^{\dim V} \Gamma\left(i\langle Q_i, \sigma \rangle + \frac{R_i}{2}\right) e^{i\langle t, \sigma \rangle} f_{\mathcal{B}}(\sigma), \tag{188}$$

where $C$ is an undetermined normalisation constant, $\alpha > 0$ are the positive roots of $G$ in case $G$ is non-abelian, $\langle \cdot, \cdot \rangle$ is the pairing $\mathfrak{g}_{\mathbb{C}}^* \times \mathfrak{g}_{\mathbb{C}} \to \mathbb{C}$, and the brane factor $f_{\mathcal{B}}(\sigma)$ is given by

$$f_{\mathcal{B}}(\sigma) = \mathrm{Tr}_M e^{i\pi r_*} \rho_M(e^{2\pi\sigma}). \tag{189}$$

Conjecturally, the hemisphere partition function computes the central charge of a D-brane. When evaluated in a given phase, and upon a suitable identification between the FI-theta parameters and complex structure parameters, the result can be expanded in terms of an integral basis of periods of the mirror Calabi-Yau, the coefficients being the charges of a given brane $\mathcal{B}$.

---

[12]In [5] the R-charges are instead chosen to always align with the geometric phase.
[13]When making the connection to the hybrid phase, we sometimes write $R(\lambda)$ instead of $\lambda^{r_*}$.

In [41] it was proposed that the hemisphere partition function, when evaluated in a given phase, has a universal expression in terms of mathematical objects which can be defined in the IR theory. Evidence was provided for geometric and Landau-Ginzburg phases. Based on results for the sphere partition function [36], we claim that this is also true for good hybrid models with orbifold group $\Gamma$ and that, in suitable settings, the exact central charge of a hybrid brane $\mathfrak{B}$ can be expressed as follows

$$Z_{D^2}^{hyb}(\mathcal{B}) = Z^{hyb}(\mathfrak{B}) = \langle \text{ch}(\mathfrak{B}), \widehat{\Gamma}^* I \rangle, \tag{190}$$

where the brane $\mathfrak{B}$ is the low-energy image of the GLSM brane $\mathcal{B}$ under the projection $\pi^{hyb}$ from the GLSM into the hybrid phase

$$\pi^{hyb}(\mathcal{B}) = \mathfrak{B}. \tag{191}$$

On the right-hand side of (190), $\text{ch}(\mathfrak{B})$ is the Chern character of $\mathfrak{B}$,[14] $\widehat{\Gamma}$ is the Gamma class such that $\widehat{\Gamma}\widehat{\Gamma}^* = e^{\frac{c_1(B)}{2}}\text{Td}(B)$, and $I$ is the $I$-function which can be expanded in terms of the "narrow", bulk cohomology of the hybrid model. This is a subset of the $(a, c)$ chiral ring in the IR CFT. The narrow state space of a hybrid has the form $\bigoplus_\delta H^*(B)_\delta$ [37, 40, 43], where $\delta$ labels a subset of the twisted sectors of the Landau-Ginzburg orbifold in the fibre so that the state in the fibre direction is the vacuum. Not all states, not even all those of the $(a, c)$-ring, are necessarily of this form. Any other states which do not have this structure are referred to as "broad". Comparing to the geometric setting, the definition of a narrow state is equivalent to restricting to the cohomology which comes from the ambient space. We note that the picture that comes from the GLSM is incomplete for models which do not only have narrow cohomology. Objects like the Chern character should be definable on the full state space of the theory in the IR, see for instance [24] for Landau-Ginzburg orbifolds and [44] for some recent work addressing the construction of states associated to marginal deformations of the CFT in the GLSM. For the models discussed here, the narrow states account for all $(a, c)$ states in the CFT. In the present context, we obtain conjectural expressions for the Chern characters with respect to the narrow cohomology from the structure of the hemisphere partition function (190). The pairing $\langle \cdot, \cdot \rangle$ in (190) for good hybrids should be taken to be

$$\langle e_{\delta,a}, e_{\delta',b} \rangle = \frac{1}{|\Gamma|} \delta_{\delta,\delta'^{-1}} \int_B a \wedge b, \text{[15]} \tag{192}$$

where the $e_{\delta,a}$ are basis elements of the state space with $\delta, \delta'$ labelling a narrow twisted sector and $a, b \in H^*(B)$.

### 5.1.2 D-brane transport

In order to study the properties of hybrid branes, we find GLSM branes that flow to branes which generate a basis of the RR-charge lattices in the geometric and hybrid phases. We further investigate how these branes are related by analytic continuation. To do so, we make use of the methods of D-brane transport in GLSMs developed in [5]. The key ingredient to transporting branes between one phase and another via a GLSM lift is the grade restriction rule, or band restriction rule in the multi-parameter case, which determines a subcategory in the category of GLSM B-branes which is equivalent to both B-brane categories associated to the respective phases. There is one choice of window category for each homotopy class of paths that avoids the singular loci in the moduli space where Coulomb or mixed branches emerge.

---

[14]Chern characters of global matrix factorisations were defined in [42]. A Chern character for hybrid branes for the bicubic hybrid was also defined in [19].

[15]The formula proposed in [37] was for $\Gamma = \mathbb{Z}_d$. The Landau-Ginzburg case [41] suggests this generalisation.

The windows $w$ are characterised by allowed subsets of brane charges, i.e. weights of $\rho_M$ that correspond to certain irreducible representations of G. For a general GLSM B-brane, M will not only consist of irreducible components that are in the chosen window. Given a phase, such branes can always be replaced by IR equivalent ones which satisfy the grade restriction rule. Using the cone construction, one can bind empty branes associated to the phase via tachyon condensation. These are objects localised on the deleted set where the moment map equation in (186) does not have a solution for a given choice of $\zeta$. As their locus of support is excluded in the given phase, these GLSM branes reduce to "nothing", in the IR. Binding them to a non-grade-restricted brane does not change the IR brane, but one can replace $\mathcal{B}$ by an IR equivalent $\mathcal{B}^w$ that fits the desired window $w$ and hence is well-defined beyond a phase boundary. Note that grade restriction is only necessary when branes are transported beyond a phase boundary. As long as one works locally in a phase, any lift of a brane to the GLSM is fine. In the following, we choose the hybrid phase as our reference point. For the purpose of D-brane transport, this means that we only need to grade restrict the geometric branes which we transport to the hybrid phase.

### 5.1.3 Lifting hybrid branes to GLSM branes

Let us give a prescription for taking hybrid B-branes realised as global matrix factorisations, and lifting them to B-branes in the corresponding GLSM. This will be used later to identify the GLSM branes corresponding to the global matrix factorisations constructed in sections 4.1 and 4.2. For simplicity, we limit our discussion to hybrid B-branes of Clifford type and hybrid models with a $\mathbb{P}^n$ base.

Consider an abelian GLSM specified by the data $(G, V, W, \rho_V, R)$, where $G = U(1)^k$ for some $k \in \mathbb{Z}_{\geq 1}$. We assume that this GLSM has a phase realised as a hybrid model $(Y, W, \rho, R)$, where $\rho$ is a representation of $\Gamma \cong \mathbb{Z}_d$. When such a hybrid phase occurs, some GLSM fields will acquire a VEV. These fields fall into two classes. The first such class consists of those fields which possess a VEV which constrains them to a single point. These fields will be said to be of point-like type. This is the situation familiar from the Landau-Ginzburg case. The second such class of fields are those which have a VEV which constrains them to take values on a submanifold of V. This determines the base $B$ of the hybrid model. This is the situation which is familiar from the study of non-linear sigma models from the GLSM perspective. These fields will be referred to as being of base-like type.

Loosely speaking, upon reducing to a hybrid phase, the point-like GLSM fields lead to the breaking of $U(1)$ factors of G down to discrete subgroups, while the base-like fields become projective coordinates on the base $\mathbb{P}^n$. The remaining GLSM fields will then span the fibres of the hybrid geometry, and will be said to be of fibre-like type.

To lift a global matrix factorisation to a GLSM matrix factorisation, we must first reinstate the GLSM coordinates by lifting the projective coordinates on the base to base-like coordinates on V in such a way that they obtain the correct VEV in the hybrid phase. We must then re-introduce the point-like fields, if there are any in the model at hand. This procedure must be consistent with the symmetry breaking pattern that breaks G to $\Gamma$ in the hybrid phase. This process is not unique. In particular, there is an ambiguity in how one distributes the point-like fields after their re-introduction. This ambiguity corresponds to the ambiguity in defining GLSM gauge charges from hybrid gauge and orbifold charges. This same ambiguity is encountered in the Landau-Ginzburg setting [5], where it can be exploited to lift branes into a specific charge window.

To describe explicitly how to lift a hybrid B-brane $\mathfrak{B}$ to a GLSM B-brane $\mathcal{B}$, suppose our hybrid model descends from a GLSM in which $0 \leq l \leq k$ point-like fields $p_1, \ldots, p_l$ get VEVs $\langle p_1 \rangle, \ldots, \langle p_l \rangle$, leading to spontaneous symmetry breaking. To avoid future confusion, we note that it is possible to have $l = 0$, and in fact this will be the case for the bicubic hybrid.

We describe the base $\mathbb{P}^n$ by homogeneous coordinates $[z_0 : \ldots : z_n]$, and the fibre of $Y$ by the (weighted homogeneous), coordinates $(x_1, \ldots, x_r)$, as per appendix A.2.

To begin with, we take a hybrid B-brane realised as $\mathfrak{B} = (E, A, Q)$, where $Q$ is a global matrix factorisation of the hybrid superpotential, $W$. We assume that $Q$ is given in terms of weighted homogeneous coordinates on $Y$, which can always be achieved via the procedure described in section 3.7.

Before explaining how to lift $Q$ to a GLSM matrix factorisation, we must first define candidate GLSM gauge and R-symmetry actions. It is important to do so at this stage, since this is where the aforementioned ambiguity arises. For a hybrid model with $l$ point-like fields, $l$ of the $U(1)$ factors are determined by the orbifold charges $\alpha_i$ of the Clifford generators $\overline{\eta}_i$, while the remaining $k - l$ factors are determined by the transition functions of the Chan-Paton bundle $E$, which are in turn determined by the boundary gauge charges $a_i$ of the $\overline{\eta}_i$s. In particular, we make a choice of some vacuum gauge and orbifold charges $a_0$ and $\alpha_0$, as well as a choice of lift from the mod $d$ integers $\alpha_i$ to integers $\tilde{\alpha}_i$ with $\tilde{\alpha}_i = \alpha_i \pmod{d}$. We then set

$$\rho_{\mathsf{M}}(g) = \begin{pmatrix} g^{q_1} & & & \\ & g^{q_2} & & \\ & & \ddots & \\ & & & g^{q_{2\ell}} \end{pmatrix}, \tag{193}$$

where $g^{q_j} := g_1^{q_j^1} \ldots g_k^{q_j^k}$ with $1 \leq j \leq n$, and

$$q_j^i = \begin{cases} \tilde{\alpha}_j + \tilde{\alpha}_0, & 1 \leq j \leq l, \\ a_j + a_0, & l < j \leq k. \end{cases} \tag{194}$$

The two choices made in defining the above expression are exactly what allow us to lift a hybrid B-brane into a specific charge window in the corresponding GLSM, and will additionally determine the distribution of point-like fields in our GLSM matrix factorisations. Before moving on, we note that the candidate gauge group action $\rho_{\mathsf{M}}(g)$ contains the data of both the hybrid orbifold charges $\rho(\gamma)$, and the transition functions $\tau_{i,i+1}$.

There is also some freedom in the choice of GLSM R-charges. We make the choice which is compatible with the hybrid phase R-charge assignments which simplifies some calculations compared to [5]. In other words, we simply assign the same R-charges to the constituent Wilson line branes as we did to the constituent line bundles:

$$\mathsf{R}(\lambda) = \lambda^{r_0} \begin{pmatrix} \lambda^{r_1} & & \\ & \ddots & \\ & & \lambda^{r_{2\ell}} \end{pmatrix}, \tag{195}$$

where $r_0$ is the R-charge of the Clifford vacuum and the $r_i$s are the R-charges of the Clifford generators $\overline{\eta}_i$.

With our candidate GLSM gauge and R-symmetry transformations in place, we can now lift $Q$ to a matrix factorisation of the GLSM superpotential, $\mathsf{W}$. To do so, we first lift the homogeneous base and fibre coordinates on $Y$ to affine coordinates on $\mathbb{C}^{n+r}$. We call the resulting endomorphism $Q'$. The second step is to then re-introduce any point-like fields. We do so by following a procedure similar to [5, section 10.2]. For clarity, we limit ourselves to the case of a single point-like field, although the generalisation to $l > 1$ is straightforward. To carry out this process explicitly, we apply the candidate GLSM gauge transformation $\rho_{\mathsf{M}}$ to $Q'$ with parameter $z \in \mathbb{C}$, then set $z^d = p$ to obtain a candidate $\mathsf{Q}$. Explicitly, we have

$$\rho_{\mathsf{M}}(z)^{-1} Q'(z \cdot \phi) \rho_{\mathsf{M}}(z)|_{z^d = p} =: Q'_0(\phi) + p Q'_1(\phi) + \ldots =: \mathsf{Q}(\phi, p), \tag{196}$$

where $\phi = (z_i; x_j)$ collectively represents the lift of the base-like and fibre-like coordinates to $V$, and where $Q'_k$ is defined to be the component of $Q'$ which transforms with weight $dk$ under a transformation by $\rho(z)$, i.e

$$\rho_M(z)^{-1}Q'_k(z \cdot \phi)\rho_M(z)|_{z^d=p} = p^k Q'_k(\phi). \tag{197}$$

Squaring our candidate Q, we then obtain the following equalities

$$\begin{aligned}
Q(z_i; x_j; p)^2 &= (Q'_0)^2 + p\{Q'_0, Q'_1\} + p^2\left((Q'_1)^2 + \{Q_0, Q'_2\}\right) + \dots, \\
Q'(z_i; x_j)^2 &= (Q'_0)^2 + \left(\{Q'_0, Q'_1\}\right) + \left((Q'_1)^2 + \{Q_0, Q'_2\}\right) + \dots = W(\phi).
\end{aligned} \tag{198}$$

By equating terms with the same weight under a transformation of the type (197), we then conclude that

$$Q(z_i; x_j, p)^2 = pW(z_i; x_j) = W(z_i; x_j, p), \tag{199}$$

as required. We have now constructed a GLSM matrix factorisation from our hybrid data. We must verify that Q, $\rho_M$, and R indeed define a GLSM B-brane. From the orbifold invariance of $Q$, and the fact that $Q$ is a lift of a system local matrix factorisations $\{Q_i\}_{i \in I}$ satisfying $Q_i = \tau_{ij}Q_j\tau_{ij}^{-1}$, we see that Q is gauge invariant

$$\rho_M(g)^{-1}Q(z_i; x_j, p)\rho_M(g) = Q(z_i; x_j, p). \tag{200}$$

Similarly, using the fact that the GLSM R-charge assignment has been chosen to be the same as the hybrid R-charge assignment, we have that

$$R(\lambda)Q(R_\lambda \cdot (z_i; x_j, p))R(\lambda)^{-1} = \lambda Q(z_i; x_j, p). \tag{201}$$

We have thus fully constructed a B-brane in the corresponding GLSM.

Let us also briefly discuss the inverse construction, namely the reduction of a GLSM B-brane to a hybrid B-brane. This follows in a straightforward manner from the above discussion. Once again, we restrict to the case $l = 1$ for simplicity. Given a GLSM B-brane $(M, Q, \rho_M, R)$, one first sets the point-like field $p$ to its VEV $\langle p \rangle$, which we choose to be 1. Then using that $p$ has R-charge 0 in our convention, we set

$$\begin{aligned}
Q(z_i; x_j) &= Q(z_i; x_j; 1), \\
R(\lambda) &= R(\lambda).
\end{aligned} \tag{202}$$

We obtain the hybrid orbifold action $\rho(\gamma)$ and the transition functions of $E$ from $\rho_M(g)$. In particular, using that we can always write $\rho_M(g)$ in the form (193), we set

$$\rho(\gamma) = \gamma^{\alpha_0}\begin{pmatrix} \gamma^{\alpha_1} & & & \\ & \gamma^{\alpha_2} & & \\ & & \ddots & \\ & & & \gamma^{\alpha_{2\ell}} \end{pmatrix}, \quad \text{with } \gamma^d = 1, \tag{203}$$

and similarly for the transition functions

$$\tau_{i,i+1} = v_i^{a_0}\begin{pmatrix} v_i^{a_1} & & & \\ & v_i^{a_2} & & \\ & & \ddots & \\ & & & v_i^{a_{2\ell}} \end{pmatrix}. \tag{204}$$

We then set $E$ to be the holomorphic vector bundle on $Y$ which is determined by the transition functions $\tau_{i,i+1}$, and is endowed with the Chern connection. It is then straightforward to verify that the data $(E, A, Q, \rho, R)$ indeed defines a hybrid B-brane on $Y$.

## 5.2 The bicubic GLSM

In this section we discuss the one-parameter GLSM associated to the Calabi-Yau hybrid from section 4.2. This GLSM has recently been discussed in [36,37] to which we refer for details. Compared to section 4, we reverse the order of the examples and discuss the cubic hybrid first since, from the GLSM perspective, the one-parameter model is simpler. We give a brief summary of the GLSM phases and the structure of the moduli space, and state the grade restriction rule following [5]. We evaluate the hemisphere partition function in the geometric and hybrid phases. We also construct bases of branes in the hybrid and geometric phases. In the hybrid phase, this provides a UV description of the branes constructed in section 4.2. We grade restrict the geometric branes and find their analytic continuation to the hybrid phase. Branes in this model have also been discussed in the context of open string mirror symmetry [45] and the mathematical formulation of the categorical Landau-Ginzburg/Calabi-Yau correspondence [19].

### 5.2.1 GLSM and phases

The bicubic GLSM has gauge group $G = U(1)$, and the following matter content:

$$
\begin{array}{c|cc|c}
 & p_1, p_2 & x_1, \ldots, x_6 & \text{FI} \\
\hline
U(1) & -3 & 1 & \zeta \\
U(1)_V & 2 - 6\kappa & 2\kappa &
\end{array}
\tag{205}
$$

with $0 \leq \kappa \leq \frac{1}{3}$. The GLSM superpotential is $W = p_1 G_3^1(x_1, \ldots, x_6) + p_2 G_3^2(x_1, \ldots, x_6)$, where $G_3^1, G_3^2$ are suitably generic homogeneous polynomials of degree 3, as discussed in section 4.2. As for the octic hybrid, we discuss a special choice of the two cubic potentials that will be useful in the context of constructing permutation-type matrix factorisations. One choice for $G_3^1, G_3^2$ which has a high amount of symmetry while still being sufficiently generic is to take the equations describing the mirror of $\mathbb{P}^5[3,3]$:

$$
\begin{aligned}
G_3^1 &= x_1^3 + x_2^3 + x_3^3 - 3\alpha x_4 x_5 x_6, \\
G_3^2 &= x_4^3 + x_5^3 + x_6^3 - 3\alpha x_1 x_2 x_3.
\end{aligned}
\tag{206}
$$

Note that, in contrast to hypersurfaces such as that of the quintic, the Calabi-Yau becomes singular at the "Fermat point" $\alpha = 0$ (see eg. [34] for a detailed discussion). Thus, $\alpha = 0$ is not a valid choice for a generic superpotential, and we will not take into account matrix factorisations that exist only for $\alpha = 0$. This is also reflected in the hybrid phase. There, we must ensure that our choice of potential satisfies the potential condition, (3). A direct calculation shows that this is the case if and only if $\alpha \neq 0$, and $\alpha^6 \neq 1$, where $\alpha^6 = 1$ is the conifold point. The potential condition thus exactly reproduces the smoothness criterion of the corresponding complete intersection.

The $\zeta \gg 0$-phase is the codimension two complete intersection $\mathbb{P}^5[3,3]$ of two cubics $\{G_3^1 = 0, G_3^2 = 0\}$ in $\mathbb{P}^5$. The $\zeta \ll 0$-phase is a hybrid model on $Y = \text{Tot}(\mathcal{O}(-1)^{\oplus 6} \to \mathbb{P}^1_{33})$. See [29,37] for further details. Compared to section 4, where we used orbibundles, this formulation of $Y$ is more natural from the GLSM perspective. Note also that we have chosen the GLSM gauge charges such that the $p$-fields have negative charge, which is the canonical choice for the geometric phase. However, in view of the hybrid phase, it would make more sense to flip the signs of the charges and the FI-parameter. We will implement this flip by hand when we discuss the connection between GLSM and hybrid branes.

The bicubic GLSM has a Coulomb branch at

$$
z = (3\alpha)^{-6} = e^{-t} = \frac{1}{3^6} \qquad \longleftrightarrow \qquad \zeta = \log(3^6) \simeq 6.59, \quad \theta = 0 \mod 2\pi.
\tag{207}
$$

This GLSM falls within the class of models for which D-brane transport is well-understood. The grade restriction rule is [5, p.139]

$$-3 < \frac{\theta}{2\pi} + q_i < 3 \,. \tag{208}$$

The brane charges $q_i$ are the weights of $\rho_{\mathsf{M}}$: since $\mathsf{G} = U(1)$, $\mathsf{M}$ can be represented by $\mathsf{M} = \bigoplus_i \mathcal{W}(q_i)_{r_i}$. Since there is a singularity in the moduli space at $\theta = 0 (\mathrm{mod}\, 2\pi)$, inequivalent paths in the moduli spaces are homotopic to straight lines in the FI-theta parameter space with constant $\theta \in (n, 2\pi + n)$ for $n \in \mathbb{Z}$. For example, for $\theta \in (-6\pi, -4\pi)$, one finds the window

$$w: \qquad q \in \{0, 1, 2, 3, 4, 5\} \,. \tag{209}$$

A shift by $\theta = 2\pi n$ shifts the allowed charges by $-n$. Different choices of window differ by monodromies around the singular point at the phase boundary. We will push all our branes through the window (209), see [19] for a more general treatment.

### 5.2.2 Hemisphere partition function

The GLSM hemisphere partition function for this model is

$$Z_{D^2}(\mathcal{B}) = C \int_{\mathbb{R}} d\sigma \, \Gamma(1 - 3\kappa - 3i\sigma)^2 \Gamma(\kappa + i\sigma)^6 e^{it\sigma} f_{\mathcal{B}}(\sigma) \,. \tag{210}$$

We evaluate this object in both phases, confirming conjectures about its universal structure [41]. The calculation for the geometric phase is completely analogous to the case of the quintic [15]. It can be found in appendix B, along with the construction of a set of GLSM branes corresponding to a basis of geometric branes.

To evaluate the hemisphere partition function in the hybrid phase, we close the integration contour such that the poles of $\Gamma(1 - 3\kappa - 3i\sigma)$ are enclosed:

$$3i\sigma = 1 + k - \varepsilon - 3\kappa, \qquad k \in \mathbb{Z}_{\geq 0} \,. \tag{211}$$

Furthermore, we choose $k = 3n + \delta - 1$ with $\delta = 1, 2$, which accounts for the narrow sectors [37]. Using standard manipulations, and taking into account that the orientation of the contour is clockwise in this case, the hemisphere partition function in this phase can be written as

$$Z_{D^2}^{\zeta \ll 0} = \frac{C}{3i} \sum_{\delta=1}^{2} \sum_{n \geq 0} \oint d\varepsilon \, \frac{\pi^2 (-1)^{6n + 2\delta - 2}}{\sin^2 \pi \varepsilon} \frac{\Gamma\left(n + \frac{\delta}{3} - \frac{\varepsilon}{3}\right)^6}{\Gamma(3n + \delta - \varepsilon)^2} e^{t\left(n + \frac{\delta}{3} - \frac{\varepsilon}{3} - \kappa\right)} f_{\mathcal{B}}\left(-\frac{i}{3}(3n + \delta - \varepsilon - 3\kappa)\right) \,. \tag{212}$$

We define $\varepsilon = -\frac{H}{2\pi i}$, where $H$ is now the hyperplane class of the base $\mathbb{P}^1$. This gives

$$Z_{D^2}^{\zeta \ll 0} = -\frac{2\pi C}{3} \sum_{\delta=1}^{2} \sum_{n \geq 0} \oint dH \frac{e^{-H}}{(1 - e^{-H})^2} \frac{\Gamma\left(n + \frac{\delta}{3} + \frac{H}{6\pi i}\right)^6}{\Gamma\left(3n + \delta + \frac{H}{2\pi i}\right)^2}$$
$$\times e^{t\left(n + \frac{\delta}{3} + \frac{H}{6\pi i} - \kappa\right)} f_{\mathcal{B}}\left(-\frac{i}{3}\left(3n + \delta + \frac{H}{2\pi i} - 3\kappa\right)\right) \,. \tag{213}$$

Now we use that

$$\mathrm{Td}(\mathbb{P}^1) = \frac{H^2}{(1 - e^{-H})^2}, \qquad \int_{\mathbb{P}^1} g(H) = \oint dH \frac{1}{H^2} g(H), \tag{214}$$

to obtain

$$Z_{D^2}^{\zeta \ll 0} = -\frac{2\pi C}{3} \sum_{\delta=1}^{2} \sum_{n \geq 0} \int_{\mathbb{P}^1} \mathrm{Td}(\mathbb{P}^1) e^{-H} \frac{\Gamma\left(n + \frac{\delta}{3} + \frac{H}{6\pi i}\right)^6}{\Gamma\left(3n + \delta + \frac{H}{2\pi i}\right)^2} e^{t\left(n + \frac{\delta}{3} + \frac{H}{6\pi i} - \kappa\right)} f_{\mathcal{B}}. \tag{215}$$

Now, recall the *I*-function for the hybrid model [36, 40]

$$I_{\delta}^{\zeta \ll 0}(H) = \frac{\Gamma\left(1 + \frac{H}{2\pi i}\right)^2}{\Gamma\left(\frac{H}{6\pi i} + \frac{\delta}{3}\right)^6} \sum_{n=0}^{\infty} \frac{\Gamma\left(n + \frac{H}{6\pi i} + \frac{\delta}{3}\right)^6}{\Gamma\left(3n + \delta + \frac{H}{2\pi i}\right)^2} u^{\left(\frac{H}{6\pi i} + n + \frac{\delta}{3} - \frac{1}{3}\right)}, \tag{216}$$

and the Gamma class of $\mathbb{P}^1$

$$\widehat{\Gamma}_{\mathbb{P}^1} = \Gamma\left(1 + \frac{H}{2\pi i}\right)^2, \tag{217}$$

as well as the Gamma class

$$\widehat{\Gamma}_{\delta}(H) = \Gamma\left(1 - \frac{H}{2\pi i}\right)^2 \Gamma\left(\frac{H}{6\pi i} + \frac{\delta}{3}\right)^6. \tag{218}$$

Since $\mathbb{P}^1$ is not Calabi-Yau, the relation between the Todd class and the Gamma class is

$$\mathrm{Td}(\mathbb{P}^1) = e^{\frac{c_1(\mathbb{P}^1)}{2}} \widehat{\Gamma}_{\mathbb{P}^1} \widehat{\Gamma}_{\mathbb{P}^1}^* = e^H \Gamma\left(1 + \frac{H}{2\pi i}\right)^2 \Gamma\left(1 - \frac{H}{2\pi i}\right)^2. \tag{219}$$

This cancels the $e^{-H}$ that appears in the hemisphere partition function, and we finally get, after setting $\kappa = \frac{1}{3}$ and $u = e^{\frac{t}{3}}$

$$Z_{D^2}^{\zeta \ll 0} = -\frac{2\pi C}{3} \sum_{\delta=1}^{2} \int_{\mathbb{P}^1} \widehat{\Gamma}_{\delta}(H) I_{\delta}(H) f_{\mathcal{B}}. \tag{220}$$

Setting $C = -(\frac{1}{2\pi})$, this is consistent with (190). This suggests a definition for the Chern character of a hybrid brane:

$$\mathrm{ch}_{\delta}(\mathfrak{B}) = f_{\mathcal{B}}\left(-\frac{i}{3}\left(3n + \delta + \frac{H}{2\pi i} - 3\kappa\right)\right), \quad \kappa = \frac{1}{3}. \tag{221}$$

For a single Wilson line brane $\mathcal{W}(q_i)_{\alpha + \beta(2\kappa)}$ ($\alpha, \beta \in \mathbb{Z}$) we get the contribution

$$f_{\mathcal{W}(q_i)_{\alpha + \beta(2\kappa)}}\left(-\frac{i}{3}\left(1 + 3n - \delta - 1 - 3\kappa + \frac{H}{2\pi i}\right)\right) = (-1)^{\alpha} \gamma^{\beta} \gamma^{-q_i(\delta - 1)} e^{-q_i \frac{H}{3}}. \tag{222}$$

Any shift in $q_i$ not only shifts the $\mathbb{Z}_3$-orbifold element $\gamma$, but also the base-dependent factor $e^{-\frac{H}{3}}$. This destroys a shift periodicity with respect to the orbifold action that is known from Landau-Ginzburg orbifolds. We will come back to this point in section 5.2.3.

To evaluate the partition function in a given phase, we have to expand all the components of the objects in the hemisphere partition function in terms of $H$. Schematically, this looks as follows

$$\begin{aligned}
f_{\mathcal{B},\delta} &= f_{\mathcal{B},\delta}^{\mathbf{1}} + f_{\mathcal{B},\delta}^{H} H + \mathcal{O}(H^2), \\
\widehat{\Gamma}_{\delta} &= \widehat{\Gamma}_{\delta}^{\mathbf{1}} + \widehat{\Gamma}_{\delta}^{H} H + \mathcal{O}(H^2), \\
I_{\delta} &= I_{\delta}^{\mathbf{1}} + I_{\delta}^{H} H + \mathcal{O}(H^2).
\end{aligned} \tag{223}$$

The integrand of the hemisphere partition function can be expanded as

$$f_{\mathcal{B},\delta}^{\mathbf{1}} \widehat{\Gamma}_{\delta}^{\mathbf{1}} I_{\delta}^{\mathbf{1}} + \left(f_{\mathcal{B},\delta}^{H} \widehat{\Gamma}_{\delta}^{\mathbf{1}} I_{\delta}^{\mathbf{1}} + f_{\mathcal{B},\delta}^{\mathbf{1}} \widehat{\Gamma}_{\delta}^{H} I_{\delta}^{\mathbf{1}} + f_{\mathcal{B},\delta}^{\mathbf{1}} \widehat{\Gamma}_{\delta}^{\mathbf{1}} I_{\delta}^{H}\right) H + \mathcal{O}(H^2). \tag{224}$$

Since $B = \mathbb{P}^1$, the $\mathcal{O}(H)$-term encodes the central charge of the brane, and the structure is consistent with the inner product in (190). The overall factor $\frac{1}{3}$ is consistent with the definition (192) of the pairing on the state space. For every $\delta$, the components of the brane factors enter as follows:

$$
\begin{aligned}
& f^H_{\mathcal{B},\delta} \cdot \widehat{\Gamma}^{\mathbf{1}}_\delta I^{\mathbf{1}}_\delta , \\
& f^{\mathbf{1}}_{\mathcal{B},\delta} \cdot \left( \widehat{\Gamma}^H_\delta I^{\mathbf{1}}_\delta + \widehat{\Gamma}^{\mathbf{1}}_\delta I^H_\delta \right) .
\end{aligned}
\tag{225}
$$

The terms multiplying the brane factors are compatible with $\widehat{\Gamma}^* I$.[16] We can thus interpret the brane factors $f^a_\delta$ as the charges of certain types of branes. In [37], the components of the $I$-function for this model were related to certain states in the hybrid CFT. In particular, $I^{\mathbf{1}}_1 = 1 + \ldots$, which is the power series solution which starts with 1, is associated with an element of the state space that has R-charges $(q_L, q_R) = (0,0)$, so it is to be understood as the hybrid version of the fundamental period of the mirror. Computing the central charge in a geometric phase, the coefficient multiplying the fundamental period is the charge of a D0-brane. This suggests that the coefficient $f^H_{\mathcal{B},1}$ multiplying the hybrid analogue should be interpreted as the Chern character of an object with a D0-brane component on the base, $B$. The twisted sector label encodes information on the properties of the Landau-Ginzburg fibre. This suggests that branes with non-zero $f^H_{\mathcal{B},\delta}$ can be interpreted as D0-branes in the base while those with $f^{\mathbf{1}}_{\mathcal{B},\delta}$ have D2-brane components in the base. We will see in the examples that this is consistent with the expectation from the constructions of hybrid branes. The structure also implies that the integral basis of periods is linked to the basis given by the $I$-function as follows:

$$
\begin{aligned}
& \Pi^{\mathbf{1}}_\delta = \widehat{\Gamma}^{\mathbf{1}}_\delta I^{\mathbf{1}}_\delta , \\
& \Pi^H_\delta = \left( \widehat{\Gamma}^H_\delta I^{\mathbf{1}}_\delta + \widehat{\Gamma}^{\mathbf{1}}_\delta I^H_\delta \right) .
\end{aligned}
\tag{226}
$$

### 5.2.3 Hybrid branes

The aim of this section is to construct GLSM branes which correspond to the examples of hybrid branes we have constructed in section 4.2.1, and to show that the UV and IR branes are related via the lifts/projections discussed in section 5.1.3. Since we will not need to transport the resulting IR branes, we will not pay much attention to lifting the IR branes into specific windows.

Before constructing the actual branes, we note that the empty branes in the hybrid phase are characterised by the following matrix factorisation

$$
\mathsf{Q}_1 = \sum_{i=1}^{2} G^i_3 \eta_i + p_i \overline{\eta}_i , \,^{17}
\tag{227}
$$

which encodes the deleted set $p_1 = p_2 = 0$ of the hybrid phase. We then denote by $\mathcal{B}_1(k)_r$, a complex of Wilson line branes whose leftmost entry is $\mathcal{W}(k)_r$. For example, we have

$$
\mathcal{B}_1(0)_0 : \quad \mathcal{W}(0)_0 \xleftrightarrow{\hspace{1cm}} \mathcal{W}(3)^{\oplus 2}_{1-6\kappa} \xleftrightarrow{\hspace{1cm}} \mathcal{W}(6)_{2-12\kappa} .
\tag{228}
$$

---

[16]There is some ambiguity as to what should be called $\widehat{\Gamma}$ and $\widehat{\Gamma}^*$ for this hybrid. This is linked to the GLSM charge "sign flip", which is required to make contact with the hybrid geometry.

[17]In the following, and in the appendices, we adhere to the following labelling conventions for the branes. If a GLSM brane is a lift of a hybrid brane discussed in section 4 with an integer label $\mathfrak{B}_k$, $k = 1, 2, \ldots$, we will call the corresponding GLSM brane $\mathcal{B}_k$. We will also construct GLSM branes which can be understood as lifts from the geometric phase. We label these branes with letters from the beginning of the alphabet, e.g. $\mathcal{B}_a, \mathcal{B}_b$. If branes are grade-restricted, they have an interpretation in both phases, so the same GLSM brane can have two labels, depending on whether it is understood as a lift from the geometric or hybrid phase.

The brane factor for $\mathcal{B}_1(0)_0$ is

$$f_{\mathcal{B}_1(0)_0} = (-1 + e^{6\pi(-i\kappa+\sigma)})^2.$$ (229)

This leads to a vanishing hemisphere partition function in the hybrid phase. This is easy to see by evaluating the brane factor in the hybrid phase:

$$f_{\mathcal{B}_1(0)_0}|_{hyb} = (-1 + \gamma^{-3\delta}e^{-H})^2 = (-1 + (1 - H + \mathcal{O}(H^2)))^2 = \mathcal{O}(H^2).$$ (230)

The integral over $\mathbb{P}^1$ in (220) always gives zero with a brane factor of this form.

We now seek to identify the GLSM empty branes $\mathcal{B}_1(k)_r$ with members of the family $\mathfrak{B}_1^{(a_0,r_0,\alpha_0,n)}$ constructed in section 4.2. To do so, we will utilise the lifting procedure described in section 5.1.3. We note that for the case at hand, there are no point-like fields as defined in section 5.1.3. Instead, the $\mathbb{Z}_3$ orbifold arises from the non-minimal gauge charge of two base-like fields $p_1$ and $p_2$. The bicubic GLSM also possesses six fibre-like fields $x_i$, $i = 1, 2, \ldots, 6$.[18] To apply the hybrid to GLSM map of section 5.1.3, we need only to realise global matrix factorisations in projective coordinates $[p_1 : p_2]$ and $x_1, \ldots, x_6$, then lift these coordinates to $V \cong \mathbb{C}^8$. Upon applying the map of section 5.1.3, a set of GLSM matrix factorisations corresponding to the hybrid B-branes $\mathfrak{B}_1^{(a_0,r_0,\alpha_0,n)}$ is

$$Q_1^{(n)} = p_1^{1-\frac{n}{3}} G_3^1 \eta_1 + p_1^{\frac{n}{3}} \overline{\eta}_1 + G_3^2 \eta_2 + p_2 \overline{\eta}_2, \quad n = 0, 1, 2, 3.$$ (231)

We note that since $p_1$ and $p_2$ are affine coordinates in the GLSM, only the $n = 0$ and $n = 3$ cases give polynomial-valued GLSM matrix factorisations.[19] We can then compute the corresponding complexes of Wilson line branes

$$\mathcal{W}(3a_0 + n + 3)_{r_0+2} \xrightleftharpoons[\mathcal{W}(3a_0 + 3)_{r_0+1}]{\mathcal{W}(3a_0 + n)_{r_0+1}} \oplus \xrightleftharpoons{} \mathcal{W}(3a_0)_{r_0}.$$ (232)

In particular, we note that we cannot make a choice of vacuum charges for which (228) and (232) match. However, we recall that the bicubic GLSM assigned charges of $-3$ to the fields $p_1$ and $p_2$, along with charges of $+1$ to the fields $x_i$. This is the opposite sign to the charge assignments of the hybrid fields on $Y = \text{tot}\left(\mathcal{O}\left(-\frac{1}{3}\right) \to \mathbb{P}^1\right)$, and so our lift must also include a reversing of the hybrid gauge charges. Taking this sign flip into account, we find that the choice $(a_0, r_0, \alpha_0, n) = \left(\frac{k}{3} - 2, r - 2, \alpha_0, 3\right)$ leads to the empty GLSM branes $\mathcal{B}_1(k)_r$

$$\pi^{hyb}(\mathcal{B}_1(k)_r) = \mathfrak{B}_1^{\left(\frac{k}{3}, r-2, 3\right)}.$$ (233)

In the GLSM, we also have trivial branes with matrix factorisation

$$Q_2 = W\eta + \overline{\eta}.$$ (234)

This leads to complexes of Wilson line branes $\mathcal{B}_2(k)_r$ of the form

$$\mathcal{W}(k)_r \xrightleftharpoons{} \mathcal{W}(k)_{r+1}.$$ (235)

---

[18]This is somewhat confusing notationally, since in our general discussion of lifting hybrid branes, we used the letter $p$ to denote point-like fields. However, using $p_i$ to denote the base-like fields is the standard convention in this model.

[19]The non-polynomial lifts may be able to be interpreted as some sort of gerby GLSM matrix factorisations. It would be interesting to investigate this further. Here, we will focus only on the $n = 0$ and $n = 3$ cases.

Obviously, the corresponding brane factor is zero, and so is the hemisphere partition function in any phase. We recall that these trivial branes can also be realised directly in the IR hybrid model by globalising the following local matrix factorisation

$$Q_{2,u} = W_u \eta + \overline{\eta}, \tag{236}$$

resulting in the hybrid B-branes $\mathfrak{B}_2^{(a_0,r_0,\alpha_0)}$. For the choice of vacuum charges $(a_0, r_0, \alpha_0) = \left(\frac{q}{3}, r+1, \alpha_0\right)$, these hybrid B-branes clearly lift to the complexes of Wilson line branes (235). We have thus confirmed that at the GLSM level, the hybrid B-branes $\mathfrak{B}_2^{(a_0,r_0,\alpha_0)}$ and $\mathfrak{B}_1^{(a_0,r_0,\alpha_0,n)}$ do indeed correspond to trivial and empty branes respectively.

Next, we consider a GLSM lift of the canonical matrix factorisation (179). This class of branes was also considered in the context of D-brane transport in [19]. Consider the following GLSM matrix factorisation

$$Q_3 = \sum_{i=1}^{6} x_i \eta_i + \frac{1}{3} \frac{\partial W}{\partial x_i} \overline{\eta}_i. \tag{237}$$

This matrix factorisation describes empty branes in the geometric phase, see $Q_a$, (B.21), in appendix B. Choosing a complex of Wilson line branes as in the appendix, which we now give the name $\mathcal{B}_3$, and evaluating the brane factor $f_{\mathcal{B}_3}$, (B.22), in the hybrid phase, we find

$$f_{\mathcal{B}_3}|_{hyb} = (-1 + \gamma^{-\delta} e^{-\frac{H}{3}})^6 = (-1 + \gamma^{-\delta})^6 - 2\gamma^{-\delta}(-1 + \gamma^{-\delta})^5 H + \mathcal{O}(H^2). \tag{238}$$

Recalling our proposed interpretation of $f_\delta^a$ in terms of geometric branes of the base, all four charges of this brane will be non-zero, so this brane has both D0 and D2-components on the base $\mathbb{P}^1$. This behaviour is analogous to what is known for the canonical matrix factorisation in Landau-Ginzburg orbifolds [5, 24]. We emphasise that by evaluating the brane factor in the hybrid phase, we have not computed the analytic continuation of the empty brane of the geometric phase to the hybrid phase. This is an invalid picture, since the brane is not grade-restricted. Rather, the above complex of Wilson line branes should be viewed as a "local" lift of the canonical matrix factorisation of the hybrid phase to the GLSM that is only consistent in the hybrid phase. We now seek to identify this GLSM brane as a lift of a hybrid B-brane in the family $\mathfrak{B}_3^{(a_0,r_0,\alpha_0)}$. Applying the procedure of sections 3.7 and 5.1.3, we can obtain, upon making appropriate choices for the GLSM gauge group action $\rho_M(g)$, the GLSM matrix factorisation (B.21). Since the bicubic GLSM has no base-like fields, the $U(1)$ gauge charges are simply given by the hybrid gauge charges. The only complication is that since this hybrid model is described by a $\mathbb{Z}_3$-gerbe, we must multiply these charges by 3 to obtain integer $U(1)$ charges in the corresponding GLSM. This is consistent with our interpretation of $Y$ as the total space of an orbibundle over a $\mathbb{P}^1$. The resulting complex of Wilson line branes is

$$\mathcal{W}(0)_0 \rightleftarrows \mathcal{W}(1)_{\frac{1}{3}}^{\oplus 6} \rightleftarrows \dots \rightleftarrows \mathcal{W}(1)_{\frac{5}{3}}^{\oplus 6} \rightleftarrows \mathcal{W}(6)_2, \tag{239}$$

where we have set $a_0 = -2$, $r_0 = 2$, $\alpha_0 = 0$ and performed the usual flip of the bicubic gauge charges. Note that with the identification $\kappa = \frac{1}{3}$, this is exactly the GLSM brane (B.22).

Finally, we construct a GLSM brane which reduces to a pure D0-brane (183) in the base $B$. Consider the GLSM matrix factorisation

$$Q_4 = \sum_{i=1}^{6} x_i \eta_i + \frac{1}{3} \frac{\partial W}{\partial x_i} \overline{\eta}_i + f(p_1, p_2) \overline{\eta}_7, \tag{240}$$

where $f(p_1, p_2) = \alpha p_1 + \beta p_2$. Upon making a suitable choice of vacuum charges, this matrix factorisation describes a brane given by the following complex of Wilson line branes associated

to a GLSM brane $\mathcal{B}_4$:

$$\mathcal{W}(0)_0 \xrightleftharpoons{\quad} \begin{matrix} \mathcal{W}(1)^{\oplus 6}_{1-2\kappa} \\ \oplus \\ \mathcal{W}(3)_{1-6\kappa} \end{matrix} \xrightleftharpoons{\quad} \begin{matrix} \mathcal{W}(2)^{\oplus 15}_{2-4\kappa} \\ \oplus \\ \mathcal{W}(4)^{\oplus 6}_{2-8\kappa} \end{matrix} \xrightleftharpoons{\quad} \dots \xrightleftharpoons{\quad} \mathcal{W}(9)_{7-18\kappa}. \tag{241}$$

This brane is not grade-restricted, so the IR images of the brane in the geometric and hybrid phases are not related by analytic continuation. The brane factor is

$$f_{\mathcal{B}_4} = -(-1 + e^{2\pi(-i\kappa+\sigma)})^7 (1 + e^{2\pi(-i\kappa+\sigma)} + e^{4\pi(-i\kappa+\sigma)}). \tag{242}$$

Evaluation in the hybrid phase and extracting the four components gives

$$f^{\mathbf{1}}_{\mathcal{B}_4, \delta=1,2} = 0, \qquad f^{H}_{\mathcal{B}_4, \delta=1,2} = 9(-2 + \gamma + \gamma^2). \tag{243}$$

So, in agreement with expectations, this brane only has a D0-component in the base, and no D2-component. Once again, we can identify this GLSM B-brane as a lift of a hybrid B-brane. In particular, we will see that it is a member of the family $\mathfrak{B}_4^{(a_0, r_0, \alpha_0, n)}$. Upon lifting these hybrid B-branes to the GLSM, we find the following family of GLSM matrix factorisations

$$Q_4^{(n)} = \sum_{i=1}^{6} \left( x_i \eta_i + \frac{1}{3} \frac{\partial W_{GLSM}}{\partial x_i} \overline{\eta}_i \right) + 0\eta_7 + p_1^n (\alpha p_1 + \beta p_2) \overline{\eta}_7. \tag{244}$$

From this perspective, it is quite clear that this brane represents a configuration of a single D0 brane at $\frac{p_2}{p_1} = -\frac{\alpha}{\beta}$ and $n$ D0 branes at $\frac{p_2}{p_1} = \infty$. Furthermore, we observe that $Q_4^{(n=0)}$ is exactly the GLSM matrix factorisation given above. Upon setting the vacuum charges to $(a_0, r_0, \alpha_0) = (-3, 1, \alpha_0)$, we obtain the following complexes of Wilson line branes

$$\mathcal{W}(n)_0 \xrightleftharpoons{\quad} \begin{matrix} \mathcal{W}(1-n)^{\oplus 6}_{\frac{1}{3}} \\ \oplus \\ \mathcal{W}(3)_{-1} \end{matrix} \xrightleftharpoons{\quad} \begin{matrix} \mathcal{W}(2-n)^{\oplus 15}_{\frac{2}{3}} \\ \oplus \\ \mathcal{W}(4)^{\oplus 6}_{-\frac{2}{3}} \end{matrix} \xrightleftharpoons{\quad} \dots \xrightleftharpoons{\quad} \mathcal{W}(9)_1. \tag{245}$$

We thus see that the above GLSM B-brane corresponds to the case in which we have no D0-insertions at the north pole, i.e. this brane is the lift of the hybrid B-brane $\mathfrak{B}_4^{(3,3,\alpha_0,0)}$.

### 5.2.4 Basis of hybrid branes

We would like to identify a set of objects that gives a basis of the charge lattice in the hybrid phase. To do so, we aim to find branes with minimal charge that generate the charge lattice. In the case of Landau-Ginzburg orbifolds it is known [46–48] that the canonical matrix factorisations of type (237), which are related to certain Recknagel-Schomerus boundary states in the associated Gepner model, do not generate the charge lattice. The objects to consider instead are certain linear matrix factorisations associated to permutation branes in the CFT, which can be constructed if we choose the form (206) for $G_3^i$ in the superpotential.

We consider the following matrix factorisation, which is associated to a D2-brane $\mathcal{B}_c$ of minimal charge in the geometric phase, see appendix B,

$$\begin{aligned} Q_5 = {}& f_{12}\eta_1 + p_1 g_{12}\overline{\eta}_1 + f_{45}\eta_2 + p_2 g_{45}\overline{\eta}_2 + x_3\eta_3 + (p_1 x_3^2 - 3\alpha p_2 x_1 x_2)\overline{\eta}_3 \\ & + x_6\eta_4 + (p_2 x_6^2 - 3\alpha p_1 x_4 x_5)\overline{\eta}_4, \end{aligned} \tag{246}$$

where $f_{ij}$ and $g_{ij}$ are defined in (B.20). This corresponds to a Koszul complex with five entries. We choose the brane

$$\mathcal{W}(0)_0 \rightleftarrows \mathcal{W}(1)^{\oplus 4}_{1-2\kappa} \rightleftarrows \mathcal{W}(2)^{\oplus 6}_{2-4\kappa} \rightleftarrows \mathcal{W}(3)^{\oplus 4}_{3-6\kappa} \rightleftarrows \mathcal{W}(4)_{4-8\kappa}, \qquad (247)$$

and call it $\mathcal{B}_5$. We can associate the following brane factor to this matrix factorisation

$$f_{\mathcal{B}_5} = -(1 - e^{2\pi(-i\kappa+\sigma)})^4. \qquad (248)$$

This brane is also automatically grade restricted and can be given an interpretation in terms of the hybrid phase: the coefficients of the $\eta_i$ are all given in terms of the fibre coordinates, which suggests that this brane describes a D2-brane in the base and a permutation brane in the fibre. We can generate further branes by shifting the complex by 1, and the R-charge by $1-2\kappa$. This modifies the brane factor by an overall factor of $e^{i\pi(1-2\kappa)}e^{2\pi\sigma}$. Iterating these shifts generates a set of branes. In the context of Landau-Ginzburg orbifolds with orbifold group $\mathbb{Z}_d$, these "fractional", branes [24, 49] are related by monodromies around the Landau-Ginzburg point. If $d$ is larger than the dimension of the charge lattice, a subset of these branes gives a linearly independent basis. The shifts are $d$-periodic. The situation for our hybrid example is a little more subtle. The orbifold group is $\mathbb{Z}_3$, so two more branes are in the same $\mathbb{Z}_3$-orbit and one might naively expect to find at most three independent branes. However, the periodicity of the shifts is spoiled by the non-trivial fibration structure. This is easily seen in the brane factors. Let us consider a single Wilson line brane $\mathcal{W}(q_i)_{\alpha+\beta(2\kappa)}$, with $\alpha, \beta \in \mathbb{Z}$. The brane factor evaluated in the hybrid phase with $\kappa = \frac{1}{3}$ is given in (222). Any shift in $q_i$ not only shifts the orbifold element $\gamma$, but also the base-dependent factor $e^{-\frac{H}{3}}$. Hence, we can construct a set of four branes by shifting the complex (247) three times. We note that this lack of periodicity is not a generic feature of hybrid branes. The situation will be different in the two-parameter model associated to the octic hybrid, to be discussed in section 5.3. Denoting the associated branes by $\mathcal{B}_{5,j}$ ($j = 0, 1, 2, 3$), the associated brane factors, evaluated in the hybrid phase, are

$$f_{\mathcal{B}_{5,j}}|_{hyb} = (-1)^j e^{-j\frac{H}{3}} \gamma^{-j\delta} (1 - e^{-\frac{H}{3}}\gamma^{-\delta})^4. \qquad (249)$$

The four components in the expansion in $\delta$ and $H$ give the charge vectors of each of the four branes labelled by $j$. Only the brane with $j = 0$ is grade restricted to our chosen window, the other three branes are not. Since we opt to transport geometric branes to the hybrid phase we will not grade restrict the hybrid branes.

### 5.2.5   D-brane transport

A construction and analysis of GLSM branes that produce a basis of branes in the geometric phase can be found in appendix B. This gives all the information needed to analytically continue the large volume branes to the hybrid phase and to express them in terms of our chosen hybrid basis. For the large volume D0-brane $\mathcal{B}_b$, (B.26), we compute

$$f_{\mathcal{B}_b}|_{hyb} = (1 - \gamma^{-\delta}e^{-\frac{H}{3}})^5 = (1 - \gamma^{-\delta})^5 + \frac{5}{3}(1 - \gamma^{-\delta})^4 \gamma^{-\delta} H + \mathcal{O}(H^2). \qquad (250)$$

For the D2-brane $\mathcal{B}_c$, (B.31), we get

$$f_{\mathcal{B}_c}|_{hyb} = (1 - \gamma^{-\delta}e^{-\frac{H}{3}})^4 = (1 - \gamma^{-\delta})^4 + \frac{4}{3}(1 - \gamma^{-\delta})^3 \gamma^{-\delta} H + \mathcal{O}(H^2). \qquad (251)$$

The grade-restricted brane factor for the D4-brane $\mathcal{B}_d^w$ evaluates to

$$f_{\mathcal{B}_d^w}|_{hyb} = 3(1 - \gamma^{-\delta}e^{-\frac{H}{3}})^3(2 - 4\gamma^{-\delta}e^{-\frac{H}{3}} + 5\gamma^{-2\delta}e^{-\frac{2H}{3}})$$
$$= 3(1 - \gamma^{-\delta})^3(2 - 4\gamma^{-\delta} + 5\gamma^{-2\delta}) + (1 - \gamma^{-\delta})^2(10 - 26\gamma^{-\delta} + 25\gamma^{-2\delta})\gamma^{-\delta} H + \mathcal{O}(H^2). \qquad (252)$$

For the grade restricted D6-brane $\mathcal{B}_e^w$ we get

$$
\begin{aligned}
f_{\mathcal{B}_e^w}|_{hyb} &= 3\gamma^{-\delta}e^{-\frac{H}{3}}(1-\gamma^{-\delta}e^{-\frac{H}{3}})^2(2-\gamma^{-\delta}e^{-\frac{H}{3}})(2+2\gamma^{-2\delta}e^{-\frac{2H}{3}}) \\
&= 3\gamma^{-\delta}(1-\gamma^{-\delta})^2(2-\gamma^{-\delta}+2\gamma^{-2\delta}) \\
&\quad + 2\gamma^{-\delta}(1-\gamma^{-\delta})(1-4\gamma^{-\delta}+5\gamma^{-2\delta}-5\gamma^{-3\delta})H + \mathcal{O}(H^2).
\end{aligned}
\tag{253}
$$

The GLSM lift of the canonical matrix factorisation (237) yields a different brane factor which evaluates to the same expression in the hybrid phase. In agreement with expectations, the canonical matrix factorisation can thus be seen as the object corresponding to the structure sheaf in the hybrid phase.

Inserting explicit values for $\delta$, we find that all expressions have non-zero $\mathcal{O}(1)$ and $\mathcal{O}(H)$-components for $\delta = 1, 2$, and are zero for $\delta = 3$, consistent with the broad sectors. We now can extract the charge vectors of each brane, and express them in terms of the charge vectors of the hybrid branes $\mathcal{B}_{5,j}$. Taking into account that the brane factors evaluated in the hybrid phase should be identified with the Chern characters of the hybrid branes, this calculation determines the analytic continuation matrix $T_w$ defined as $(f_{\mathcal{B}_b}, f_{\mathcal{B}_c}, f_{\mathcal{B}_d^w}, f_{\mathcal{B}_e^w})^T|_{hyb} = T_w \cdot (f_{\mathcal{B}_{5,0}}, f_{\mathcal{B}_{5,1}}, f_{\mathcal{B}_{5,2}}, f_{\mathcal{B}_{5,3}})^T|_{hyb}$. We find

$$
T_w = \begin{pmatrix} 1 & 1 & 0 & 0 \\ 1 & 0 & 0 & 0 \\ 5 & 9 & 3 & -1 \\ -1 & -2 & -1 & 0 \end{pmatrix}.
\tag{254}
$$

The matrix has determinant 1, and can be inverted over the integers, indicating that we have found a basis of minimally charged branes in both phases. Grade-restricting the hybrid branes, transporting them through the same window, then expressing the brane factors in terms of the large volume branes will give the inverse of this matrix. We have made this consistency check for some examples of branes, but omit the details here.

## 5.3 The octic GLSM

In this section, we consider the two-parameter GLSM which realises the octic hybrid discussed in section 4.1 as one of its phases. A detailed discussion of the GLSM of this model can be found in [50]. D-brane transport with focus on the connection between the Landau-Ginzburg and Calabi-Yau phases of this model has been worked out in [5]. Our main focus will be on the hybrid branes and on transporting branes between the geometric and hybrid phases.

### 5.3.1 GLSM and phases

The GLSM for this model has $\mathsf{G} = U(1)^2$. The matter content is

|         | $p$          | $x_6$             | $x_3$       | $x_4$       | $x_5$       | $x_1$       | $x_2$       | FI      |
|---------|--------------|-------------------|-------------|-------------|-------------|-------------|-------------|---------|
| $U(1)_1$ | $-4$         | $1$               | $1$         | $1$         | $1$         | $0$         | $0$         | $\zeta_1$ |
| $U(1)_2$ | $0$          | $-2$              | $0$         | $0$         | $0$         | $1$         | $1$         | $\zeta_2$ |
| $U(1)_V$ | $2-8\kappa_1$ | $2\kappa_1-4\kappa_2$ | $2\kappa_1$ | $2\kappa_1$ | $2\kappa_1$ | $2\kappa_2$ | $2\kappa_2$ |         |

(255)

The R-charge ambiguities satisfy $0 \leq \kappa_1 \leq \frac{1}{4}$ and $0 \leq \kappa_2 \leq \frac{1}{8}$. The superpotential is $\mathsf{W} = pG_{(4,0)}(x_1, \ldots, x_6)$, where $G_{(4,0)}(x_1, \ldots, x_6)$ is generic and of degree $(4, 0)$. We use the form, consistent with section 4.1 and [5],

$$
\mathsf{W} = p\left(x_3^4 + x_4^4 + x_5^4 + x_6^4(x_1^8 + x_2^8)\right),
\tag{256}
$$

as this will make it possible to describe minimally charged geometric branes which do not come from branes in the ambient space.

The hybrid phase is at $\zeta_1 \ll 0, \zeta_2 \gg 0$. We have a base $\mathbb{P}^1$ with coordinates $\{x_1, x_2\}$. The Landau-Ginzburg orbifold fibred over this base manifold has $G = \mathbb{Z}_4$ with fibre coordinates $\{x_3, \ldots, x_6\}$. The corresponding hybrid geometry is $Y = \text{tot}(O(-2) \oplus \mathcal{O}^{\oplus 3} \to \mathbb{P}^1)$, so we recover the model of section 4.1. To match with the vector R-charges in the hybrid theory, we have to choose $\kappa_1 = \frac{1}{4}, \kappa_2 = 0$.

The geometric phase is at $\zeta_1 \gg 0, \zeta_2 \gg 0$. It is a smooth Calabi-Yau hypersurface in the toric ambient space determined by the $U(1)^2$ weights of $x_1, \ldots, x_6$, namely the toric resolution of $\mathbb{P}^4_{11222}$. The Calabi-Yau is a K3 fibration over a $\mathbb{P}^1$-base. The transition to the hybrid phase can be understood as the K3 fibre undergoing a Landau-Ginzburg transition, while the base $\mathbb{P}^1$ remains unchanged.

If we want to transport our branes to the hybrid phase, there is a band restriction rule. This has been worked out in [5, p.174]. GLSM B-branes in this model are complexes of Wilson line branes $\mathcal{W}(q_1, q_2)_r$, with gauge charges $(q_1, q_2)$, and R-charge $r$. We choose the following window for $q_1$:

$$q_1 \in \{0, 1, 2, 3\}. \tag{257}$$

As long as we only transport between the geometric phase and the hybrid phase, there is no restriction on the brane charge $q_2$, hence the name band restriction rule. There is a "global" grade restriction rule, given in [5], that is valid for transporting branes between any phases, but we will not impose it here.

### 5.3.2 Hemisphere partition function

Inserting the defining data of the octic GLSM into the definition of the hemisphere partition function, we get

$$Z_{D^2}(\mathcal{B}) = C \int d^2\sigma \, \Gamma(-4i\sigma_1 + 1 - 4\kappa_1) \Gamma(i\sigma_1 - 2i\sigma_2 + \kappa_1 - 2\kappa_2) \Gamma(i\sigma_1 + \kappa_1)^3 \Gamma(i\sigma_2 + \kappa_2)^2$$
$$\times e^{i(t_1\sigma_1 + t_2\sigma_2)} f_{\mathcal{B}}(\sigma_1, \sigma_2). \tag{258}$$

A discussion of geometric phases and a basis of geometric branes can be found in appendix C.

Let us compute the hemisphere partition function for the hybrid phase. For later reference, we recall the building blocks of the result in the IR [36]. The $I$-function in the $\delta$th narrow sector is ($\delta = 1, 2, 3$)

$$I_\delta(t_1, t_2, H) = \frac{\Gamma\left(1 + \frac{H}{2\pi i}\right)^2}{\Gamma\left(\frac{\delta}{4} + \frac{H}{\pi i}\right) \Gamma\left(\frac{\delta}{4}\right)^3} e^{-t_2 \frac{H}{2\pi i}}$$
$$\times \sum_{a,n \geq 0} \frac{\Gamma\left(a + \frac{\delta}{4} + 2n + 2\frac{H}{2\pi i}\right) \Gamma\left(a + \frac{\delta}{4}\right)^3}{\Gamma(4a + \delta) \Gamma\left(1 + n + \frac{H}{2\pi i}\right)^2} e^{\frac{t_1}{4}(4a + \delta - 1)} e^{-t_2 n}, \tag{259}$$

where $H$ is the hyperplane class of $\mathbb{P}^1$. For the Gamma class we have

$$\widehat{\Gamma}_\delta(H) = \Gamma\left(\frac{\delta}{4} + \frac{H}{\pi i}\right) \Gamma\left(\frac{\delta}{4}\right)^3 \Gamma\left(1 - \frac{H}{2\pi i}\right)^2. \tag{260}$$

To account for the poles that contribute in the hybrid phase, we take

$$-4i\sigma_1 = -k_1 + \varepsilon_1 - 1 + 4\kappa_1, \qquad i\sigma_2 = -k_2 + \varepsilon_2 - \kappa_2. \tag{261}$$

From now on, we will set $\kappa_1 = \frac{1}{4}, \kappa_2 = 0$. Inserting this, and making use of the reflection formula for the Gamma function, we get

$$
Z_{D^2}^{hyb} = \frac{C}{4i} \sum_{k_i} \oint d^2\varepsilon \frac{\pi^3(-1)^{k_1+2k_2}}{\sin \pi\varepsilon_1 \sin^2 \pi\varepsilon_2} \frac{\Gamma\left(\frac{k_1+1}{4} + 2k_2 - \frac{\varepsilon_1}{4} - 2\varepsilon_2\right)\Gamma\left(\frac{k_1+1}{4} - \frac{\varepsilon_1}{4}\right)^3}{\Gamma(k_1+1-\varepsilon_1)\Gamma(1+k_2-\varepsilon_2)^2}
$$
$$
\times e^{t_1\left(\frac{k_1}{4} - \frac{\varepsilon_1}{4}\right)}e^{-t_2(k_2-\varepsilon_2)}f_{\mathcal{B}}\left(\frac{i}{4}(-k_1+\varepsilon_1), -i(-k_2+\varepsilon_2)\right). \tag{262}
$$

To account for the $\mathbb{Z}_4$-orbifold in the fibre, we set

$$
k_1 = 4n + (\delta - 1), \qquad n \in \mathbb{Z}_{\geq 0}, \quad \delta = 1, 2, 3. \tag{263}
$$

The integrand has a first order pole in $\varepsilon_1$. Carrying out the $\varepsilon_1$-integration, we find

$$
Z_{D^2}^{hyb} = \frac{(2\pi i)C}{4i} \sum_{n,k_2} \sum_{\delta=1}^{3} \oint d\varepsilon_2 \frac{\pi^3(-1)^{4n+\delta-1+2k_2}}{\sin^2 \pi\varepsilon_2} \frac{\Gamma\left(n + \frac{\delta}{4} + 2k_2 - 2\varepsilon_2\right)\Gamma\left(n + \frac{\delta}{4}\right)^3}{\Gamma(4n+\delta)\Gamma(1+k_2-\varepsilon_2)^2}
$$
$$
\times e^{t_1\left(n + \frac{\delta-1}{4}\right)}e^{-t_2(k_2-\varepsilon_2)}f_{\mathcal{B}}\left(\frac{i}{4}(-4n - (\delta-1)), -i(-k_2+\varepsilon_2)\right). \tag{264}
$$

To rewrite the second integral as an integral over $\mathbb{P}^1$, we set $\varepsilon_2 = -\frac{H}{2\pi i}$ and use

$$
\int_{\mathbb{P}^1} g(H) = \oint dH \frac{1}{H^2} g(H), \qquad \sin^2 \frac{H}{2i} = \frac{1}{(2i)^2} \frac{(1-e^{-H})^2}{e^{-H}}. \tag{265}
$$

Furthermore using the definition of the Todd class, the Gamma class, and the $I$-function, we can write the hemisphere partition function in the hybrid phase as

$$
Z_{D^2}^{hyb} = -\pi(2\pi i)^2 \frac{C}{4i} \sum_{\delta=1}^{3} \int_{\mathbb{P}^1} (-1)^{\delta-1} \widehat{\Gamma}_\delta(H) I_\delta(t_1, t_2, H)\text{ch}_\delta(\mathcal{B}). \tag{266}
$$

Choosing $C$ so that the overall factor is $\frac{1}{4}$ to account for the contribution of the pairing (192) in the state space, we propose

$$
\text{ch}_\delta(\mathfrak{B}) = f_{\mathcal{B}}\left(\frac{i}{4}(-4n - (\delta-1)), -i(-k_2+\varepsilon_2)\right). \tag{267}
$$

For a single Wilson line brane $\mathcal{W}(q_1, q_2)_{\alpha+\beta(2\kappa_1)+\gamma(2\kappa_2)}$ $(\alpha, \beta, \gamma \in \mathbb{Z})$ with $\kappa_1 = \frac{1}{4}, \kappa_2 = 0$, this becomes

$$
f_{\mathcal{W}(q_1,q_2)_{\alpha+\beta(2\frac{1}{4})}}\big|_{hyb} = e^{i\pi r}\gamma^{-q_1(\delta-1)}e^{q_2 H} = e^{i\pi\alpha}\gamma^\beta\gamma^{-q_1(\delta-1)}e^{q_2 H}, \tag{268}
$$

where $\gamma = e^{\frac{2\pi i}{4}}$. In contrast to the bicubic example, shifts in $q_1$ and $q_2$ now act independently on the fibre and base directions respectively. The same reasoning as for the bicubic GLSM implies that the component $f_{\mathcal{B},\delta}^H$ computes a D0-charge with respect to the base $B$, and that $f_{\mathcal{B},\delta}^1$ is a D2-charge.

### 5.3.3 Hybrid branes

Let us provide some examples of GLSM branes and establish a connection to the hybrid branes discussed in section 4.1.1. Most of these branes are permutation-type matrix factorisations of the superpotential (256) which are automatically band restricted, and often also have a simple interpretation in the geometric phase, see appendix C for details on this. We go through the branes of section 4.1.1 in order of their appearance.

We start off with the GLSM empty branes. To this end, consider the matrix factorisation

$$Q_1 = G_{(4,0)}\eta + p\overline{\eta}. \tag{269}$$

A simple example of a complex of Wilson line branes associated to this matrix factorisation is

$$\mathcal{B}_1: \qquad \mathcal{W}(0,0)_0 \overset{\longrightarrow}{\longleftarrow} \mathcal{W}(4,0)_{1-8\kappa_1}. \tag{270}$$

Any shifts of this complex will also describe empty branes in the hybrid phase. It is easy to see at the level of the brane factor that this is an empty brane. When evaluated in the hybrid phase using (268), the brane factor is

$$f_{\mathcal{B}_1} = 1 + e^{i\pi}\gamma^{-4}\gamma^{-4(\delta-1)} = 0. \tag{271}$$

As for all empty branes, this brane is not grade-restricted to any window, and hence is not globally defined on the moduli space. When interpreted with respect the the geometric phase, the brane represents the structure sheaf of the Calabi-Yau, see appendix C. We now identify this GLSM B-brane as the lift of a hybrid B-brane belonging to the family $\mathfrak{B}_1^{(a_0,r_0,\alpha_0)}$. To do so, we first use the procedure of section 3.7 to rewrite the corresponding octic hybrid global matrix factorisation (140) in weighted projective coordinates

$$Q_1 = \left(x_3^4 + x_4^4 + x_5^4 + (x_1^8 + x_2^8)x_6^4\right)\eta + \overline{\eta}, \tag{272}$$

where $[x_1 : x_2]$ are homogeneous coordinates on the base $\mathbb{P}^1$ with $u = \frac{x_1}{x_2}$. We now apply the lifting procedure of section 5.1.3. Firstly, we note that the octic GLSM has two base-like fields $x_1, x_2$, and four fibre like fields $x_3, \ldots, x_6$. These fields all correspond to the GLSM lifts of the weighted projective coordinates appearing in the above factorisation. In contrast to the bicubic GLSM, the octic GLSM also possesses a single point-like field $p$. To construct a GLSM lift of (140), we thus apply $\rho_M(g)$ with parameter $z$, then set $z^4 = p$. For an appropriate choice of GLSM boundary gauge charges, the result is

$$Q(x_i, p) := \rho_M(z)^{-1}Q(z \cdot x_i)\rho_M(z)|_{z^4=p} = G_{(4,0)}\eta + p\overline{\eta}. \tag{273}$$

This is of course the GLSM matrix factorisation given above. Noting that the first and second GLSM $U(1)$ charges correspond to the hybrid orbifold and gauge charges respectively, we find the following family of complexes of Wilson line branes

$$\mathcal{W}(\alpha_0 - 4, a_0)_{r_0+1} \overset{\longrightarrow}{\longleftarrow} \mathcal{W}(\alpha_0, a_0)_{r_0}, \tag{274}$$

which for the choice of vacuum charges $a_0 = 0$, $r_0 = -1$, and $\alpha_0 = 4$, corresponds exactly to the complex of Wilson line branes given above. We have thus shown that with the identification $\kappa_1 = \frac{1}{4}$, this brane indeed corresponds to a lift of $\mathfrak{B}_1^{(0,-1,4)}$ to the GLSM. From the hybrid perspective, the first charge entry in the Wilson line branes in the above complex is a choice made modulo 4. This choice corresponds to lifting $\mathfrak{B}_1^{(0,-1,0)}$ in different ways. In particular, the GLSM matrix factorisation $Q = W\eta + \overline{\eta}$ is also a valid lift of $Q_1$. However, from the GLSM perspective, this brane corresponds to the trivial matrix factorisation which represents a trivial brane in every phase, and the associated brane factor is identically zero, while our choice of lift, $Q_1$, is an empty brane in the sense that it becomes trivial in the hybrid phase but represents a non-trivial brane in other phases. Next, we identify GLSM lifts of the hybrid empty branes $\mathfrak{B}_2^{(a_0,r_0,\alpha_0,1)}$. To do so, we begin by considering the GLSM matrix factorisation

$$Q_2 = \sum_{i=1}^{5}\left(x_i\eta_i + \frac{1}{d_i}\frac{\partial W}{\partial x_i}\right), \tag{275}$$

where the $d_i$ are the exponents of the $x_i$ in (256). An associated complex of Wilson line branes is

$$\mathcal{W}(0,0)_0 \xleftrightarrow{\quad} \begin{array}{c} \mathcal{W}(1,0)^{\oplus 3}_{1-2\kappa_1} \\ \oplus \\ \mathcal{W}(0,1)^{\oplus 2}_{1-2\kappa_2} \end{array} \xleftrightarrow{\quad} \dots \xleftrightarrow{\quad} \begin{array}{c} \mathcal{W}(3,1)^{\oplus 2}_{4-6\kappa_1-2\kappa_2} \\ \oplus \\ \mathcal{W}(2,2)^{\oplus 3}_{4-4\kappa_1-4\kappa_2} \end{array} \xleftrightarrow{\quad} \mathcal{W}(3,2)_{5-6\kappa_1-4\kappa_2}.$$

(276)

We call this brane $\mathcal{B}_2$. The brane factor evaluated in the hybrid phase is

$$f_{\mathcal{B}_2} = (1 - \gamma^{-\delta})^3 (1 - e^H)^2 = \mathcal{O}(H^2).$$

(277)

The hemisphere partition function, evaluated in the hybrid phase, is thus zero, confirming that this is indeed an empty brane. We now argue that this GLSM B-brane represents a lift of a hybrid B-brane belonging to the family $\mathfrak{B}_2^{(a_0, r_0, \alpha_0, 1)}$. Applying the procedure of section 3.7, we find that the corresponding global matrix factorisation can be realised as

$$Q_2^{(1)} = x_1\eta_1 + x_1^7 x_6^4\overline{\eta}_1 + x_2\eta_2 + x_2^7 x_6^4\overline{\eta}_2 + \sum_{a=3}^5 (x_a\eta_a + x_a^3\overline{\eta}_a),$$

(278)

where $[x_1 : x_2 : \dots : x_6]$ are weighted homogeneous coordinates on $Y$. We lift this endomorphism to the GLSM by instead regarding the fields $x_i$ as spanning $\mathbb{C}^6$. To complete the construction of a GLSM matrix factorisation, we reintroduce the point-like field $p$. With an appropriate choice of GLSM gauge charges, we then recover the GLSM matrix factorisation $Q_2$. Furthermore, we then find that the hybrid B-brane $\mathfrak{B}_2^{(2,7/2,3,1)}$ lifts to the complex of Wilson line branes given above. We thus conclude that the empty hybrid B-branes on the octic indeed correspond to empty B-branes in the GLSM. We now move on to consider the GLSM brane associated to the canonical matrix factorisation (149). A GLSM matrix factorisation representing this is

$$Q_3 = \sum_{i=3}^6 x_i\eta_i + \frac{1}{4}\frac{\partial W}{\partial x_i}\overline{\eta}_i.$$

(279)

An example of an associated complex of Wilson line branes, $\mathcal{B}_3$, is given by

$$\mathcal{W}(0,0)_0 \xleftrightarrow{\quad} \begin{array}{c} \mathcal{W}(1,0)^{\oplus 3}_{1-2\kappa_1} \\ \oplus \\ \mathcal{W}(1,-2)_{1-2\kappa_1+4\kappa_2} \end{array} \xleftrightarrow{\quad} \dots \xleftrightarrow{\quad} \begin{array}{c} \mathcal{W}(3,0)_{3-6\kappa_1} \\ \oplus \\ \mathcal{W}(3,-2)^{\oplus 3}_{3-6\kappa_1+4\kappa_2} \end{array} \xleftrightarrow{\quad} \mathcal{W}(4,-2)_{4-8\kappa_1+4\kappa_2}.$$

(280)

In the geometric phase, this reduces to an empty brane, see appendix C, where this brane is labeled $\mathcal{B}_a$. The brane factor evaluated in the hybrid phase is

$$f_{\mathcal{B}_3}|_{hyb} = (1 + \gamma^2\gamma^{-\delta})^3(1 + e^{-2H}\gamma^2\gamma^{-\delta}).$$

(281)

Expanding this in terms of $H$ shows that all six components $f_{\mathcal{B},\delta}^a$ are non-zero. This means that the brane has both D0 and D2-components in $B$.

Next, we identify this GLSM brane with a certain lift of a hybrid B-brane belonging to the family $\mathfrak{B}_3^{(a_0, r_0, \alpha_0)}$. Applying the procedure of section 3.7, we obtain the following realisation of the corresponding global matrix factorisation

$$Q_3 = x_6\eta_1 + (x_1^8 + x_2^8)x_6^3\overline{\eta}_1 + \sum_{a=3}^5 (x_a\eta_a + x_a^3\overline{\eta}_a).$$

(282)

Upon making an appropriate choice of boundary GLSM gauge charges, and reintroducing the point-like field $p$, we indeed obtain the GLSM matrix factorisation $Q_3$. According to our prescription in section 5.1.3, the $U(1)_1$ gauge charges of the GLSM B-brane are given by the orbifold charges, while the $U(1)_2$ charges are given by the corresponding hybrid gauge charges. We thus obtain the following family of complexes of Wilson line branes

$$
\mathcal{W}(\alpha_0-4,a_0+2)_{r_0-2} \underset{\longrightarrow}{\overset{\mathcal{W}(\alpha_0-3,a_0+2)^{\oplus 3}_{r_0-\frac{3}{2}}}{\longleftarrow}} \oplus \underset{\longrightarrow}{\overset{}{\longleftarrow}} \cdots \underset{\longrightarrow}{\overset{\mathcal{W}(\alpha_0-1,a_0+2)_{r_0-\frac{1}{2}}}{\longleftarrow}} \oplus \underset{\longrightarrow}{\overset{}{\longleftarrow}} \mathcal{W}(\alpha_0,a_0)_{r_0}.
$$

with $\mathcal{W}(\alpha_0-3,a_0)_{r_0-\frac{3}{2}}$ and $\mathcal{W}(\alpha_0-1,a_0)^{\oplus 3}_{r_0-\frac{1}{2}}$.
$$(283)$$

Hence, the complex of Wilson line branes given above corresponds to a lift of the hybrid B-brane $\mathfrak{B}_3^{(-2,2,4)}$. Had we chosen a different set of GLSM boundary gauge charges, we could have instead obtained a lift with final term $\mathcal{W}(0,-2)_2$ rather than $\mathcal{W}(4,-2)_2$. As mentioned previously, such choices can be utilised to lift hybrid B-branes into specific charge windows.

As it is expected that the canonical matrix factorisation does not represent a brane with minimal charge, we move on to consider permutation-type matrix factorisations in the fibre direction. We take the following GLSM brane:

$$
Q_4 = (x_3 - e^{-\frac{i\pi}{4}}x_4)\eta_1 + ph(x_3,x_4)\overline{\eta}_1 + x_5\eta_2 + px_5^3\overline{\eta}_2 + x_6\eta_3 + px_6^3(x_1^8 + x_2^8)\overline{\eta}_3, \quad (284)
$$

where $h(x_3,x_4) = \prod_{k=1}^{3}(x_3 - e^{-i\pi\frac{2k+1}{4}}x_4)$. Among the branes described by this matrix factorisation, we consider a GLSM brane which we label by $\mathcal{B}_4$:

$$
\mathcal{W}(0,0)_0 \underset{\longrightarrow}{\overset{\mathcal{W}(1,0)^{\oplus 2}_{1-2\kappa_1}}{\longleftarrow}} \underset{\oplus}{} \underset{\longrightarrow}{\overset{\mathcal{W}(2,0)_{2-4\kappa_1}}{\longleftarrow}} \underset{\oplus}{} \underset{\longrightarrow}{\overset{}{\longleftarrow}} \mathcal{W}(3,-2)_{3-6\kappa_1+4\kappa_2}.
$$

with $\mathcal{W}(1,-2)_{1-2\kappa_1+4\kappa_2}$ and $\mathcal{W}(2,-2)^{\oplus 2}_{2-4\kappa_1+4\kappa_2}$.
$$(285)$$

This brane is already band restricted. In the geometric phase, this brane is a D2-brane in the base direction, see the brane $\mathcal{B}_c$ in appendix C. It is also a D2 with respect to the base $B$ in the hybrid phase. This is confirmed by expressing the brane factor in the hybrid phase

$$
f_{\mathcal{B}_4}|_{hyb} = (1-\gamma^{-\delta})^3 + H\gamma^{-\delta}(1-\gamma^{-\delta})^2 + \mathcal{O}(H^2). \quad (286)
$$

Since $f_{\mathcal{B}_4}|_{hyb}$ is non-zero for all "narrow" values of $\delta$, we conclude that the brane has D0 and D2-charges with respect to $B$. To identify this GLSM B-brane as the lift of a hybrid B-brane, we recall the family of hybrid B-branes $\mathfrak{B}_4^{(a_0,r_0,\alpha_0)}$. The corresponding global matrix factorisation can be lifted to $Q_4$, while $\mathfrak{B}_4^{(a_0,r_0,\alpha_0)}$ itself can be lifted to

$$
\mathcal{W}(\alpha_0-3,a_0+2)_{r_0-\frac{3}{2}} \underset{\longrightarrow}{\overset{\mathcal{W}(\alpha_0-2,a_0+2)^{\oplus 2}_{r_0-1}}{\longleftarrow}} \oplus \underset{\longrightarrow}{\overset{\mathcal{W}(\alpha_0-1,a_0+2)_{r_0-\frac{1}{2}}}{\longleftarrow}} \oplus \underset{\longrightarrow}{\overset{}{\longleftarrow}} \mathcal{W}(\alpha_0,a_0)_{r_0}.
$$

with $\mathcal{W}(\alpha_0-2,a_0)_{r_0-1}$ and $\mathcal{W}(\alpha_0-1,a_0)^{\oplus 2}_{r_0-\frac{1}{2}}$.
$$(287)$$

We thus see that

$$
\pi^{hyb}(\mathcal{B}_4) = \mathfrak{B}_4^{(-2,3/2,3)}. \quad (288)
$$

The next example is a matrix factorisation that is of permutation-type in the base coordinates rather then the fibre coordinates. We expect this to yield a D0-brane in the base. We look at the matrix factorisation

$$
Q_5 = (x_1 - e^{-\frac{i\pi}{8}}x_2)\eta + px_6^4 g(x_1,x_2)\overline{\eta} + \sum_{i=3}^{5}x_i\eta_i + px_i^3\overline{\eta}_i, \quad (289)
$$

where $g(x_1, x_2) = \prod_{k=1}^{7}(x_1 - e^{-i\pi\frac{2k+1}{8}}x_2)$. An associated complex of Wilson line branes $\mathcal{B}_5$ is

$$\mathcal{W}(0,0)_0 \rightleftarrows \begin{matrix} \mathcal{W}(1,0)^{\oplus 3}_{1-2\kappa_1} \\ \oplus \\ \mathcal{W}(0,1)_{1-2\kappa_2} \end{matrix} \rightleftarrows \cdots \rightleftarrows \begin{matrix} \mathcal{W}(3,0)_{3-6\kappa_1} \\ \oplus \\ \mathcal{W}(2,1)^{\oplus 3}_{3-4\kappa_1-2\kappa_2} \end{matrix} \rightleftarrows \mathcal{W}(3,1)_{4-6\kappa_1-2\kappa_2}.$$

(290)

This brane is automatically band restricted, and thus globally defined on the moduli space. The geometric information of this brane is a minimally charged D0-brane on the Calabi-Yau, see the object $\mathcal{B}_b$ in appendix C. We compute the brane factor in the hybrid phase to confirm that this brane describes a D0-brane on $B$. We find

$$f_{\mathcal{B}_5}|_{hyb} = -H(1-\gamma^{-\delta})^3,$$

(291)

so the base D2-charges $f^{1}_{\mathcal{B},\delta}$ are all zero, as expected. We now identify this brane as a lift of a member of the family $\mathfrak{B}_5^{(a_0, r_0, \alpha_0)}$. Applying the procedure of section 5.1.3, we find that the corresponding global matrix factorisation lifts to $Q_5$, while the family of complexes can be lifted to

$$\mathcal{W}(\alpha_0-3,a_0-1)_{r_0-\frac{5}{2}} \rightleftarrows \begin{matrix} \mathcal{W}(\alpha_0-2,a_0-1)^{\oplus 3}_{r_0-2} \\ \oplus \\ \mathcal{W}(\alpha_0-3,a_0)_{r_0-\frac{3}{2}} \end{matrix} \rightleftarrows \cdots \rightleftarrows \begin{matrix} \mathcal{W}(\alpha_0,a_0-1)_{r_0-1} \\ \oplus \\ \mathcal{W}(\alpha_0-1,a_0)^{\oplus 3}_{r_0-\frac{1}{2}} \end{matrix} \rightleftarrows \mathcal{W}(\alpha_0,a_0)_{r_0}.$$

(292)

We thus see that

$$\pi^{hybrid}(\mathfrak{B}_5) = \mathfrak{B}_5^{(1,5/2,3)}.$$

(293)

Finally, we consider a matrix factorisation of permutation-type in both, the base and the fibre direction:

$$Q_6 = (x_1 - e^{-\frac{i\pi}{8}}x_2)\eta + px_6^4 g(x_1, x_2)\overline{\eta} + (x_3 - e^{-\frac{i\pi}{4}}x_4)\eta_3 + ph(x_3, x_4)\overline{\eta}_3 + x_5\eta_4 + px_5^3\overline{\eta}_4.$$ (294)

Making the usual choice for the global shifts, an example of a brane described by this matrix factorisation is

$$\mathcal{W}(0,0)_0 \rightleftarrows \begin{matrix} \mathcal{W}(1,0)^{\oplus 2}_{1-2\kappa_1} \\ \oplus \\ \mathcal{W}(0,1)_{1-2\kappa_2} \end{matrix} \rightleftarrows \begin{matrix} \mathcal{W}(2,0)_{2-4\kappa_1} \\ \oplus \\ \mathcal{W}(1,1)^{\oplus 2}_{2-2\kappa_1-2\kappa_2} \end{matrix} \rightleftarrows \mathcal{W}(2,1)_{3-4\kappa_1-2\kappa_2}.$$

(295)

We denote this brane by $\mathcal{B}_6$. This brane is also automatically band restricted. In the geometric phase, this brane has the interpretation of a D2-brane in the direction of the K3 fibre. This brane has the label $\mathcal{B}_d$ in appendix C. Computing the brane factor in the hybrid phase, we obtain

$$f_{\mathcal{B}_6,\delta}|_{hyb} = -H(1-\gamma^{-\delta})^2 + \mathcal{O}(H^2).$$

(296)

In the hybrid phase, this brane only has D0-brane charges with respect to the base $B$. Because the geometric brane is a D2-brane in the K3-fibre direction, which undergoes a Landau-Ginzburg transition in the hybrid phase, the geometric content of the brane in the hybrid phase is that of a D0-brane. This GLSM B-brane is a lift of a member of the family $\mathfrak{B}_6^{(a_0, r_0, \alpha_0)}$ of hybrid B-branes. We first note that the corresponding global matrix factorisation lifts to $Q_6$. We then

see that the family $\mathfrak{B}_6^{(a_0,r_0,\alpha_0)}$ can be lifted to the following family of GLSM B-branes

$$
\mathcal{W}(\alpha_0-2,a_0-1)_{r_0-2} \xleftarrow{\quad} 
\begin{array}{c}
\mathcal{W}(\alpha_0-1,a_0-1)^{\oplus 2}_{r_0-\frac{3}{2}} \quad \mathcal{W}(\alpha_0,a_0-1)_{r_0-1} \\
\oplus \xleftarrow{\quad} \oplus \xleftarrow{\quad} \mathcal{W}(\alpha_0,a_0)_{r_0}, \\
\mathcal{W}(\alpha_0-2,a_0)_{r_0-1} \quad \mathcal{W}(\alpha_0-1,a_0)^{\oplus 2}_{r_0-\frac{1}{2}}
\end{array}
\tag{297}
$$

and so

$$
\pi^{hyb}(\mathcal{B}_6) = \mathfrak{B}_6^{(1,2,2)}.
\tag{298}
$$

### 5.3.4 Basis of branes

Our strategy to find a basis of branes that generate the RR-charge lattice is the same as for the bicubic example. To find a set of minimally charged branes, we start off with a suitable set of permutation-type matrix factorisations, and then generate further objects by shifting the gauge charge $q_1$, associated to the fibre direction of the complexes. This produces images under Landau-Ginzburg monodromy in the fibre. In contrast to the bicubic example, the shifts exhibit a $\mathbb{Z}_4$-periodicity. It turns out that the permutation-type D2-branes in the geometric phase can be used as building blocks for a basis of minimally charged branes. We have already encountered these branes in the previous section. We build three branes out of the brane $\mathcal{B}_4$ in (285). Starting with $\mathcal{B}_4$, we consider the shifts obtained by tensoring the above complex by $\mathcal{W}(1,0)_{1-2\kappa_1}$, and by $\mathcal{W}(2,0)_{2-4\kappa_1}$, and denote the three resulting branes by $\mathcal{B}_{4,j}$ ($j=0,1,2$). Making use of (268), the brane factors evaluated in the hybrid phase are

$$
f_{\mathcal{B}_{4,j}}|_{hyb} = (-1)^j \gamma^{-j\delta}(1-\gamma^{-\delta})^2(1-\gamma(-\delta)e^{-2H}).
\tag{299}
$$

The second building block is $\mathcal{B}_6$ in (295). As we have seen, this is a D0-brane in the base. We can make the same two shifts as for the other matrix factorisation to create two more branes which we denote by $\mathcal{B}_{6,j}$. The corresponding brane factors are

$$
f_{\mathcal{B}_{6,j}}|_{hyb} = (-1)^j \gamma^{-j\delta}(1-\gamma^{-\delta})^2(1-e^H).
\tag{300}
$$

We have constructed six branes in the hybrid phase, and we claim that they form a basis in the hybrid phase. We collect evidence for this in the next section by establishing a connection to a basis of branes in the geometric phase.

### 5.3.5 D-brane transport

We evaluate the brane factors of the band-restricted geometric branes, and express the result in terms of the basis of hybrid branes proposed above. This determines the analytic continuation of the geometric branes to the hybrid phase. The geometric branes can be found in appendix C. We have already encountered some of the branes as hybrid branes, but we will repeat some of the results for convenience.

Since the geometric D0 brane $\mathcal{B}_b$ specified in appendix C is automatically band-restricted, we can simply evaluate the brane factor in the hybrid phase. We get

$$
f_{\mathcal{B}_b,\delta}|_{hyb} = -H(1-\gamma^{-\delta})^3 + \mathcal{O}(H^2).
\tag{301}
$$

Note that $f_{\mathcal{B}_b,\delta=0 \mod 4}|_{hyb} = 0$, consistent with the expectations for broad sectors. Similarly, we find for the geometric D2-branes $\mathcal{B}_c = \mathcal{B}_4$ and $\mathcal{B}_d = \mathcal{B}_6$

$$
f_{\mathcal{B}_c,\delta}|_{hyb} = (1-\gamma^{-\delta})^3 + 2H\gamma^{-\delta}(1-\gamma^{-\delta})^2 + \mathcal{O}(H^2),
\tag{302}
$$

$$
f_{\mathcal{B}_d,\delta}|_{hyb} = -H(1-\gamma^{-\delta})^2 + \mathcal{O}(H^2).
\tag{303}
$$

The band-restricted brane factors of the geometric D4-branes $\mathcal{B}_e^w$ and $\mathcal{B}_f^w$ discussed in appendix C are

$$f_{\mathcal{B}_e^w,\delta}|_{hyb} = -\left((e^H - 1)\gamma^{-3\delta}(\gamma^\delta - 1)\left(\gamma^{2\delta} + e^{2H}(\gamma^\delta - 1)^2 + 3\right)\right), \tag{304}$$

$$f_{\mathcal{B}_f^w,\delta}|_{hyb} = -\gamma^{-3\delta}(\gamma^\delta - 1)^2\left(-\gamma^\delta + 2e^{2H}(\gamma^\delta - 1) + e^{4H}(\gamma^\delta - 1) - 3\right). \tag{305}$$

The expansions in terms of $H$ are not very enlightening.

For the band-restricted D6-brane $\mathcal{B}_g^w$ in (C.30), the brane factor evaluates to

$$\begin{aligned} f_{\mathcal{B}_6^w,\delta}|_{hyb} &= -(-1 + \gamma^{-\delta})(e^{2H} - 2\gamma^{-\delta}e^{2H} + \gamma^{-2\delta}(3 + e^{2H})) \\ &= -2(-1 + \gamma^{-\delta})(1 - \gamma^{-\delta} + 2\gamma^{-2\delta}) - 2(-1 + \gamma^{-\delta})^3 H + \mathcal{O}(H^2). \end{aligned} \tag{306}$$

With the band-restricted brane factors of the geometric branes evaluated in the hybrid phase, we can compute the analytic continuation matrix from the geometric to the hybrid phase with our chosen band restriction rule. We find

$$\begin{pmatrix} f_{\mathcal{B}_b} \\ f_{\mathcal{B}_c} \\ f_{\mathcal{B}_d} \\ f_{\mathcal{B}_e^w} \\ f_{\mathcal{B}_f^w} \\ f_{\mathcal{B}_g^w} \end{pmatrix}\Bigg|_{hyb} = \begin{pmatrix} 0 & 0 & 0 & 1 & 1 & 0 \\ 1 & 0 & 0 & 0 & 0 & 0 \\ 0 & 0 & 0 & 1 & 0 & 0 \\ 0 & 0 & 0 & 1 & 2 & 1 \\ -3 & -2 & 1 & 6 & 16 & 2 \\ 1 & 1 & 0 & -2 & -4 & -2 \end{pmatrix} \begin{pmatrix} f_{\mathcal{B}_{4,0}} \\ f_{\mathcal{B}_{4,1}} \\ f_{\mathcal{B}_{4,2}} \\ f_{\mathcal{B}_{6,0}} \\ f_{\mathcal{B}_{6,1}} \\ f_{\mathcal{B}_{6,2}} \end{pmatrix}\Bigg|_{hyb} . \tag{307}$$

This can be inverted over the integers.

# 6 Outlook

In this work, we have studied B-type D-branes in good hybrid models and established a connection to geometric branes through GLSMs. There are many directions for further research.

The main focus of our work was to give a physics derivation of B-branes in hybrid models, and to study explicit constructions and examples. There is a lot more structure to the category of B-branes which would be interesting to investigate further. So far, we have only discussed the objects of the category, i.e. the branes themselves, but not the open string states, i.e. the morphisms. We expect that the generalisation from what is known in Landau-Ginzburg and non-linear sigma models should be straightforward. Once one has the open string states, one could study bound states of branes and disk correlation functions. In particular, there should be a hybrid-generalisation of the Kapustin-Li formula [51] for boundary three-point functions and bulk-boundary two-point functions. The latter should be compatible with the Chern character [24] computed by the hemisphere partition function for those states that can be lifted to the GLSM. It should also be possible to study the $A_\infty$-structure of the category of hybrid B-branes, generalising results from Landau-Ginzburg models [52–57]. Some of these structures have recently been studied for hybrids in the context of homological projective duality [17]. It would be interesting to expand on this.

Another direction is to explore boundary states in the IR CFTs associated to hybrid models. In the Landau-Ginzburg case, there is a dictionary between certain boundary states in the CFT and matrix factorisations [38, 39, 49, 58]. When considering the fibre direction in a hybrid, one can construct matrix factorisations that have properties which should be associated to some Recknagel-Schomerus or permutation-type boundary states. It would be interesting to see how to fully characterise the boundary states associated to the hybrid branes that we have constructed here.

From the mirror perspective, the central charge of a brane is an expansion in terms of the periods of the of the mirror Calabi-Yau. The charges of mi- namally charged branes we have found give an integral basis of periods. In the one-parameter case, integral bases for certain hybrid models arising at "K-points" of the complex structure moduli space have been proposed [59]. This should be compatible with the results that we have found.

Finally, it should prove worthwhile to consider branes in more complicated hybrid models. In this work, we have focused on examples with a $\mathbb{P}^1$-base, simply because the best-studied models have this structure. Our results hold for more general models. There are known examples which relate to Calabi-Yaus in toric varieties which have $\mathbb{P}^2$ and $\mathbb{P}^3$ bases (see eg. [37]). We expect that our brane constructions generalise in a straightforward manner. There is also a class of models which describe non-commutative resolutions of branched double covers [29,60–62]. These models have interpretations both, as hybrid models and in terms of geometry. It would be interesting to study these dual descriptions at the level of branes in more detail. There is also the possibility of studying hybrid B-branes over bases which are not projective space. For example, in the closed string setting [6] discussed hybrid models with bases given by Hirzebruch surfaces. Beyond toric geometry, Calabi-Yau hybrid models also appear as phases of non-abelian Calabi-Yau GLSMs. Studying branes and D-brane transport in such models as was recently considered in [17,18], will provide new explicit constructions of categorical equivalences.

# Acknowledgments

We would like to thank Joseph McGovern and Emanuel Scheidegger for discussions and comments on the manuscript. We thank the participants of the GLSMs@30 workshop at the Simons Center for Geometry and Physics in May 2023 for helpful conversations. We thank MATRIX Institute and SCGP for hospitality.

**Funding information**   JK is supported by the Australian Research Council Discovery Project DP210101502 and the Australian Research Council Future Fellowship FT210100514.

# A   Coordinate conventions

In this appendix, we discuss explicit coordinates on complex projective space $\mathbb{P}^n$, as well as transition functions for holomorphic line bundles over $\mathbb{P}^n$, and explicit coordinates on the total space of these bundles.

## A.1   Coordinates for $\mathbb{P}^n$

The complex projective space $\mathbb{P}^n$ is given by

$$\mathbb{P}^n = ((\mathbb{C})^{n+1} - \{0\})/\sim, \tag{A.1}$$

where the equivalence relation is given by

$$(z_0, z_1, \ldots, z_n) \sim (\lambda z_0, \lambda z_1, \ldots, \lambda z_n), \text{ for } \lambda \in \mathbb{C}^*. \tag{A.2}$$

We denote the equivalence classes in $\mathbb{P}^n$ by the homogeneous coordinates $[z_0 : z_1 : \ldots : z_n]$, and introduce local coordinate charts

$$U_i = \{[z_0 : z_1 : \ldots : z_n] | z_i \neq 0\}, \quad i = 0, 1, \ldots, n. \tag{A.3}$$

Local coordinates in the charts $U_i$ and $U_{i+1}$ can then be given by

$$u_k = \begin{cases} \frac{z_{k-1}}{z_i}, & k \leq i, \\ \frac{z_k}{z_i}, & k > i, \end{cases} \text{ and } v_k = \begin{cases} \frac{z_{k-1}}{z_{i+1}}, & k \leq i+1, \\ \frac{z_k}{z_{i+1}}, & k > i+1, \end{cases} \quad k = 1, 2, \ldots, n, \tag{A.4}$$

respectively. The transition functions between these coordinate charts are then

$$v_{i+1} = u_{i+1}^{-1}, \text{ and } v_k = u_k u_{i+1}^{-1}, \text{ for } k \neq i+1, \tag{A.5}$$

which are clearly holomorphic.

## A.2 Holomorphic line bundles on $\mathbb{P}^n$

All holomorphic line bundles on $\mathbb{P}^n$ are of the form $\mathcal{O}(m) \to \mathbb{P}^n$ for some $m \in \mathbb{Z}$ (see e.g [25]). Recalling the transition functions of the tautological bundle $\mathcal{O}(-1) \to \mathbb{P}^n$

$$\tau_{ij} : U_i \cap U_j \to \mathbb{C}^*, \quad [z_0 : z_1 : \ldots : z_n] \mapsto \frac{z_i}{z_j}, \tag{A.6}$$

we can explicitly construct the holomorphic line bundles on $\mathbb{P}^n$ as follows. The total space is given by

$$\text{tot}(\mathcal{O}(m) \to \mathbb{P}^n) := \coprod_{i=0}^{n+1} (U_i \times \mathbb{C})/\sim, \tag{A.7}$$

where the equivalence relation is given by

$$([z_0 : z_1 : \ldots : z_n], \tilde{z}) \sim ([z_0 : z_1 : \ldots : z_n], \tau_{ij}^{-m}([z_0 : z_1 : \ldots : z_n])\tilde{z}), \tag{A.8}$$

for all $[z_0 : z_1 : \ldots : z_n] \in U_i \cap U_j$, $\tilde{z} \in \mathbb{C}$, and $0 \leq i, j \leq n$. Note that we use integer powers of $\tau_{ij}$ to ensure holomorphicity, and that we use $\tau_{ij}^{-m}$ to define $\mathcal{O}(m) \to \mathbb{P}^n$ due to the fact that the $\tau_{ij}$ are the transition functions of $\mathcal{O}(-1) \to \mathbb{P}^n$. The projection map is given by

$$\pi_m : \text{tot}(\mathcal{O}(m) \to \mathbb{P}^n) \to \mathbb{P}^n. \tag{A.9}$$

A system of local trivialisations is given by the following open sets

$$\mathcal{U}_i = \pi_m^{-1}(U_i). \tag{A.10}$$

It will be useful to utilise explicit coordinates on the total space of these line bundles. To this end, we state the following theorem.

**Theorem 1.** *Let* $E = \coprod_i (U_i \times \mathbb{C}^k)/\sim$ *be the total space of a rank-k vector bundle over a complex n-dimensional manifold M, where* $\sim$ *is an equivalence relation. Let* $(U_i, \tau_i)$ *be charts for M, and* $\phi_i$ *be local trivialisations for E. Then, charts for E as a manifold are given by* $(\mathcal{U}_i, \rho_i)$*, where* $\mathcal{U}_i := \pi^{-1}(U_i)$*, and*

$$\rho_i := (\tau_i \times \mathbb{1}) \circ \phi_i^{-1} : \mathcal{U}_i \to U_i \times \mathbb{C}^k \to \mathbb{C}^n \times \mathbb{C}^k \cong \mathbb{C}^{n+k}. \tag{A.11}$$

For the case at hand, we find that $\text{tot}(\mathcal{O}(m) \to \mathbb{P}^n)$ can be given the following explicit coordinates in the charts $\mathcal{U}_i$ and $\mathcal{U}_{i+1}$

$$u_j = v_j v_{i+1}^{-1}, \text{ for } j \neq i, \quad u_{i+1} = v_{i+1}^{-1}, \text{ and } x_{ju} = v_{i+1}^{-m} x_{jv}, \tag{A.12}$$

where the coordinate changes of the base coordinates are familiar from our discussion of $\mathbb{P}^n$, and where we have chosen fibre coordinates $x_{ju}$ and $x_{jv}$ in the patches $\mathcal{U}_i$ and $\mathcal{U}_{i+1}$ respectively.

# B  Bicubic GLSM

In this section, we give further details on the GLSM that has $\mathbb{P}^5[3,3]$ as a geometric phase. We construct a basis of geometric branes and grade-restrict them to the window (209) so that they can be analytically continued to the hybrid phase.

## B.1  Picard-Fuchs equation and large volume periods

When evaluating the hemisphere partition function in the geometric phase, the result can be expressed in terms of the mirror periods $\varpi_i$ satisfying the Picard-Fuchs equation $\mathcal{L}\varpi_i = 0$. The Picard-Fuchs-operator is

$$\mathcal{L} = \theta^4 - 9z(3\theta + 1)^2(3\theta + 2)^2 . \tag{B.1}$$

It can be found as AESZ 4 in the database of Calabi-Yau differential operators [63, 64]. We identify $z = e^{-t}$ with $t = \zeta - i\theta$ the FI-theta parameter of the GLSM. The periods are

$$\varpi_0 = 1 + 36z + 8100z^2 + 2822400z^3 + 1200622500z^4 + 572679643536z^5 + O\left(z^6\right), \tag{B.2}$$

$$\varpi_1 = \frac{1}{2\pi i}\left(\varpi_0 \log z + a_1(z)\right), \tag{B.3}$$

$$\varpi_2 = \frac{1}{(2\pi i)^2}\left(\varpi_0 \log^2 z + 2a_1(z)\log z + a_2(z)\right), \tag{B.4}$$

$$\varpi_3 = \frac{1}{(2\pi i)^3}\left(\varpi_0 \log^3 z + 3a_1(z)\log^2 z + 3a_2(z)\log z + a_3(z)\right), \tag{B.5}$$

with

$$a_1(z) = 180z + 46170z^2 + 16860480z^3 + 7346926125z^4 + 3555982065636z^5 + O\left(z^6\right), \tag{B.6}$$

$$a_2(z) = 234z + \frac{212949z^2}{2} + 45545768z^3 + \frac{171136538325z^4}{8} + \frac{540897828396093z^5}{50} + O\left(z^6\right), \tag{B.7}$$

$$a_3(z) = -1404z - \frac{494991z^2}{2} - 61765324z^3 - \frac{304319320335z^4}{16} + O\left(z^5\right), \tag{B.8}$$

## B.2  Hemisphere partition function in the geometric phase

In the $\zeta \gg 0$-phase, we fix $\kappa = 0$ and take

$$i\sigma = -k + \varepsilon, \qquad k \in \mathbb{Z}_{\geq 0}, \tag{B.9}$$

to account for the contributing poles. Using the reflection formula for the gamma function, the integral can be written as

$$Z_{D^2}^{\zeta \gg 0} = C\sum_{k\geq 0}\oint d\varepsilon \frac{\Gamma(1 + 3k - 3\varepsilon)^2}{\Gamma(1 + k - \varepsilon)^6}\frac{\pi^6(-1)^{6k}}{\sin^6 \pi\varepsilon}e^{t(-k+\varepsilon)}f_{\mathcal{B}}(-i(-k+\varepsilon)). \tag{B.10}$$

We define

$$\varepsilon = -\frac{H}{2\pi i}, \tag{B.11}$$

where we identify $H$ with the hyperplane class of $\mathbb{P}^5$. Denoting our Calabi-Yau complete intersection as $X \subset \mathbb{P}^5$, we further note that the Todd class is

$$\mathrm{Td}(X) = \frac{H^6}{(3H)^2}\frac{(1 - e^{-3H})^2}{(1 - e^{-H})^6} . \tag{B.12}$$

Furthermore, note that

$$\int_X g(H) = \int_{\mathbb{P}^5} (3H)^2 g(H) = 9 \oint dH \frac{1}{H^4} g(H) \,. \tag{B.13}$$

We can then write

$$Z_{D^2}^{\zeta \gg 0} = -(2\pi i)^5 \widetilde{C} \sum_{k \geq 0} \oint dH (-1)^{6n+6} \frac{(3H^2)}{H^6} \frac{\Gamma\left(1 + 3k + \frac{3H}{2\pi i}\right)^2}{\Gamma\left(1 + k + \frac{H}{2\pi i}\right)^6} \frac{H^6}{(3H)^2} \frac{(1 - e^{-3H})^2}{(1 - e^{-H})^6}$$

$$\times e^{-tk - tq} e^{-(t - 6\pi i)\frac{H}{2\pi i}} \frac{f_{\mathcal{B}}}{(1 - e^{3H})^2} \,. \tag{B.14}$$

Now we define $t^{\zeta \gg 0} = t - 6\pi i$. Furthermore, we recall that the $I$-function is

$$I_X(u) = \frac{\Gamma\left(1 + \frac{H}{2\pi i}\right)^6}{\Gamma\left(1 + \frac{3H}{2\pi i}\right)^2} \sum_{k \geq 0} (-1)^{6k} \frac{\Gamma\left(1 + 3k + \frac{3H}{2\pi i}\right)^2}{\Gamma\left(1 + k + \frac{H}{2\pi i}\right)^6} u^{\frac{H}{2\pi i} + k} \,, \tag{B.15}$$

and that the Gamma class is given by

$$\widehat{\Gamma}_X = \frac{\Gamma\left(1 + \frac{H}{2\pi i}\right)^6}{\Gamma\left(1 + \frac{3H}{2\pi i}\right)^2} \,. \tag{B.16}$$

Further, note that for a Calabi-Yau $X$, there is the relation $\widehat{\Gamma}_X \widehat{\Gamma}_X^* = \mathrm{Td}(X)$, where $\widehat{\Gamma}^* = \widehat{\Gamma}|_{i \to -i}$. Following [15], we define

$$\mathrm{ch}(\mathcal{B}^{\zeta \gg 0}) = \frac{f_{\mathcal{B}}}{(1 - e^{3H})^2} \,. \tag{B.17}$$

Then we get

$$Z_{D^2}^{\zeta \gg 0} = -(2\pi i)^5 \widetilde{C} \int_X \widehat{\Gamma}_X^* I_X(e^{-t^{\zeta \gg 0}}) \mathrm{ch}(\mathcal{B}^{\zeta \gg 0}) \,. \tag{B.18}$$

### B.3 Geometric branes

We aim to investigate how a set of geometric branes can be analytically continued to the hybrid phase. For this purpose, we construct a set of GLSM matrix factorisations that reduces to a basis of branes which generate the charge lattice in the geometric phase. This is very similar to the hypersurface case [5]. We will grade restrict all branes with respect to the window (209) so that they are also valid in the hybrid phase. We compute matrix factorisations of the superpotential

$$W = p_1(x_1^3 + x_2^3 + x_3^3 - 3\alpha x_4 x_5 x_6) + p_2(x_3^3 + x_4^3 + x_5^3 - 3\alpha x_1 x_2 x_3) \,. \tag{B.19}$$

For some of the branes we construct, the cubic equations can be replaced by their generic versions. For the explicit construction of the branes we define

$$f_{ij} = x_i + x_j \,, \qquad g_{ij} = x_i^2 - x_i x_j + x_j^2 \,, \tag{B.20}$$

so that $f_{ij} g_{ij} = x_i^3 + x_j^3$ (no summation over indices!).

The first object we identify is the empty brane in the geometric phase. It describes a skyscraper sheaf localised at $x_1 = \ldots = x_6 = 0$, and corresponds to the following GLSM matrix factorisation:

$$Q_a = \sum_{i=1}^{6} x_i \eta_i + \frac{1}{3} \frac{\partial W}{\partial x_i} \overline{\eta}_i \,. \tag{B.21}$$

This is the same as the canonical matrix factorisation, $Q_3$, (237) in the hybrid phase. With a suitable choice of gauge and R-charges for the Clifford vacuum $|0\rangle$ to fix M, one example of a brane associated to this matrix factorisation is given in terms of the following Koszul-type complex of Wilson line branes

$$\mathcal{B}_a: \quad \mathcal{W}(0)_0 \xleftrightarrow{\qquad} \mathcal{W}(1)^{\oplus 6}_{1-2\kappa} \xleftrightarrow{\qquad} \mathcal{W}(2)^{\oplus 15}_{2-4\kappa} \xleftrightarrow{\qquad} \ldots \xleftrightarrow{\qquad} \mathcal{W}(6)_{6-12\kappa}. \tag{B.22}$$

This brane is not grade-restricted to any window. The brane factor is

$$f_{\mathcal{B}_a} = (-1 + e^{2\pi(-i\kappa+\sigma)})^6. \tag{B.23}$$

When evaluating the brane factors in the geometric phase, we always fix $\kappa = 0$. It is easy to show that $Z^{\zeta \gg 0}_{D^2}(\mathcal{B}_a) = 0$.

An example of a D0-brane of minimal charge is given by the equations

$$(x_1 + x_2) = x_3 = x_4 = x_5 = x_6 = 0, \tag{B.24}$$

which implies $G^1_3 = G^2_3 = 0$ for the choice (206) of the hypersurface equations. Hence, this describes a brane on the Calabi-Yau. A GLSM lift of this brane can be given by the matrix factorisation

$$\begin{aligned} Q_b = & f_{12}\eta_1 + p_1 g_{12}\overline{\eta}_1 + x_3\eta_2 + (p_1 x_3^2 - 3\alpha p_2 x_1 x_2)\overline{\eta}_2 + x_4\eta_3 + p_2 x_4^2 \overline{\eta}_3 \\ & + x_5\eta_4 + p_2 x_5^2 \overline{\eta}_4 + x_6\eta_5 + (p_2 x_6^2 - 3\alpha p_1 x_4 x_5)\overline{\eta}_5. \end{aligned} \tag{B.25}$$

We pick a brane $\mathcal{B}_b$ associated to this matrix factorisation given by the following complex of Wilson line branes:

$$\mathcal{W}(0)_0 \xleftrightarrow{\qquad} \mathcal{W}(1)^{\oplus 5}_{1-2\kappa} \xleftrightarrow{\qquad} \mathcal{W}(2)^{\oplus 10}_{2-4\kappa} \xleftrightarrow{\qquad} \ldots \xleftrightarrow{\qquad} \mathcal{W}(5)_{5-10\kappa}. \tag{B.26}$$

The corresponding brane factor is

$$f_{\mathcal{B}_b} = (1 - e^{2\pi(-i\kappa+\sigma)})^5. \tag{B.27}$$

This brane is automatically grade restricted and hence defined throughout the moduli space. Evaluating the hemisphere partition function in the geometric phase, we get with a suitable choice of overall normalisation

$$Z^{\zeta \gg 0}_{D^2}(\mathcal{B}_b) = \varpi_0, \tag{B.28}$$

where the $\varpi_i$ are defined above.

A D2-brane on the Calabi-Yau with minimal charge can be specified, for instance, by the equations

$$f_{12} = f_{45} = x_3 = x_6 = 0, \tag{B.29}$$

which have the following GLSM realisation:

$$\begin{aligned} Q_c = & f_{12}\eta_1 + p_1 g_{12}\overline{\eta}_1 + f_{45}\eta_2 + p_2 g_{45}\overline{\eta}_2 + x_3\eta_3 + (p_1 x_3^2 - 3\alpha p_2 x_1 x_2)\overline{\eta}_3 \\ & + x_6\eta_4 + (p_2 x_6^2 - 3\alpha p_1 x_4 x_5)\overline{\eta}_4. \end{aligned} \tag{B.30}$$

We choose the brane $\mathcal{B}_c$ specified by

$$\mathcal{W}(0)_0 \xleftrightarrow{\qquad} \mathcal{W}(1)^{\oplus 4}_{1-2\kappa} \xleftrightarrow{\qquad} \mathcal{W}(2)^{\oplus 6}_{2-4\kappa} \xleftrightarrow{\qquad} \mathcal{W}(3)^{\oplus 4}_{3-6\kappa} \xleftrightarrow{\qquad} \mathcal{W}(4)_{4-8\kappa}. \tag{B.31}$$

The corresponding brane factor is

$$f_{\mathcal{B}_c} = (1 - e^{2\pi(-i\kappa+\sigma)})^4. \tag{B.32}$$

This brane is also automatically grade restricted and the hemisphere partition function evaluates to

$$Z_{D^2}^{\zeta \gg 0}(\mathcal{B}_c) = \varpi_1 - \varpi_0 \,. \tag{B.33}$$

There is not really a generic choice for a D4-brane, but we can model a matrix factorisation which corresponds to intersecting the Calabi-Yau with a divisor $f_\beta = \sum_i \beta_i x_i$. We get

$$Q_d = G_3^1 \eta_1 + p_1 \overline{\eta}_1 + G_3^2 \eta_2 + p_2 \overline{\eta}_2 + f_\beta \eta_3 \,. \tag{B.34}$$

We consider the following GLSM brane $\mathcal{B}_d$ associated to this matrix factorisation:

$$\mathcal{W}(0)_0 \; \underset{\longleftarrow}{\overset{\longrightarrow}{}} \; \begin{matrix} \mathcal{W}(3)_{1-6\kappa}^{\oplus 2} \\ \oplus \\ \mathcal{W}(1)_{1-2\kappa} \end{matrix} \; \underset{\longleftarrow}{\overset{\longrightarrow}{}} \; \begin{matrix} \mathcal{W}(6)_{2-12\kappa} \\ \oplus \\ \mathcal{W}(4)_{2-8\kappa}^{\oplus 2} \end{matrix} \; \underset{\longleftarrow}{\overset{\longrightarrow}{}} \; \mathcal{W}(7)_{3-14\kappa} \,. \tag{B.35}$$

This brane is not grade-restricted. The corresponding brane factor is

$$f_{\mathcal{B}_d} = -(-1 + e^{2\pi(-i\kappa+\sigma)})^3 (1 + e^{2\pi(-i\kappa+\sigma)} + e^{4\pi(-i\kappa+\sigma)})^2 \,, \tag{B.36}$$

which yields

$$Z_{D^2}^{\zeta \gg 0}(\mathcal{B}_d) = \frac{9}{2}\varpi_2 + \frac{9}{2}\varpi_1 + \frac{15}{4}\varpi_0 \,. \tag{B.37}$$

To grade restrict to our window of choice, we have to remove $\mathcal{W}(6)$ and $\mathcal{W}(7)$ by binding copies of $\mathcal{B}_a(k)_r$, where the parenthesis means to shift the complex so that the left-most entry is $\mathcal{W}(k)_r$. We call the resulting brane $\mathcal{B}_d^w$. Following the algorithm of [65] based on [5], we obtain a complex of Wilson line branes for $\mathcal{B}_d^w$:

$$\begin{aligned} f_{\mathcal{B}_d^w} &= f_{\mathcal{B}_d} + f_{\mathcal{B}_a(1)_{-2-2\kappa}} + 5 f_{\mathcal{B}_a(0)_{-3}} \\ &= -3(-1 + e^{2\pi(-i\kappa+\sigma)})^3 (2 - 4e^{2\pi(-i\kappa+\sigma)} + 5e^{4\pi(-i\kappa+\sigma)}) \,. \end{aligned} \tag{B.38}$$

Both brane factors yield the same $Z_{D^2}^{\zeta \gg 0}$.

The D6-brane representing the structure sheaf of the Calabi-Yau is associated with the GLSM matrix factorisation

$$Q_e = G_3^1 \eta_1 + p_1 \overline{\eta}_1 + G_3^2 \eta_2 + p_2 \overline{\eta}_2 \,. \tag{B.39}$$

This is the same as the empty brane $Q_1$ in the hybrid phase. Grade restriction only requires us to bind one copy of $Q_a$. This gives a brane $\mathcal{B}_e^w$

$$f_{\mathcal{B}_e^w} = f_{\mathcal{B}_e} - f_{\mathcal{B}_a} = 3e^{2\pi(-i\kappa+\sigma)}(1 - e^{2\pi(-i\kappa+\sigma)})^2 (2 - e^{2\pi(-i\kappa+\sigma)} + 2e^{4\pi(-i\kappa+\sigma)}) \,. \tag{B.40}$$

The hemisphere partition function evaluates to

$$Z_{D^2}^{\zeta \gg 0}(\mathcal{B}_e) = \frac{3}{2}\varpi_3 + \frac{9}{4}\varpi_1 - \frac{18i\zeta(3)}{\pi^3}\varpi_0 \,. \tag{B.41}$$

Comparing this with the general form of the central charge of a structure sheaf

$$Z(\mathcal{O}_X) = \frac{H^3}{6}\varpi_3 + \frac{c_2 H}{24}\varpi_1 + i\frac{\zeta(3)c_3}{(2\pi i)^3}\varpi_0 \,, \tag{B.42}$$

we confirm that

$$H^3 = 9 \,, \qquad c_2 H = 54 \,, \qquad c_3 = -144 \,. \tag{B.43}$$

The set $(\mathcal{B}_b, \mathcal{B}_c, \mathcal{B}_d^w, \mathcal{B}_e^w)$ gives a basis of branes in the geometric phase associated to an integral basis of periods on the mirror. Since the branes are grade restricted, they are well-defined objects in the hybrid phase.

# C Octic GLSM

## C.1 Picard-Fuchs equations and large volume periods

Details about the explicit form of the periods in the geometric phases can be found in many references, see eg. [31, 32]. The Picard-Fuchs system associated to the Calabi-Yau geometry is

$$
\begin{aligned}
\mathcal{L}_1 &= \theta_1^2(\theta_1 - 2\theta_2) - 4z_1(4\theta_1 + 3)(4\theta_1 + 2)(4\theta_1 + 1), \\
\mathcal{L}_2 &= \theta_2^2 - z_2(2\theta_2 - \theta_1 + 1)(2\theta_1 - \theta_1).
\end{aligned}
\tag{C.1}
$$

We make the following identification with the GLSM FI-theta parameters: $z_i = e^{-t_i}$, and $\theta_i = z_i \frac{\partial_i}{\partial z_i}$. The modulus $t_1$ is associated to the K3-fibre, while the modulus $t_2$ is associated to the base $\mathbb{P}^1$. One can solve these equations to obtain a basis of periods

$$
\left( \varpi_0, \varpi_{1,1}, \varpi_{1,2}, \varpi_{2,1}, \varpi_{2,2}, \varpi_3 \right).
\tag{C.2}
$$

The leading terms of the periods are

$$
\varpi_0 = 1 + \dots,
\tag{C.3}
$$

$$
\varpi_{1,1} = \frac{1}{(2\pi i)} \log z_1 + \dots,
\tag{C.4}
$$

$$
\varpi_{1,2} = \frac{1}{(2\pi i)} \log z_2 + \dots,
\tag{C.5}
$$

$$
\varpi_{2,1} = \frac{1}{(2\pi i)^2} (\log z_1)^2 + \dots,
\tag{C.6}
$$

$$
\varpi_{2,2} = \frac{1}{(2\pi i)^2} \log z_1 \log z_2 + \dots,
\tag{C.7}
$$

$$
\varpi_3 = \frac{1}{(2\pi i)^3} \left( (\log z_1)^3 + \frac{3}{2} (\log z_1)^2 \log z_2 \right) + \dots
\tag{C.8}
$$

## C.2 Hemisphere partition function in the geometric phase

We state the results for the geometric phase only briefly. A detailed discussion of which poles contribute in the large volume phase has been given in [36] for the sphere partition function. This is also valid for the hemisphere. Fixing the R-charge ambiguity to $\kappa_1 = \kappa_2 = 0$, all contributing poles are accounted for if we consider

$$
i\sigma_1 = -k_1 + \varepsilon_1, \qquad i\sigma_2 = -k_2 + \varepsilon_2, \quad k_i \in \mathbb{Z}_{\geq 0}.
\tag{C.9}
$$

Inserting and using the reflection formula, the hemisphere partition function in the geometric phase can be rewritten as

$$
\begin{aligned}
Z_{D^2}^{geom} = \overline{C} \oint d^2\varepsilon \, &\frac{\pi^5 (-1)^{3k_1 + 2k_2}}{\sin^3 \pi\varepsilon_1 \sin^2 \pi\varepsilon_3} \frac{\Gamma(1 + 4k_1 - 4\varepsilon_2)}{\Gamma(1 + k_1 - \varepsilon_1)^3 \Gamma(1 + k_2 - \varepsilon_2)^2} \\
&\times e^{t_1(-k_1 + \varepsilon_1)} e^{t_2(-k_2 + \varepsilon_2)} f_{\mathcal{B}}(-i(-k_1 + \varepsilon_1), -i(-k_2 + \varepsilon_2)) \\
&\times \begin{cases} \Gamma(-k_1 + 2k_2 - \varepsilon_1 - 2\varepsilon_2), & k_1 < 2k_2, \\ \frac{\pi(-1)^{k_1 + k_2}}{\sin \pi(\varepsilon_1 - 2\varepsilon_2)\Gamma(1 + k_1 - 2k_2 - \varepsilon_1 + 2\varepsilon_2)}, & k_1 \geq 2k_2. \end{cases}
\end{aligned}
\tag{C.10}
$$

Evaluating the residues, one obtains an expansion in terms of the mirror periods.

## C.3 Geometric branes

In the geometric phase, we can use standard constructions to obtain GLSM lifts of simple geometric branes and use the hemisphere partition function to confirm that these objects have the expected charges. We use the form (256) of the superpotential.

We begin by describing an empty brane in the geometric phase. Since we are only concerned with going between the geometric and hybrid phases and since the band-restriction rule (257) only constrains the gauge charge $q_1$, we only need one empty brane for our purposes. This is easily found by constructing the GLSM brane associated to the following component of the deleted set associated to the geometric phase:

$$x_3 = x_4 = x_5 = x_6 \,. \tag{C.11}$$

This is encoded by the matrix factorisation

$$Q_a = \sum_{i=3}^{6} x_i \eta_i + \frac{1}{4} \frac{\partial W}{\partial x_i} \overline{\eta}_i \,. \tag{C.12}$$

This is the canonical matrix factorisation $Q_3$, (279), in the hybrid phase. The empty branes $\mathcal{B}_a(k_1, k_2)_r$ in the geometric phase are thus the same as the complex $\mathcal{B}_3$ in (280), and its shifts by $k_1, k_2$ in the gauge charges and by $r$ in the R-charge.

Let us start with a D0-brane. One way to get a point-like object is to impose

$$(x_1 - e^{-\frac{i\pi}{8}} x_2) = 0\,, \qquad x_3 = x_4 = x_5 = 0 \,. \tag{C.13}$$

A lift of this brane to the GLSM is

$$Q_b = (x_1 - e^{-\frac{i\pi}{8}} x_2)\eta + p x_6^4 g(x_1, x_2)\overline{\eta} + \sum_{i=3}^{5} x_i \eta_i + p x_i^3 \overline{\eta}_i \,, \tag{C.14}$$

which we have already encountered as $Q_5$ in section 5.3.3. It describes a hybrid brane which is a D0-brane in the base. The brane $\mathcal{B}_b$ associated to this matrix factorisation is the complex of Wilson line branes $\mathcal{B}_5$, (290). It is easy to check that, upon a suitable choice of the constant $\overline{C}$, the hemisphere partition function evaluates to

$$Z_{D^2}^{geom}(\mathcal{B}_b) = \varpi_0 \,. \tag{C.15}$$

Moving on to the D2-branes, there is an obvious choice, namely to construct a D2-brane that wraps the base and another one that wraps a 2-cycle in the fibre. Let us begin with the D2 that wraps the base. To describe this, we are not allowed to have any conditions on the base coordinates $x_1, x_2$. In view of constructing a matrix factorisation, this can be achieved by factoring off $x_6$. We thus consider the following equations:

$$(x_3 - e^{-\frac{i\pi}{4}} x_4) = 0\,, \quad x_5 = 0\,, \quad x_6 = 0 \,. \tag{C.16}$$

This can be lifted to the GLSM matrix factorisation $Q_4$, (284) in section 5.3.3. To describe our D2-brane, we take the GLSM brane $\mathcal{B}_4$ given by (285). To comply with our labelling systematics, we also give this brane the name $\mathcal{B}_c$ whenever we refer to it in the context of the geometric phase. The hemisphere partition function in the geometric phase is

$$Z_{D^2}^{geom}(\mathcal{B}_c) = -\varpi_{1,2} + \varpi_0 \,. \tag{C.17}$$

A D2-brane which wraps a 2-cycle in the fibre can be characterised by

$$(x_1 - e^{-\frac{\pi i}{8}} x_2) = 0\,, \quad (x_3 - e^{-\frac{\pi i}{4}} x_4) = 0\,, \quad x_5 = 0 \,. \tag{C.18}$$

An associated matrix factorisation is the same as $Q_6$ in (294). We take the GLSM brane $\mathcal{B}_6$ in (295) and give it the name $\mathcal{B}_d$ when referring to the geometric D2-brane. The hemisphere partition function evaluates to the expected result:

$$Z_{D^2}^{geom}(\mathcal{B}_d) = -\varpi_{1,1} + \varpi_0 \, . \tag{C.19}$$

One standard construction of D4-branes is to intersect the Calabi-Yau with some divisor of the ambient space. The canonical choices are to take a divisor in the base or one in the fibre. In the former case, we want to lift

$$G_{(4,0)} = 0 \, , \qquad f(x_1, x_2) = 0 \, , \tag{C.20}$$

to a matrix factorisation, where $G_{(4,0)}$ is the hypersurface equation and $f(x_1, x_2)$ is a linear equation in the base coordinates. This is a D0 in the base and a D4 in the K3-fibre. A GLSM lift is

$$Q_e = G_{(4,0)}\eta_1 + p\overline{\eta}_1 + f(x_1, x_2)\eta_2 + 0 \cdot \overline{\eta}_2 \, . \tag{C.21}$$

As an associated complex of Wilson line branes, we choose

$$
\mathcal{W}(0,0)_0 \; \underset{\longleftarrow}{\overset{\longrightarrow}{}} \;
\begin{array}{c}
\mathcal{W}(4,0)_{1-8\kappa_1} \\
\oplus \\
\mathcal{W}(0,1)_{1-2\kappa_2}
\end{array}
\; \underset{\longleftarrow}{\overset{\longrightarrow}{}} \; \mathcal{W}(4,1)_{2-8\kappa_1-2\kappa_2} \, ,
\tag{C.22}
$$

which we call $\mathcal{B}_e$. This GLSM brane is not band-restricted. The hemisphere partition function evaluates to

$$Z_{D^2}^{geom}(\mathcal{B}_e) = 2\varpi_{2,1} + \varpi_0 \, . \tag{C.23}$$

A second D4-brane can be characterised by

$$G_{(4,0)} = 0 \, , \qquad f(x_3, x_4, x_5) = 0 \, , \tag{C.24}$$

where $f(x_3, x_4, x_5)$ is linear in the variables. An obvious GLSM lift of this brane is

$$Q_f = G_{(4,0)}\eta_1 + p\overline{\eta}_1 + f(x_3, x_4, x_5)\eta_2 + 0 \cdot \overline{\eta}_2 \, . \tag{C.25}$$

We then take the following associated complex of Wilson line branes:

$$
\mathcal{B}_f : \quad \mathcal{W}(0,0)_0 \; \underset{\longleftarrow}{\overset{\longrightarrow}{}} \;
\begin{array}{c}
\mathcal{W}(4,0)_{1-8\kappa_1} \\
\oplus \\
\mathcal{W}(1,0)_{1-2\kappa_1}
\end{array}
\; \underset{\longleftarrow}{\overset{\longrightarrow}{}} \; \mathcal{W}(5,0)_{2-10\kappa_1} \, .
\tag{C.26}
$$

This is also not band restricted. The hemisphere partition function evaluated in the geometric phase is

$$Z_{D^2}^{geom}(\mathcal{B}_f) = 4(\varpi_{2,2} + \varpi_{2,1}) + 2(\varpi_{1,2} + 2\varpi_{1,1}) + \frac{11}{3}\varpi_0 \, . \tag{C.27}$$

Finally, the D6-brane describing the structure sheaf is given by the matrix factorisation

$$Q_g = G_{(4,0)}\eta + p\overline{\eta} \, . \tag{C.28}$$

As expected, this is the same as the matrix factorisation $Q_1$ (269) for the empty brane in the hybrid phase. To construct our D6-brane, we pick the complex of Wilon line branes $\mathcal{B}_1$, (270), and assign to it the additional label $\mathcal{B}_g$. The hemisphere partition function evaluates to

$$Z_{D^2}^{geom}(\mathcal{B}_g) = -\frac{4}{3}\varpi_3 - \varpi_{1,2} - \frac{7}{3}\varpi_{1,1} + \frac{168i\zeta(3)}{8\pi^2}\varpi_0 \, . \tag{C.29}$$

Up to an overall sign, which can be fixed by shifting the R-charge by 1, amounting to exchanging the brane with its antibrane, this is the expected result for the central charge of the structure sheaf of this Calabi-Yau. This brane is not band restricted, and thus is not well-defined beyond the geometric phase.

To be able to transport all these branes to the hybrid phase, we have to band-restrict the D4- and D6-branes. To band restrict the D6-brane, we bind a shifted version of the empty brane, $\mathcal{B}_a(0,2)_{2+4\kappa_2}$ as follows

$$
\begin{array}{ccccc}
\mathcal{W}(0,0)_0 & \rightleftarrows & \mathcal{W}(4,0)_{1-8\kappa_1} & & \\
 & & & \searrow^{1} & \\
\cdots \rightleftarrows & \begin{array}{c}\mathcal{W}(2,2)^{\oplus 3}_{-4\kappa_1-4\kappa_2} \\ \oplus \\ \mathcal{W}(2,0)^{\oplus 3}_{-4\kappa_1}\end{array} & \rightleftarrows & \begin{array}{c}\mathcal{W}(3,2)_{1-6\kappa_1-4\kappa_2} \\ \oplus \\ \mathcal{W}(3,0)^{\oplus 3}_{1-6\kappa_1}\end{array} & \rightleftarrows \mathcal{W}(4,0)_{2-8\kappa_1} .
\end{array}
\tag{C.30}
$$

This brane, which we label by $\mathcal{B}_g^w$, is band-restricted, and has the same central charge in the geometric phase as the original D6-brane. Band restriction of the D4-branes is only slightly more complicated, and we omit the details. We call the resulting branes $\mathcal{B}_e^w$ and $\mathcal{B}_f^w$. We take the set of D0-D6-branes $(\mathcal{B}_b, \mathcal{B}_c, \mathcal{B}_d, \mathcal{B}_e^w, \mathcal{B}_f^w, \mathcal{B}_g^w)$ as our basis of geometric branes. Since all of them are band-restricted with respect to $w$, they are also well-defined in the hybrid phase.

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
