# Peer review of "B-type D-branes in hybrid models"

_SciPost Physics, doi:SciPost Phys. 17, 165 (2024)_

## Round 2 · Referee Report · Anonymous (Referee 1) · 2024-10-27

Strengths

Generally the paper is of good quality. The authors carefully develop the subject in a well readable manner, and spice it up with a variety of detailed computations for examples. These are at the core of the paper, and help the moderately informed physicist to grasp the considerable mathematical complexity of the subject. Upon cursory inspection the computations appear to be correct (there is a minor typo in the second-to-last paragraph of Sect. 6.)

Weaknesses

The subject of hybrid branes is quite specialised but definitely of interest to people working in the field.

Report

The authors extend the framework of matrix factorisations, which describe topological B-branes in N=(2,2) superconformal field theories, to so-called hybrid branes. "Hybrid" denotes here the general phase of two-dimensional
N=(2,2) superconformal field theories, which can be thought of as a mixture between the geometrical sigma-model phase and the Landau-Ginzburg phase; concretely this can be described as a sigma-model with additional superpotential. Correspondingly, also at the boundary where the D-branes live, general D-branes can be hybrid mixtures displaying both geometric and algebraic components; the latter being described in terms of matrix factorisations which in a concrete sense are fibered over the geometry. The total hybrid object is described as a "global matrix factorisation".

Given that hybrid phases of two dimensional N=(2,2) superconformal field theories are the most generic ones, it is evidently important to extend this structure to D-branes at the boundary. It is somewhat surprising that the issue wasn't addressed in depth before, and in retrospective the present work was overdue.

Recommendation

Publish (easily meets expectations and criteria for this Journal; among top 50%)

---

## Round 2 · Referee Report · Anonymous (Referee 2) · 2024-10-31

Strengths

This is a well-written paper, which discusses matrix factorizations in hybrid Landau-Ginzburg models. Examples of such matrix factorizations have been discussed elsewhere, but this is the first detailed, general, first-principles account of which I'm aware. It explicitly discussses their construction as a set of ordinary matrix factorizations in Landau-Ginzburg models in sections of the hybrid, together with overlap data, and also gives detailed computations.

Weaknesses

A minor point is that this is a rather technical area, of interest primarily to experts. Another minor point is that it might be nice to have a few more examples, but, the paper is already rather long.

Report

Briefly, I recommend this paper for publication. It meets the journal's standards.

Recommendation

Publish (easily meets expectations and criteria for this Journal; among top 50%)

---

## Editorial Decision

published